# Atmospheric pollution from ships and their impacts on local air quality in a port site in Shanghai

Xinning Wang[1], Yin Shen[1], Yanfen Lin[1], Jun Pan[1], Yan Zhang[2], Peter K.K. Louie[3], Mei Li[4], Qingyan Fu[1]

[1] Shanghai Environmental Monitoring Center, Shanghai 200030, P.R. China.
[2] Department of Environmental Science and Engineering, Fudan University, Shanghai 200433, P.R. China.
[3] Hong Kong Environmental Protection Department, Hong Kong, P.R. China.
[4] Institute of Mass Spectrometer and Atmospheric Environment, Jinan University, Guangzhou, 510632, China.

*Correspondence to*: Qingyan Fu (qingyanf@semc.gov.cn)

**Abstract.**

Growing shipping activities in port areas have generated negative impacts on climate, air quality and human health. To better evaluate the environmental impact of ship emissions, an experimental characterization of air pollutions from ships was conducted in Shanghai port in the summer of 2016. The ambient concentrations of gaseous ($NO$, $NO_2$, $SO_2$, $O_3$) and particulate concentrations ($PM_{2.5}$), particle size distributions and chemical composition of individual particles from ship emission were continuously monitored for 3 months. Ship emission plumes were visible at the port site in terms of clear peaks of gaseous and particulate matter concentrations. The $SO_2$ and vanadium particles numbers were found to correlate best with ship emissions in Shanghai port. Single particle data showed that ship emission particles at the port site mainly concentrated in smaller size range ($< 0.4$ μm) where their number contributions were more important than their mass contributions to ambient PM. Composition of ship emission particles at the portside suggested they were mostly fresh particles, with their mass spectra dominated by peaks of sulfate, elemental carbon (EC), metallic composition of V, Ni, Fe and Ca, and very low nitrate signals. For some cases of plumes, the gaseous $NO_x$ composition in plumes has the evidence of atmospheric transformation by ambient $O_3$, resulted ozone depletion in this area. Quantitative estimation in the present study show that in port region ship emissions contributed 36.4 % $SO_2$, 0.7 % $NO$, 5.1 % $NO_2$, -0.9 % $O_3$, 5.9 % $PM_{2.5}$, 49.5 % vanadium particles if land-based emissions were included, and 57.2 % $SO_2$, 71.9 % $NO$, 30.4 % $NO_2$, -16.6 % $O_3$, 27.6 % $PM_{2.5}$, 77.0 % vanadium particles if land-based emissions were excluded.

**Keywords**

Ship emission; Shanghai port; emission source contribution; SPAMS

## 1 Introduction

Ship emissions constitute an important type of gaseous and particulate source in global scale. It has become important in recent years due to the increasing shipping activities. The annual ship emission of $NO_x$, $SO_2$ and $PM_{2.5}$ were estimated to be $2 \times 10^7$,

$9.7 \times 10^6$, and $1.5 \times 10^6$ tons respectively (Johansson et al., 2017). The large emission intensities from ships have generated great burdens to regional and global environment (Fuglestvedt et al., 2009) and negative impacts to human health (Corbett et al., 2007). In marine environment, ships were found to be the dominant contributor to surface $NO_2$ and $SO_2$ concentrations (Dalsoren et al., 2009). Altered clouds properties were identified in these areas along ship cruising route by satellites (Petzold

et al., 2008;Coggon et al., 2012), which could generate impacts to earth radiation budget and climate. For coastal or port regions, ship emissions have made significant contributions to atmospheric $NO_x$, $SO_2$ and PM in local environment (Donateo et al., 2014; Merico et al., 2017).

The typical fuel that ships burn is Residual Fuel Oil (RFO) with high sulfur content. Combustion of RFO in ship engines produces high concentration of gaseous and particulate pollutants including $NO_x$, $SO_2$, Elemental Carbon (EC), Organic

Carbon (OC), sulfate and trace metals. Emission Factors of these pollutants from various ship types have been determined to develop emission inventories (Moldanova et al., 2013;Buffaloe et al., 2014;Cappa et al., 2014). In ambient measurement, however, the chemical and physical attributes of ship emissions are critical for identifying ship emission and assess their impacts (Murphy et al., 2009). Owing to more stringent regulations against ship emissions by restricting sulfur content in fuel, the detection of ship emissions relying only on individual tracers is unreliable because of the changing composition of RFO in

different areas. To better identify ship emission in this context, multi-component characterizations including both gaseous and particulate are necessary in studies of field measurements (Xiao et al., 2018;Viana et al., 2009).

In Yangtze River Delta (YRD) region in China the shipping activities has increased significantly due to intensified international trades in recent years. The accompanying potential environmental and health problems from ship emissions in YRD are well recognized (Chen et al., 2018;Zhang et al., 2017;Fu et al., 2017). Global distributions of ship emission indicate that South and

Eastern China Sea regions have the highest pollutants emission densities (Johansson et al., 2017). As shown in an emission inventory in China, shipping traffics emitted about 1.3 Tg $SO_2$, 1.9 Tg $NO_x$ and 0.16 Tg PM in 2013, with $NO_x$ and PM being equivalent to $\sim$ 34 % and 29 % of total mobile vehicle emissions in China (Fu et al., 2017). To cope with severe air pollution caused by ship emissions, Shanghai government has initiated Domestic Emission Control Areas (DECA) in YRD. At present stage, according to YRD DECA regulations, the sulfur content of any fuel used on board while berthing at Shanghai port shall

not exceed 0.5 % (m/m), except for the first hour after arrival and the last hour before departure, which has taken effect on April 1, 2016. This limitation level of sulphur is still higher than the implemented legislation in many harbors/ports in Europe and US (0.1%) (IMO, 2017). The DECA measure was currently implemented mainly in three major shipping areas including PRD, Pearl River Delta - PRD, and Bohai Rim region in China. Efficiency of the ECA measures has been tested in other places (Contini et al., 2015;Merico et al., 2017). It was shown that the control strategies in sulphur in fuel could generate synergetic

reduction in both $SO_2$ and primary PM release from ships. The benefits of DECA measure in YRD were also suggested by the reduction of $SO_2$ concentration at several monitoring sites in port areas. There is a published study which dealt with the effectiveness of DECA in PRD region, estimating that the DECA measure could result average reduction of 9.54% $SO_2$ and 2.7% $PM_{2.5}$ in land areas (Liu et al., 2018).

An quantitative estimation of ship emission contribution to air quality is needed for better understanding of its environmental roles and controlling policies. In East Asia, an earlier emission inventory in Shanghai area estimated that the ship emissions were 58160, 51180, 6960 tons/year for $NO_x$, $SO_2$ and PM respectively in 2003 (Yang et al., 2007). Over the last decade Shanghai port throughput of goods has dramatically increased. In 2010, the total ship emissions of $NO_X$, $SO_2$ and $PM_{2.5}$ in

YRD have grown to $7.1 \times 10^5$, $3.8 \times 10^5$ and $5.1 \times 10^4$ tons/year, respectively (Fan et al., 2016). A more recent study estimated that the primary $PM_{2.5}$ from ships ranged from 0.63 to 3.58 μg/m$^3$, accounting for 4.23 % of the total $PM_{2.5}$ in Shanghai Port (Zhao et al., 2013), based on a marine port measurement off coast of Shanghai. Such information of port in coastal areas is needed since their closer distance to the urban area of Shanghai city.

In the summer of 2016, an in-site sampling campaign focusing on ship emissions was performed at Shanghai Port. Gaseous

and particulate matters concentrations were monitored for 3 months to identify and characterize the ship emissions in Shanghai port areas. Based on the measurement data, quantitative assessment of the contribution of ship emissions to portside air quality was performed. Ship emission aerosol particles were characterized by a single particle aerosol mass spectrometer (SPAMS) which was deployed at the same site in parallel to the gaseous measurement. The SPAMS was utilized to identify ship emission aerosol composition and size with high temporal resolution, which is useful in detecting fast transient ship plumes, as

demonstrated previously (Ault et al., 2010;Healy et al., 2009). In addition, the ship emission particle signatures obtained here is valuable in SPAMS source apportionment in future studies. The present study represents a comprehensive characterization of gaseous and particulate ship emissions in YRD and serves to provide essential scientific information for future evidence-based ship emission control policies.

## 2 Experimental

**2.1 Sampling site**

The Waigaoqiao Port (31.337° N, 121.665° E) locates in the northeast of Shanghai city (Figure 1) and is the largest port in China. The port has about 7 km of docks (3 km north section and 4 km south section). In 2016 the port has yearly traffic of 367 M-tons of goods and container volume of 37.13 million TEU (Twenty-foot Equivalent Unit). Ship categories in port consist of container vessel (62.4 %), tug (18.6 %), oil tanker (9.0 %), bulk (1.8 %), Ro-Ro (1.7 %) and other ships (6.5 %)

(private data from authority). A power plant and a shipbuilding factory reside between the north and south section of port, which have their own docks. The portside air monitoring station locates on the south bank of Yangtze River, 400 m away from the nearest dock.

Gaseous and particulate monitor instruments were installed in the station room. The station was equipped with a main sampling tube extending through the roof. The outlets of the main sampling tube was 1m above the station roof and 3.5m above the

ground. Ship emission plumes could influence the site in wind direction of about 300°-0°-120° sector (Figure 1). In the summer season the prevailing wind direction of the site is southeast direction. In the supplementary file the wind rose during the sampling period is provided (Figure S1). In ~55% of time the site was under the impact from port emissions. To the south and

west of site there were intense road traffics of container trucks and the Shanghai outer ring. Except emissions from ships in port directions, the site could also receive important influences from traffics when inland wind prevails.

## 2.2 Gaseous, PM$_{2.5}$ and peripheral measurement

From Jun-21 to Sep-21, 2016, the portside gaseous pollutants NO-NO$_2$-NO$_x$, SO$_2$, and O$_3$ were monitored continuously with a suit of Thermo Scientific analyzers (NO-NO$_2$-NO$_x$, model 42i; SO$_2$, model 43i; O$_3$, 49i). Verification and calibration of the instruments were performed regularly by zero checks (through a zero air generator) and span checks (through standard NO$_2$ and SO$_2$ gas of known concentrations; the O$_3$ standard was generated through a calibration photometer system); The PM$_{2.5}$ concentrations were monitored by oscillating microbalance method (Thermo TEOM 1405-F). Calibration of TEOM was not relied on standard, for the aerosol mass on a filter was monitored by the oscillation frequency change of the tapered element over specified time. The regular maintenance of TEOM included the replacement of filters before their mass loadings approached 100%. The flow rate of TEOM was checked using a flowmeter. The lower detection limits of these instruments are: 0.4 µg/m$^3$ (NO, NO$_2$); 0.5 µg/m$^3$ (SO$_2$); 0.5 µg/m$^3$ (O$_3$); 1 µg/m$^3$ (PM$_{2.5}$). Weather conditions (temperature, humidity, pressure, wind speed and direction) were monitored by a mini-weather station installed on the rooftop of the station. The weather station sensor was installed about 1 m above the station roof and 3.5 m above the ground. Data from all the instruments and the weather monitor was managed in a customized database and were set to 5 min resolution. Atmospheric pollutants concentrations in Shanghai city area, including gaseous pollutants and PM$_{2.5}$ concentrations, were monitored concurrently at 9 national air quality monitoring stations in 1h resolution. The averaged pollutants concentrations at these stations during the same period were included for comparison.

## 2.3 Single particle aerosol mass spectrometer (SPAMS)

From Jun-21 to Sep-21 in 2016, a SPAMS (HeXin Analytical Instrument Co., Ltd., China) was applied to characterized single particle composition and size of ambient aerosol at the port site (Li et al., 2011). Operation principle of SPAMS is briefly described. Ambient aerosol was drawn into SPAMS vacuum system through a critical orifice with limited aerosol flow. The particles then entered an aerodynamic focusing lens (AFL) where they were focused into thin beam, with transiting velocities in the vacuum as a function of their aerodynamic size. In SPAMS sizing region the particles consecutively encounter two continuous laser beams (532 nm wavelength), reflect light and generate signals in two photomultiplier tubes. The time lag between two PMT signals was used to calculate particle velocity and to trigger a ionizing laser pulse (266 nm wavelength) at appropriate time to ionize the same particle. Chemical composition of particles was determined by a dual polar time-of-flight mass spectrometer to record signals for both negative and positive ions. The time lags between two PMTs of PSL particles of known sizes were used to calibrate the aerodynamic size of ambient particles. Particle size, dual polar mass spectra, particle reflecting signals from two PMTs were saved for each particle. A PM$_{2.5}$ cyclone was placed at the outlet of sampling tube on the roof of the station to remove particles > 2.5 µm before analysed by SPAMS.

Specific composition in particles, such as vanadium, is identified by their characteristic mass peaks in particle spectra. Particles producing vanadium peaks were labelled as vanadium particles. SPAMS quantifies their concentrations in a semi-quantitative manner through the number of detected particles in specific duration of time. Considering that the aerosol flow was introduced into SPAMS at fixed flow rate (0.1L/min), the detected particle numbers (or particle detecting velocity) could be utilized as indication of ambient particle concentrations. In ambient sampling it was shown particle numbers in SPAMS were positively correlated with ambient $PM_{2.5}$ concentrations ($R^2$=0.69 in this study). In present study, we used particle detecting velocity of vanadium containing particles as the metric of their concentrations. To derive ambient particle number concentrations from SPAMS particle numbers, we need to consider the efficiency issues of SPAMS on AFL transmission, laser detection and laser ionization (Wenzel et al., 2003).

## 2.4 SPAMS data analysis

The temporal resolution of SPAMS (seconds or minutes) makes it suitable to couple with online gaseous data in identifying ship emissions. The fluctuations of gaseous concentrations, shifting of wind directions and the arrival of emission plumes, were well responded by SPAMS data. Additionally, present study took advantage of the ability of SPAMS to identify individual ship emission particles by their characteristic composition. Composition patterns of ship emission particles were identified firstly and were then applied to extract desired particles from all analysed particles. The temporal trends, size distribution, chemical composition, and wind roses of the extracted particles could be examined in further detail.

During the sampling period of 3 months SPAMS generated a large particle set (>2.3 million particles were chemically analysed). To identify ship emission particles from all the analysed particles, we applied a combined method of peak searching and clustering algorithm. Specifically, the individual particle mass spectra were visually inspected to get a general MS pattern during ship plumes. It is not feasible to inspect the large amount of spectra exhaustively. Instead, we used the concurrent $SO_2$ concentrations to locate ship emission plumes when sharp $SO_2$ peaks occurred, which is typical for RFO combustions (Murphy et al., 2009;Merico et al., 2016). Compared with non-plumes periods, the most indicative peaks during plumes occurred at $V^+$(51), $VO^+$(67), $Fe^+$(56), $Ni^+$(58) and serial peaks of elemental carbon at $C_n^+$(n=1,2,3…,12) in the positive mass spectrum (Ault et al., 2010;Healy et al., 2009;Ault et al., 2009). In this study the vanadium mass peaks (peak $V^+$(51) and $VO^+$(67))) were determined to be a prerequisite to indicate ship particles during plumes. Further notes on this particle identification method of ship emission particles were provided in supplementary file. We applied a rough searching with peak criteria of m/z = 51 and 67 (i.e., just the existence of mass peak at 51 and 67 with no peak area limitation) to search all possible candidates from the entire dataset. This particle criteria is not stringent because particles producing organic peaks at the same nominal mass (e.g. $C_4H_3^+$(51), $C_4H_3O^+$(67)) could interfere and may enter into searched clusters. Then the ART-2a algorithm (Song et al., 1999) was applied to the searched particles to generate sub-clusters of similar MS patterns (Vigilance=0.85; Learning=0.05; Iteration=20). By inspecting composition, size and wind rose patterns of sub-clusters, a small fraction of outlier particles from non-shipping emission sources were thus picked out and discarded.

**2.5 Evaluation of ship emission contribution**

The calculations method of ship emission contributions used in this study, which was originally developed by (Contini et al., 2011), is based on the extraction of ship emission plumes from background concentrations of pollutants :

$$\varepsilon_A = \frac{\Delta C_A F_{plm}}{C_A}$$

Where: $\varepsilon_A$, ship emission contributions of pollutants $A$; $\Delta C_A$, the difference between average concentrations during plumes and non-plumes; $F_{plm}$, fraction of cases of plumes; $C_A$, the average concentration of pollutant $A$ during reference period. The uncertainties of $\varepsilon_A$ determined in this method could arise from several factors, such as the definition of port direction sector, the definition of plumes (the threshold level that discriminate plumes and the background), and pollutants and wind field measurements. This study estimated the uncertainties by subjecting $\varepsilon_A$ to slight adjustment of the port directions by ±10° and

pollutants threshold levels by 20% to inspect its variations. To conform to the original work (Contini et al., 2011), calm wind periods (wind speed < 0.5 m/s) were considered in the evaluation of uncertainties (either excluding or including calm wind periods).

**3 Results and discussions**

**3.1 Identification and description of ship emission plumes**

In the vicinity of port, the ship emission pollutant concentrations often produce obvious peaks in relatively short period (Figure 2). These peaks were caused by ship emission plumes relating to shipping activities such as arrival, hoteling and departure, which typically persist for a few (mostly 3-6) hours. The ambient $SO_2$, NO, $NO_2$, $O_3$ and $PM_{2.5}$ concentrations during a typical period (Aug 27-29) were shown in Figure 2 to illustrate several plumes. For comparison the averaged $SO_2$ concentration in Shanghai city and vanadium particle number concentrations of the same period are provided. During plume periods, ambient

$SO_2$ concentration peaks correlated well with vanadium particles numbers detected by SPAMS. The $PM_{2.5}$ peaks during plumes were not always unclear as in Figure 2. In supplementary file we present another period of $PM_{2.5}$, SO2 and vanadium particle concentrations to demonstrate stronger $PM_{2.5}$ peaks (Figure S3). The synchronized gaseous and particulate peaks in ship emission plumes was typically observed in port regions (Healy et al., 2009;Ault et al., 2010;Merico et al., 2016). These ship emission plumes were also consistent with the prevailing wind directions in plumes, as shown in Figure 2.

Considering these facts, present study defines ship plume periods using $SO_2$ concentrations and vanadium particle number concentrations. For $SO_2$, a minimum threshold of $\Delta_{SO2}$ = $SO_2$(Port) - $SO_2$(Shanghai) > 5 ug/m$^3$ was applied to indicate the arrival of ship plumes. Additionally, the number concentrations of vanadium particles ($PNC_v$) were considered because in some cases the $SO_2$ peaks were absent or obscure while typical fresh vanadium particles were increased. The occurrence probability of this kind of events was low (3% cases). This kind of events were possibly caused by the anchored ships burning low sulfur

content oil (<0.5 % m/m) to comply with regulations in the port region, which has come into force on April 1, 2016; secondly,

it cannot be excluded that vanadium particles be emitted from industry sources, such as petroleum refinery companies in this region. The wind directions during these events support both of the proposed causes. To identify plumes, we excluded the possible industries influences by limiting the prevailing winds only to port directions. Present study set the threshold of vanadium particles in ship plumes to $PNC_v > 25$ particles/hour. Therefore, ship plumes were identified as either $\Delta_{SO2} > 5 \mu g/m^3$

5 or $PNC_v > 25$ particles/hour.

There were about 210 ship emission plumes captured during the entire period. Table 1 summarized the statistics of the $SO_2$, NO, $NO_2$, $O_3$, $PM_{2.5}$ concentrations in the port site and urban area in Shanghai during this study. Vanadium particles number concentrations were represented by particle detecting velocity by SPAMS. The SPAMS particle detecting velocity were positively correlated with particle concentrations in ambient atmosphere, but should not be explained as absolute number

10 concentrations without correction for SPAMS efficiency (Wenzel et al., 2003). Statistics were performed on pollution concentration in plume and non-plumes periods. To separate influences from land-based sources, non-plume periods during winds from port direction were calculated in Table 1.

Generally the concentrations of $SO_2$ and $NO_x$ in the port site is 40~70% higher than Shanghai city (Table 1). The $SO_2$ concentrations in non-plume periods were comparable with that in Shanghai city, irrespective of wind directions, therefore the

15 non-plume $SO_2$ can be recognized as background $SO_2$ in this area. Contrastingly, the $NO_x$ concentrations showed obvious dependence on wind directions in non-plumes, whose concentrations were higher when inland wind prevails, suggesting the importance of land-based emissions to port in coastal areas. In a similar ambient observation at Yangshan port, (Zhao et al., 2013) obtained the average concentration of 29.4 and 63.7 $\mu g/m^3$ for $SO_2$ and $NO_2$ respectively, higher than the present level of 15.6 $\mu g/m^3$ ($SO_2$), 53.2$\mu g/m^3$ ($NO_2$). Noting that the $SO_2$ and $NO_2$ were only intermittently measured for about 20 days in

20 that study (May and August, 10 days each month). Therefore it is not feasible to make direct comparison. In plume period, the $SO_2$ maximum hourly concentration in Yangshan (119.0 $\mu g/m^3$) were close to present study (124 $\mu g/m^3$); Due to land-based emissions, the $NO_2$ maximum hourly concentration in Waigaoqiao port (260 $\mu g/m^3$) is higher than Yangshan port (199.8 $\mu g/m^3$).

In general the ozone concentrations in the port site were lower than Shanghai urban region by 13-33%. To inspect whether the

25 $O_3$ depletion was related to the oxidation of primary NO emissions in the port site, we calculated the $NO_2$ ratios to analyse $NO_x$ composition in plumes. The $NO_2$ ratio is defined as the ratio between $NO_2$ and the $NO_x$ ($NO+NO_2$), which was used by several relevant characterizations of ship emissions (Alföldy et al., 2013;Kurtenbach et al., 2016). Before the calculation of $NO_2$ ratio we firstly converted NO and $NO_2$ mass concentrations to molar unit, and then the background NO and $NO_2$ levels were subtracted to make sure peaks were due to plumes. The distribution of the $NO_2$ ratio in this study was shown in Figure 3,

30 where the $NO_2$ ratio distribution from ship plumes in another study was compared.

The distribution of the $NO_2$ ratios in present study showed several modes. The largest mode occurred at about 0.2 (20%). Obviously this mode was also present in the comparison study (Alföldy et al., 2013), which was recognized as fresh engine emissions from ships. A major difference between two studies is that significant fraction of $NO_2$ ratios occurred in larger range (> 0.4) in present study, which was not observed in Alföldy et al., 2013. The larger $NO_2$ ratios were once thought to be emitted

from unidentified type of ships. When we correlated the $NO_2$ ratio with ambient $O_3$ concentrations, however, we found there was obvious positive correlation between them, as shown in right panel in Figure 3. This result suggests that the higher $NO_2$ ratio of some plumes were not due to the emission characteristics of ships, but due to the transformation of NO to $NO_2$ in the ambient, for if the $NO_2$ ratio were higher at the discharges and no ambient transformation occurred, then there will be no reason to observe the dependence of $NO_2$ ratio on an ambient condition of $O_3$. This is an evidence that the primary NO emission (from ships or on-road traffics ) had contributed to the $O_3$ depletion in this area.

For particulate matter, the $PM_{2.5}$ concentrations in port area were slightly lower than Shanghai city, although $PNC_v$ in plumes were times higher than non-plumes (Table 1). Longer period of $PM_{2.5}$ data suggested the lower $PM_{2.5}$ concentration is a general trend in this port site. This trend is not unique to the port regions because we observed it in other coastal area as well, which is readily observed in $PM_{2.5}$ spatial distribution of Shanghai (Figure S4 in supplementary file). In the spatial distribution there was a general trend of decreasing $PM_{2.5}$ concentrations from inner to coastal areas in Shanghai. This fact is assumed to be caused by the dispersion or advection of clean air from the sea. The portside primary PM from ship emissions are mostly ultrafine particles, with mass emission factors much smaller than $NO_x$ and $SO_2$ (Zhang et al., 2017). Therefore the primary PM from ships or other traffics could not contribute significantly to ambient PM mass concentrations. The vanadium particle number fractions in total particles in SPAMS were obviously larger (6.7 % on average) in portside than urban areas in Shanghai (1-2 %) (Liu et al., 2017)

### 3.2 Particles properties from ship emission

### 3.2.1 Discrimination of fresh and background ship emission particles in port site

With single particle characterization, it is possible to separate fresh or 'pure' ship emission particles from aged types by particle signatures. The mass spectra, wind roses and size distributions of fresh and aged ship emission particles are displayed in Figure 4, 5. The dominant peaks in mass spectra of fresh ship emission particles include sulfate ($-97HSO_4^-$), EC ($C_n^+/C_n^-$, n are integers), and vanadium ($51V^+$, $67VO^+$) peaks. These peaks were reflecting the major composition of fresh ship emission particles found by other techniques (Moldanova et al., 2013;Becagli et al., 2012;Murphy et al., 2009). Fresh ship emission particles produced very low or no nitrate ($-62NO_3$ in negative spectra) signal in mass spectra, as commonly observed in combustion source characterizations (Spencer et al., 2006;Toner et al., 2006). In aged particle type their nitrate signals were stronger than fresh type. Except for the nitrate related peaks ($-46NO_2^-$,$62NO_3^-$), other mass spectral patterns of fresh and aged ship emission particles were similar.

Through the discrimination of ship emission particles into two types, we have identified their different temporal pattern, wind rose and size distribution of ship emission particles in the portside (Figure 4). The temporal variation of fresh ship emission particles showed many peak shaped fluctuations, which was similar to and synchronized with $SO_2$ peaks well (Healy et al., 2009;Ault et al., 2010). However, the number concentrations of aged particles were generally much lower than fresh types (20% of the latter) and have shown more stable temporal concentrations than fresh vanadium particles. We also analysed the

particle number concentrations of ship emission particles at different wind directions. The results were drawn as wind roses for each particle type in Figure 4. The differences between the wind roses of the two types were obvious. It is clear that fresh vanadium particles occurred almost entirely from port directions, and its wind rose ran nearly parallel with the direction of riverbanks (300°-0°-120°). This is a strong evidence that ships were the most predominant source of fresh vanadium particles in the port site. Aged vanadium particles, however, did not shown obvious favoured wind directions and distributed uniformly in all wind directions. Based on described characteristics of vanadium particles, present study assumed that the aged vanadium particles were background particles which have undergone atmospheric processing in local or regional scale. The source of the background vanadium particles may be emitted from other places may not restricted in current port. The size distributions of fresh and background vanadium particles are shown in Figure 5. The fresh vanadium particles dominate particle numbers in smaller size range (< 0.4 μm) where aged or background particles contributed only minor fractions. Although SPAMS detection efficiency declines in this size range, a significant number of ship emission particles were still detected in this size range. The explanation is that this fraction of particles were non-spherical fractal agglomerates, whose cross sections was larger to reflect laser light and thus be detected in SPAMS. The non-spherical fractal shape of fresh vanadium particles was observed as soot particle from fresh combustion sources. Similar observations were reported in other studies using single particle mass spectrometer in ultra-fine size range (Ault et al., 2010).

### 3.2.2 Particle types in fresh ship emission plumes

After separation of background particles, we analysed the mass spectral signatures of fresh ship emission particles using ART-2a algorithm. These particles were grouped into 4 major types based on the similarity of their composition. Temporal variations, composition, size distributions and were analysed to obtain the further information of these types, which will be helpful in particle source identification in other sites. The 4 particle types were labelled as V-OC, V-EC, V-ECFe and V-Ash according to their characteristic composition, whose averaged mass spectra were shown in Figure 6. The negative mass spectra of the four types were similar in that the $SO_4^-$ peak were dominant in addition to other negative EC peaks, which is consistent with the elevated $SO_2$ concentrations in plumes. The major differences of the four particle types were in the positive mass spectra as depicted in Figure 6. Generally the V-OC type were characterized by the dominant organic peaks including $C_2H_3^+$, $C_2H_5^+$, $C_2H_3O^+$, with non or insignificant EC peaks. Generally the organics are ionized in low efficiencies in SPAMS (Ulbrich et al., 2009). The rich organic signals of V-OC particles indicate that they were mainly composed of organics in engine exhaust plumes (Lack et al., 2009;Moldanova et al., 2013). Due to the low ionization efficiency of organics, the particle numbers of V-OC in plumes was generally low compared with other types, which was inappropriately reflecting the dominance of organic compositions in ship emission particles (Lack et al., 2009). The V-EC particles produce dominant EC peaks from $C_1^+$ to $C_{13}^+$ and metal peaks of V and Na, but without iron peaks $Fe^+$. This type is also the most abundant type of all vanadium particles. The V-ECFe type is similar to V-EC type except for the addition of $Fe^+$ and $Ca^+$, $Ni^+$ peaks of lower occurrence frequencies. The V-Ash particles produced minor or no EC peaks except some metal peaks of V, Fe and Ni in positive spectra. These metals

are used as lubricant additives or inherently present in RFO, therefore their presence in ship emission particles are expected and commonly found (Becagli et al., 2012;Moldanova et al., 2013).

Temporal concentrations and size distributions of these particle types are shown in Figure 7. Temporal concentrations of these particle types displayed daily variations, with higher concentrations in daytime than night. The temporal concentrations of these particle types were poorly correlated ($R^2$<0.4), suggesting they were emitted differently. Since these particles were detected in a portside environment, they were assumed to be emitted by ships of different engine types or modes of operation. The V-OC particles, although having low ionization probabilities, were found to concentrate in specific cases of plumes. Since the information of individual ships is not yet available, it is not possible to link V-OC particle plumes to specific ship types. The V-OC particles concentrated in specific ship emission plumes (Figure 7) and its' number concentration peaks were usually narrower (~ 1 hour) than the other particle types (3~5 hour). Sizes of V-OC particles were more uniformly distributed as compared with the other types (Figure 7). Similar organic-rich particles were identified from ship exhaust by other technique (Moldanova et al., 2013).

The V-EC particles dominated the particle numbers in ship plumes in this study. Compared with the other types their sizes enriched in smaller size ranges (< 0.4 μm), which is a typical character of soot particles from the combustion of RFO (Moldanova et al., 2013). The V-Ash particles, which was most probably the ash spheres from combustion process of inorganic constituents in RFO and lubricants, were mainly detected in larger size range (> 0.4 μm) (Figure 7). SPAMS measure particle aerodynamic size which is both determined by particle size and density. The larger densities of metal oxides or salts in V-Ash particles, as compared with soot agglomerates, was also making contributions in their size distribution. The origin of V-ECFe types were probably the result of internal mixing between V-EC and V-Ash particles. Their size distribution was more similar to V-Ash type.

### 3.3 Contributions of ship emission to ambient pollutants in port area

For a coastal port, the evaluation of ship emission to air quality needs to identify impacts from land-based emissions. Obviously these land-based emissions were making greater influences to portside air quality than a marine port far from coast (Zhao et al., 2013). To give an intuitive illustration, the averaged concentrations of $SO_2$, $O_3$, NO, $NO_2$, $PM_{2.5}$ and vanadium particle numbers in different wind directions were summarized in Figure 8. Concentrations of pollutants has demonstrated varied dependency on local wind conditions. It is evident that, for the coastal port site in this study, the $NO_x$ and $PM_{2.5}$ concentrations were highest during inland direction wind prevails. Contrarily, the $SO_2$ concentrations and vanadium particle numbers were dominant only when winds from port sectors. The hotspots in wind rose of vanadium particle were most probably produced by individual docks along the riverside. The wind dependence of ozone concentrations was less apparent, except its' depletion in regions of high $NO_x$ and $SO_2$ levels in wind roses, as previously explained. Obviously the port site was receiving very different pollution impacts from land emission and the ship emissions in port. Present study tries to separate land-based emission influences by limiting wind directions only in port directions. In the calculation of ship emission contributions, two

reference periods were considered in this study: the entire study period (irrespective of wind) and only when the site was in downwind directions of port.

Ship emission contributions of air pollutants in two reference periods are summarized in Table 2. Results show that, if the land-based emissions were considered, ship emission contributed 36.4 % $SO_2$ concentration in local air in port area, a much higher value than for NO (0.7 %), $NO_2$ (5.1 %), and $PM_{2.5}$ (5.9 %). The low contributions of $NO_x$ were due to the inclusion of traffic emissions of stronger intensities from inland directions. The main sources of $NO_x$ from inland directions was considered not far from the site because the average $NO_x$ levels in Shanghai city is lower than the port site, as evidenced in Table 1. For vanadium particle number concentrations ($PNC_v$), ship emissions were the predominant source in present site (49.5 %). The PNCv contribution is a lower estimation considering that SPAMS detect particles less efficiently for smaller particles, where the vanadium particles tend to concentrate. Contributions of $PNC_v$ in different particle size ranges were also calculated in table 2. In either of reference periods (excluding or including land-based emissions), ship emission contributions to $PNC_v$ in smaller size range (0-0.4 μm) are larger compared with $PNC_v$ in larger size ranges (0.4-0.8 μm, 0.8-2.5 μm).

The relative contributions of $PNC_v$ from ship emission is apparently higher than $PM_{2.5}$ on mass concentration. Previous study showed that the direct $PM_{2.5}$ contribution from ship traffics lies within 1-8% range (Contini et al., 2011;Contini et al., 2015). Recent studies carried in Mediterranean region found that ship emission contributed 0.3-7.4% $PM_{2.5}$ concentrations in port areas (Merico et al., 2016).  Ship emission studies in Europe and other regions was reviewed, and its concluded that shipping traffics contributions to $PM_{2.5}$ were in 1-14% range, with higher contributions with decreasing particle size (Viana et al., 2014). The calculated value of $PM_{2.5}$ in the present site is within the reported ranges. Recently (Merico et al., 2017) compared ship traffic atmospheric impacts using inventories, experimental data and modelling approaches in Adriatic-Ionian port areas, and found that ships contributed 0.5-7.4% $PM_{2.5}$ in these areas. The same study further found that ship traffics contribution to particle number concentrations (PNC) is 2-4 time larger than mass concentrations of $PM_{2.5}$. The PNC is not currently measured, instead the size distributions, PNC contributions of vanadium particles in different sizes, as measured by SPAMS, apparently agrees with these previous work.

In a study carried out at Yangshan marine port of Shanghai, the calculated $PM_{2.5}$ contribution (~4 %)  is smaller than present study (5.9 %) (Zhao et al., 2013). In this study a different method was used to evaluate ship emissions, relying on vanadium concentrations to indicate ship emissions. Considering the methodology differences, it is deemed that the results from the two studies are similar within the uncertainty range (Table 2). A previous estimation in Shanghai area using inventories method showed that ship emissions contributed 9 % $NO_x$ and 5.3 % $PM_{2.5}$ in Shanghai area (Zhang et al., 2017), generally agrees with this study in the condition of including land-based emissions (Table 2). However, for $SO_2$ the contribution in that estimation (12 %) is significantly smaller than the 36.4 % in this study. The high $SO_2$ levels in this study is a local character of the port site which is close to emission sources. After transported to the urban region the high $SO_2$ concentrations will dissipate and strength weakened. It is noted that, the synchronized $SO_2$ and vanadium particles plumes as observed in the port site, are observed in a much less frequency in a urban site in Shanghai city where another SPAMS is monitoring. Estimation of ship emission impacts to the urban area will be the subject of future studies.

By limiting the analysis just to period when the winds were from port directions, the influences of land-based emissions could be largely eliminated. As shown in table 2, for all pollutants the ship emission contributions were magnified considerably in amplitude. The most significant variation occurs for gaseous $NO_x$, whose contributions from ship emission increased to levels larger or comparable with $SO_2$. Contributions obtained here can be compared with a similar study carried out in a European port (Merico et al., 2016). Gaseous emissions of NO, $NO_2$ and $SO_2$ were similar between these two studies, which is impressive considering the larger throughput of goods in Shanghai port. However, in an absolute sense, this study estimate that ship emissions contribute to 5.68 μg/m$^3$ $SO_2$, 3.00 μg/m$^3$ $NO_x$ and 1.57 μg/m$^3$ $PM_{2.5}$ during the sampling period. These values are comparable or higher than the reported results in ports in other regions (Viana et al., 2014). For example, a previous study found that the ship emitted particles contributed 0.8 μg/m$^3$ (primary particles) and 1.7 μg/m$^3$ (secondary particles) in Bay of Algeciras (Viana et al., 2009). Due to the adjacency of the site to port, the calculated $PM_{2.5}$ contribution could be largely deemed as primary for present site. The relative contributions of pollutants are partly compensated by the higher background pollution levels in this region.

## 4 Conclusions

In the summer of 2016, we conducted an experimental study to characterize and quantify ship emissions in the Shanghai port. Obvious ship emission plumes were detected in the port site through online measurement of gaseous and particulate matter. During plumes the $SO_2$ and vanadium particles concentrations has demonstrated well synchronized peaks, which could be reliably used to indicate the arrival of ship emission plumes. Statistics of pollutants during plumes showed that the concentrations of $SO_2$ in plumes were about 3 times higher than the background concentrations. Except the plume periods, the $SO_2$ concentrations in port site varies with the background $SO_2$ level in regional scale. $NO_x$ emissions from ships were also obvious during plumes, however, its' concentrations in port site are under much stronger influences from land emissions. For particulate matters, the primary ship emission produce dominant vanadium particle number concentrations ($PNC_v$) to the portside  while its' contribution to the mass concentrations ($PM_{2.5}$) was less significant. Other pollutants $O_3$ was depleted by elevated primary $NO_x$ and $SO_2$ emissions in port regions, resulting 11-33 % ozone consumption compared with urban region of Shanghai.

Particle size distributions and chemical composition of individual ship emission particles were characterized through single particle mass spectrometry at the same site. Similar as $SO_2$, the ship emission particles in portside could also be grouped into freshly emitted and background particle types. The mass spectra of fresh ship emission particles contain dominant peaks of EC, sulfate and trace metals (V, Ni, Fe and Ca). Size distribution of ship emission particles showed that they are tend to concentrate in smaller size range (< 0.4 μm), which was most probably composed of fractal black carbon agglomerates. Based on the different chemical composition of ship emission particles, ship emission particles during plumes could be grouped into four major types: V-OC, V-EC, V-ECFe and V-Ash. These particles types were shown to have different temporal and size distribution trends, which was a manifestation of the complexity of ship emissions in a large, busy port.

The emission contributions from ships to local air quality in Shanghai port area was quantified by extracting pollutions during plume periods from background levels. Ship emissions contributions were evaluated in two scenarios where the land-based emission sources were either included or excluded. Results showed that ship emissions were a major contributor to the ambient $SO_2$ (5.68 µg/m$^3$, 36.4%) and vanadium particle concentrations (49.5%) in portside. $NO_x$ contribution (3.00 µg/m$^3$, 5.8%) from shipping emissions was insignificant compared with emission from land-based sources, which was mainly from transportation sources. If land-based sources were excluded, shipping relative contributions of $NO_x$ became comparable with that of $SO_2$. Due to the high $NO_x$ and $SO_2$ levels in this area, a fraction of local ozone concentrations was found to be depleted. Primary particles from ship emission were estimated to contribute to 5.9% (1.57 µg/m$^3$) $PM_{2.5}$ concentration during the sampling period. For particle number concentration (PNC), over 44% vanadium particle numbers ($PNC_v$) in the port site were found to be contributed by ship emissions. The $PNC_v$ contribution from ship emission were found to increase with decreasing particle size, with 57% vanadium particles smaller than 0.4 µm were found to be emitted from ship emission. Since the size and mass of fresh exhaust particles are small, the primary mass concentrations from ships would be inappropriate to represent their real mass contribution after atmospheric aging. This study supports that particle number concentration (PNC) be included in the characterization of primary emissions from ships.

**Disclaimer**

The content of this paper does not necessarily reflect the views and policies of the HKSAR Government, nor does mention of trade names or commercial products constitute an endorsement or recommendation of their use.

**Author contribution**

Qingyan Fu, Xinning Wang and Yin Shen designed the experiment; Xinning Wang, Yin Shen and Jun Pan conducted the experiment; SPAMS data was analysed by Xinning Wang and Mei Li; Other data is analysed by Xinning Wang, Yan Zhang and Yanfen Lin; Manuscript is prepared by Xinning Wang, Qingyan Fu and Peter K.K. Louie.

**Acknowledgement**

This work was financially supported by the Shanghai Science and Technology Committee (STCSM) projects (Grant No. 15DZ1205402 and 17DZ1203100). Analysis of SPAMS data was funded by National Natural 422 Science Foundation of China (Grant No.21607056). Special thanks are addressed to Shanghai East Container Terminal Co., Ltd. (SECT) for their valuable aids during observations, and Shanghai Environmental Monitoring Technology and Appliance Ltd. for attentive instrument maintenance services in the campaign.

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

**Figures**

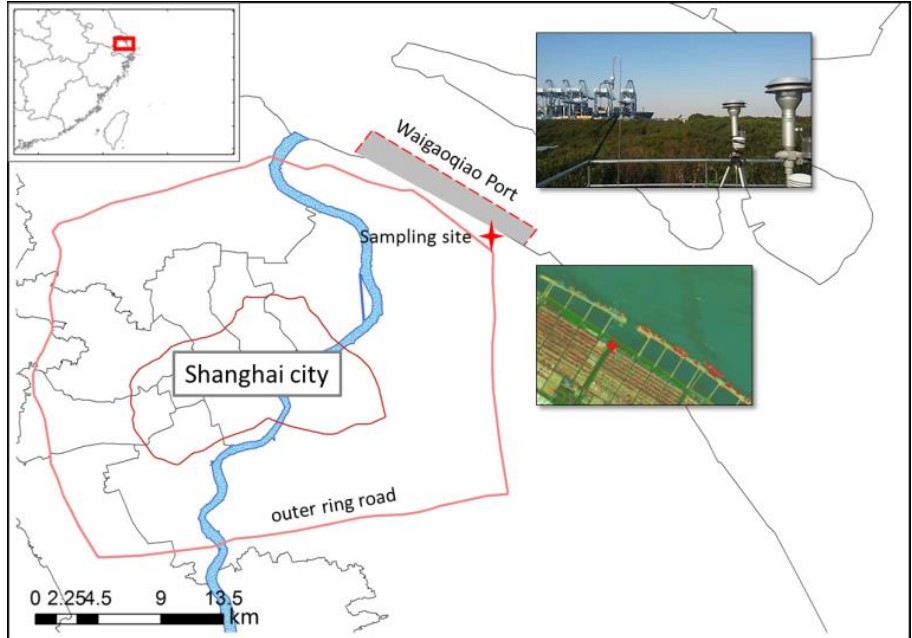

5   **Figure 1: Map of sampling site in Shanghai port and surrounding areas. Port region is indicated by shaded area. The insets are the satellite image of the port site and a photo taken on the roof of monitoring station seeing in port direction.**

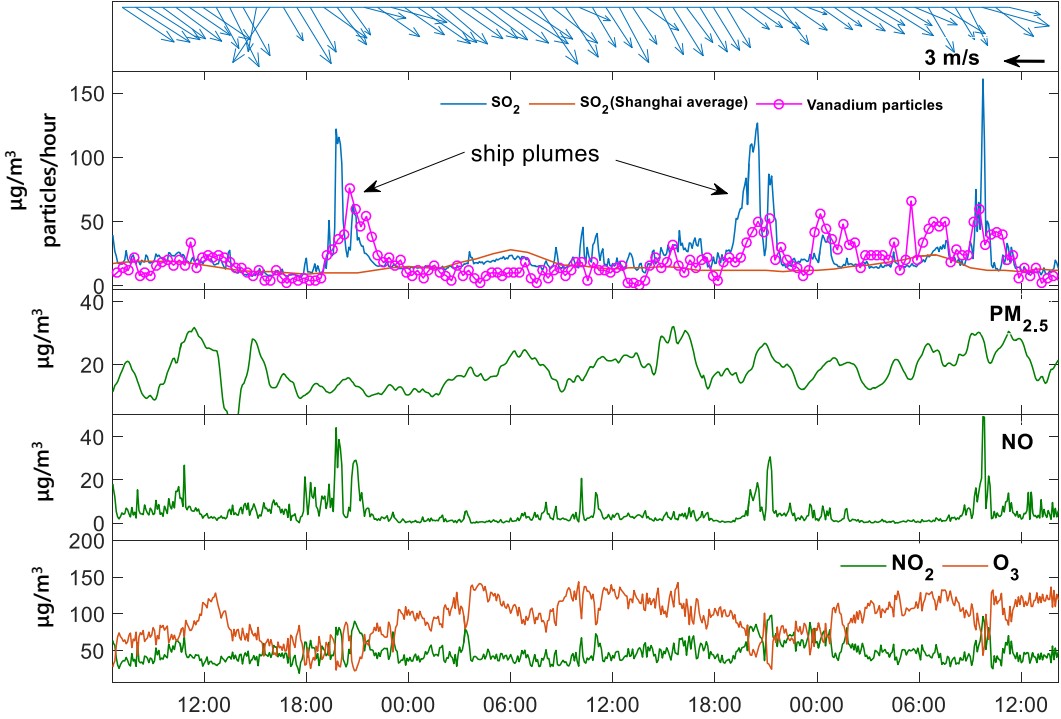

**Figure 2: Temporal concentration of pollutants SO₂, NO, NO₂, O₃ and PM₂.₅ during 27-29 August 2016. Contemporary wind direction and speed, SO₂ concentration of Shanghai city and vanadium particles number concentration as detected by SPAMS are included as a reference.**

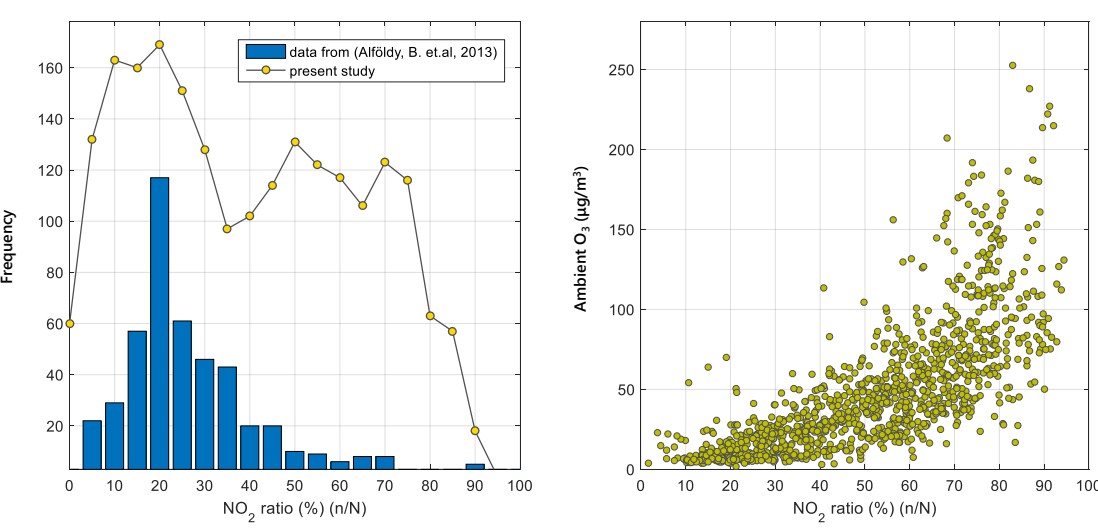

**Figure 3. The NO₂ ratio distribution during plumes in this study and a similar study (left) and the plot of NO₂ ratio against ambient ozone concentrations during plume periods (right).**

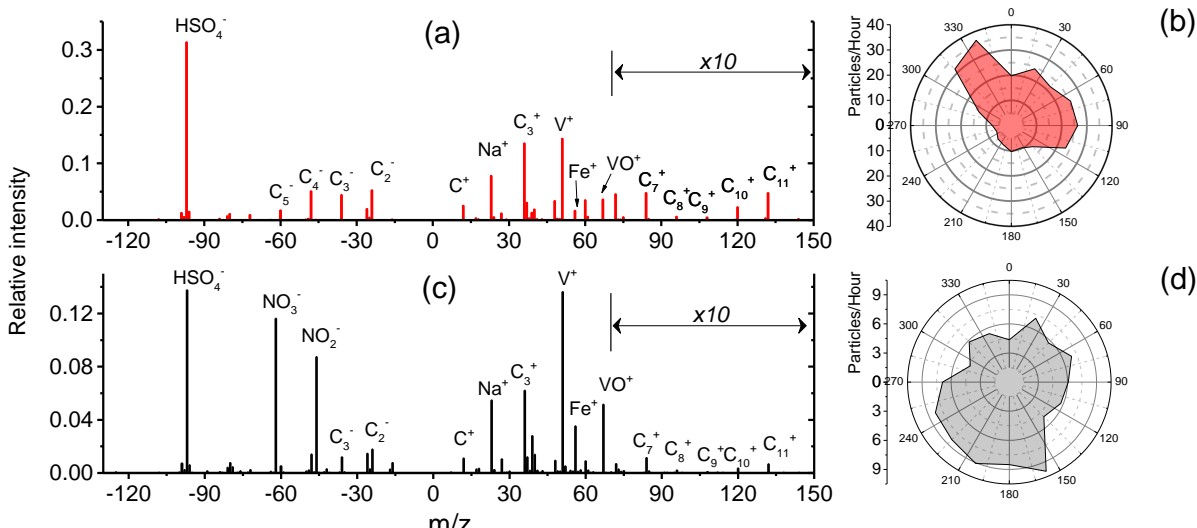

**Figure 4: Mean mass spectra of fresh and background ship emission particles in port (a, c) and the wind rose of particle number concentration (in measure of particle number per hour) of these two particle types (b, d). Peaks in mass range of 70-150 in (a) and (c) are magnified by 10 times.**

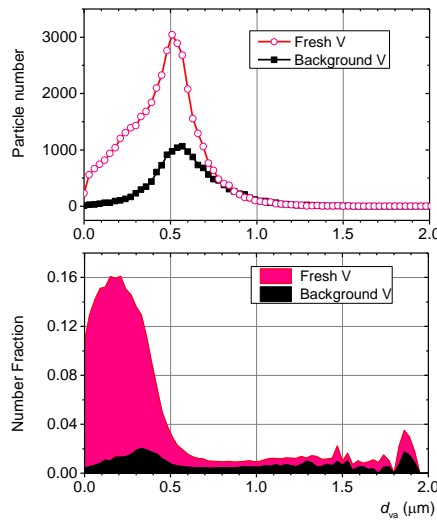

**Figure 5: Particle number size distribution of fresh and background ship emission particles by SPAMS, with the y-axis representing particle numbers detected at each size bin at constant sample flow in the entire study (Upper). Size distribution of the fresh and background ship emission particles normalized by total particles at each size (Lower).**

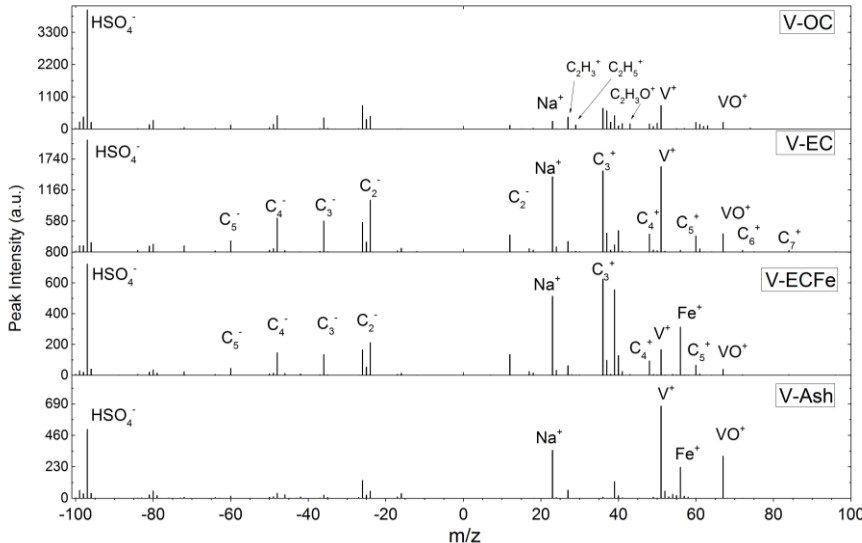

**Figure 6: Mean mass spectra of four major particle types from fresh ship emission.**

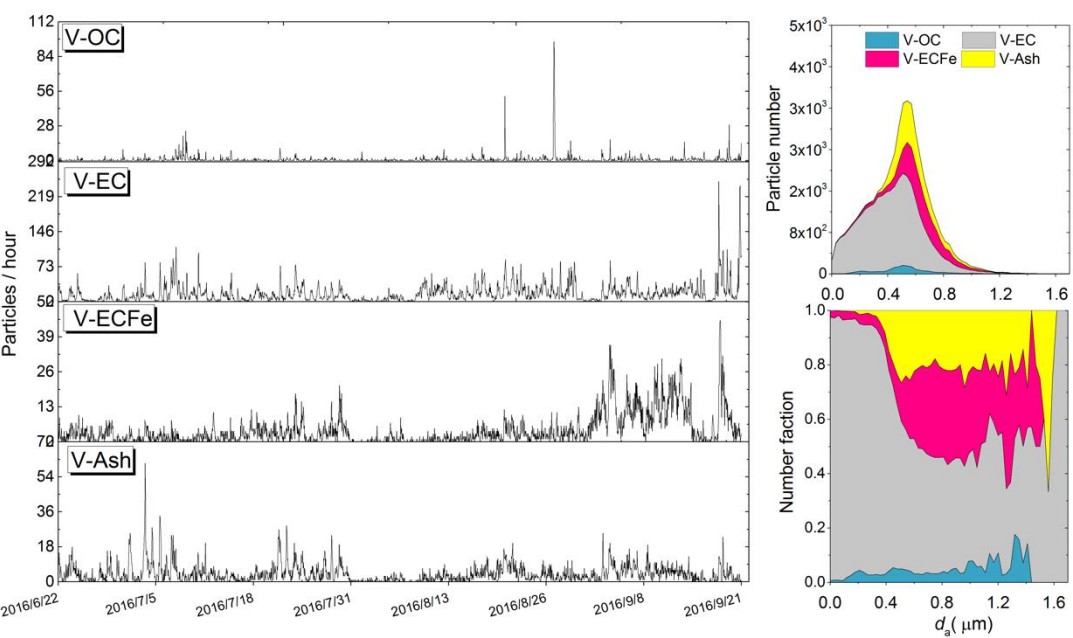

5 **Figure 7: Temporal trends of particle numbers detected per hour by SPAMS of four fresh vanadium particle types (Left panel); The upper right panel is the number-size distribution of the 4 types, with the y-axis representing particle numbers detected at each size bin at constant flow in the entire study. The Lower right panel is obtained by normalizing the particles numbers of 4 types to give their relative contributions at each size.**

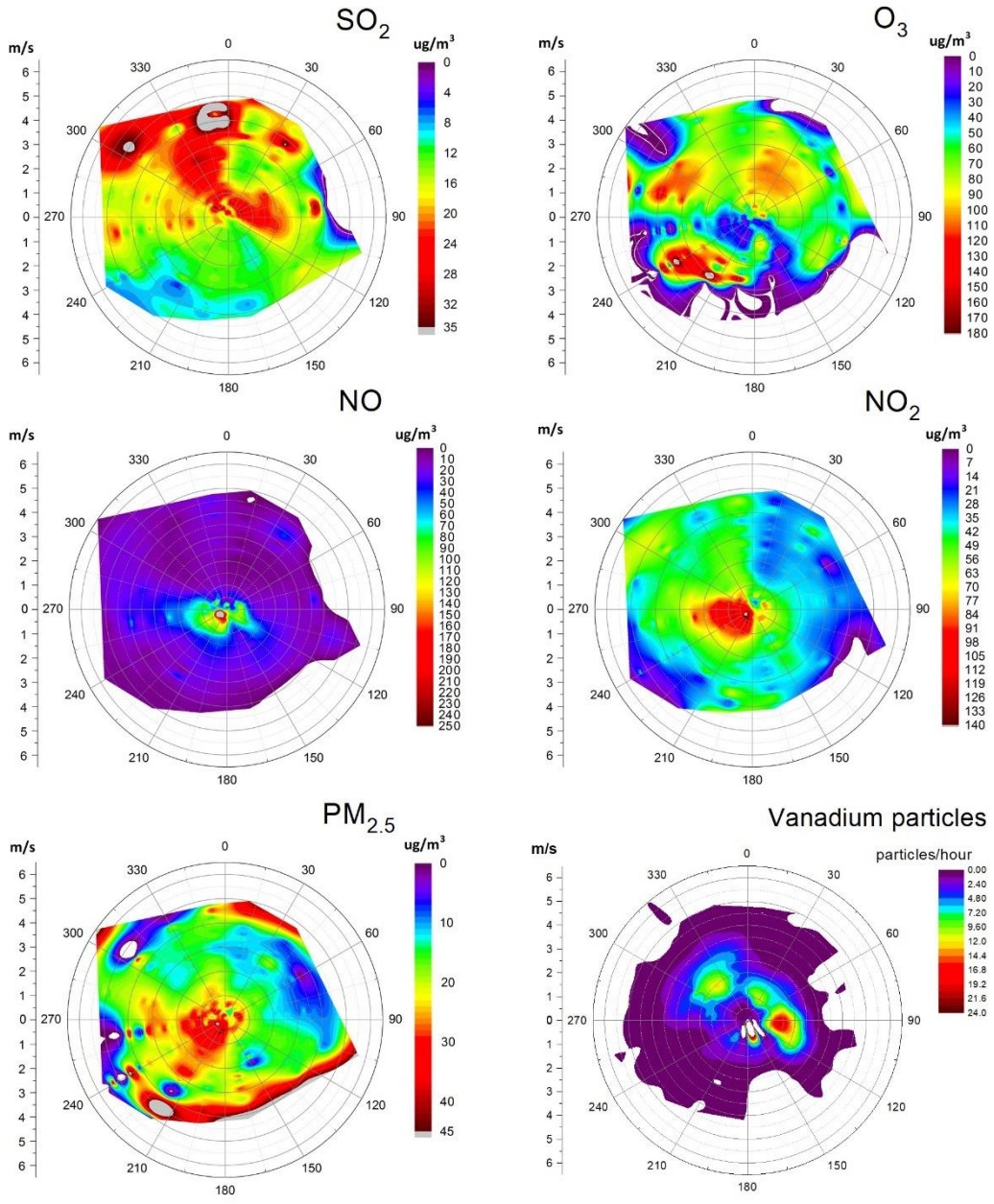

**Figure 8: Pollution roses of SO₂, NO, NO₂, O₃, PM₂.₅ and vanadium particles during the whole study period. Vanadium particles wind rose is based on number concentration as measured by SPAMS.**

**Tables.**

**Table 1: Statistics of pollutants concentration level during the whole sampling period. Numbers are average concentration followed by 25th and 75th quantiles in brackets. Average pollution levels in Shanghai city during the same period are included as a comparison.**

| | In plume | | Non-plume | | Non-plume (port sector) | | Port average | | Shanghai average | |
|---|---|---|---|---|---|---|---|---|---|---|
| $SO_2$ ($\mu g/m^3$) | **28.3** | (17.6~31.8) | **9.9** | (8.1~11.6) | **10.2** | (8.2~12.1) | **15.6** | (8.7~16.8) | **10.8** | (9~12) |
| NO ($\mu g/m^3$) | **42.5** | (7.6~47.5) | **41.6** | (7.1~59.1) | **16.5** | (1.8~18.1) | **41.9** | (7.3~55.3) | **5.8** | (3~6) |
| $NO_2$ ($\mu g/m^3$) | **59.3** | (36.1~72.4) | **50.5** | (27.8~60.8) | **36.9** | (22.1~46.1) | **53.2** | (30.3~65.0) | **30.2** | (18~38) |
| $O_3$ ($\mu g/m^3$) | **53.1** | (19.3~77.8) | **54.6** | (15.4~84.7) | **71.3** | (45.4~97.6) | **54.1** | (16.9~82.7) | **81.1** | (40~107) |
| $PM_{2.5}$ ($\mu g/m^3$) | **30.2** | (14.8~39.6) | **25.1** | (12.8~32.5) | **19.6** | (11.6~23.2) | **26.7** | (13.2~34.1) | **31.4** | (16~43) |
| Vanadium particles (#/hour) | **47.6** | (31~55) | **10.9** | (5~17) | **12.3** | (7~19) | **22.8** | (7~29) | | --- |

**Table 2: Contributions of ship emissions to ambient pollutants SO₂, NO, NO₂, O₃, PM2.5 and vanadium particles in port area. Calculations are based on two situations: entire sampling period (all wind directions included) and only when site is in downwind direction of port emissions. Total lengths (in hours) of respective periods are given.**

| (%) | | In port sector (excluding land-based emissions) | | Entire period (including land-based emissions) | |
|---|---|---|---|---|---|
| | | Average | range | Average | range |
| $SO_2$ | | 57.2 | (49.2, 64.8) | 36.4 | (29.2, 40.2) |
| NO | | 71.9 | (57.0, 84.6) | 0.7 | (0.2, 1.7) |
| $NO_2$ | | 30.4 | (24.7, 34.6) | 5.1 | (3.7, 7.9) |
| $O_3$ | | -16.6 | (-18.8, -13.4) | -0.9 | (-2.8, -0.4) |
| $PM_{2.5}$ | | 27.6 | (22.5, 33.2) | 5.9 | (3.4, 9.6) |
| Vanadium particles* | (0-0.4 μm) | 79.2 | (73.9, 85.0) | 57.1 | (50.6, 64.0) |
| | (0.4-0.8 μm) | 75.3 | (68.1, 82.0) | 44.7 | (38.1, 52.3) |
| | (0.8-2.5 μm) | 76.6 | (70.4, 82.9) | 47.0 | (41.3, 52.9) |
| | (0-2.5 μm) | 77.0 | (70.6, 83.1) | 49.5 | (43.0, 56.7) |

Length of sampling (in hours):  Entire period: 2256; Port sector: 1136; In plume: 694; Non-plume: 1563; Non-plume (port sector): 625.
 * Particle number contribution