# Peer review of "Atmospheric pollution from ships and their impacts on local air quality in a port site in Shanghai"

_Atmospheric Chemistry and Physics, 2018_

## Referee Comment (RC1) · Anonymous Referee #2 · 8 Sep 2018

General comment The paper regards an analysis of the impact of shipping to atmospheric pollutants measured in the area of Shanghai harbour (China). The approach used is based on the identification and characterisation of ship plumes using high temporal resolution measurements of gaseous pollutants and of particles using a SPAMS. The work is interesting and allowed to investigate the typical spectra of particles released by ships as well as to evaluate statistically the contribution of shipping to local air quality. The work is suitable for the Journal and generally well written (even if minor spell check is required), however, some aspects are not completely clear (see my specific comments) and an additional effort in the discussion of size distributions of the impacts should be included. In conclusion, I believe that the paper should be considered for publication after a major revision.

Specific comments

Title. I think that it is not correct to speak of "measurements of shipping emissions" because emission factors or measurements of specific emission rates are not given. I would suggest to change the title to put in evidence the core of the work: contribution of shipping to atmospheric pollution.

Introduction. The adoption of a DECA (Domestic ECA) is quite interesting and it would be even more interesting if a more detailed discussion is included. For example, it would possible to comment on the efficacy of this measure in reducing the impact of shipping on local pollution levels. It is also worth to mention that a recent work (Contini et al., 2015 - Atmospheric Environment 102, 183-190) showed that application of "domestic" restrictions on the fuel quality could be effective in reducing not only local $SO_2$ concentrations but also primary emissions of particles from ships. I believe that a discussion on this aspect would be appreciated by the readers.

Sections 2.3 and 2.4. It is often mentioned the high temporal resolution of SPAMS measurements, I would suggest to explicitly report the numerical value.

Page 6 (lines 1-5). V-particles measured without the presence of $SO_2$ peaks are interpreted as due to the use of low-sulphur content fuel, however, it would not be possible that they are coming from other industrial (or anthropic in general) sources? Some words on this should be included.

Page 7, line 23. To speak at this level of BC is not really useful, likely authors mean EC.

Page 8, lines 2-3. This sentence is not clear and should be re-written. I believe that authors means that ultrafine particle concentrations could be a better metric compared to mass concentrations to investigate the impact of shipping to atmospheric aerosol.

Page 9, line 30. The approach based on this formula was originally developed in Contini et al (Journal of Environmental Management 92 (2011) 2119e2129) and suc-

cessively used by other authors. I believe that it would be fair to mention this aspect.

Looking at the size distributions reported in figures 4 and 6, it appears that V particles are especially relevant for ultrafine particles, however this aspect is not deeply investigated on the evaluation of the impacts. It would be possible to use the approach discussed on page 9 to investigate the size dependency of the impacts of shipping, eventually estimating the impacts for different size ranges. I believe that, if a sufficient statistics could be obtained, this will give very useful additional information compared to the impact on total particle number reported in Table 2.

Page 10, lines 13-23. The comparison with shipping impact measured in other ports is certainly interesting, however, it is done on relative impacts and not on absolute contributions due to shipping activities this means that it depends not only on ship traffic but also on the contributions of the other sources acting on the specific measurement site. This should be mentioned because it could explain some of the apparent discrepancy mentioned by the authors. In addition, I would suggest to expand the comparison to other ports analysed with the high temporal resolution approach (Merico et al Transportation Research Part D 50 (2017) 431–445) but also with other complementary approaches (see for example Viana et al 2014 Atmos. Environ. 90, 96–105).

Regarding the impacts reported in Table 2, it would be possible to estimate the uncertainties?

Page 11 line 5. This sentence is not clear. Authors likely mean that the impact of shipping is more relevant and clearly discernible on SO2 and V particles compared to the other pollutant analysed. Could authors clarify?

In the supplementary material it is reported "…in present study the online single particle measurement was utilized to indicate the occurrence of shipping emission plumes…" however in the main text was mentioned that both particles and SO2 concentrations were used. Please clarify this apparent contradiction.

[Figure]

Minor corrections

Page 1, line 15. Better "particle size distributions".

Page 1, line 28. Please eliminate the initial S.

Page 2, line 19. Subscript for SO2. The same in page 4 (line 25).

Page 7, line 14. Better "different size distributions. . ."

Page 8, line 12. Better "by the dominant". In addition, I would remove etc, if necessary please mention explicitly.

Page 8, line please remove etc. as above.

Page 8, line 25. Better "is therefore not attempted. . ."

Page 9, line 2. $> 0.5 \ \mu$m

Page 11, line 10 ozone without capital letter.

---

## Referee Comment (RC2) · Anonymous Referee #1 · 11 Sep 2018

See the attached comments.

Please also note the supplement to this comment:
https://www.atmos-chem-phys-discuss.net/acp-2018-737/acp-2018-737-RC2-supplement.pdf
* * *
[Figure]
Review on the manuscript "Ambient measurement of shipping emissions in Shanghai port areas" (ACP-2018-737). Authors: Wang et al.

**General comments**

This study conducted field measurements from June to September in 2016 at Shanghai port in order to understand the impact of ship emissions on the air quality in portside. Trace gases, PM2.5 and vanadium particle number concentrations were continuously monitored at the site. Ship plumes were clearly captured by the instruments. SO2 and vanadium particle number concentrations correlated well with ship plumes. Four types of ship plumes were identified based on the mass spectra of Single Particle AMS. The contributions of ship emissions to different air pollutants in the atmosphere and in the air masses from port directions were quantified. Given that Shanghai port is the largest port in the world, this study will add values to existing literature of ship emission studies. However, the manuscript is not well organized/written and has room to be improved. In addition, there are quite a lot of grammar errors and technical mistakes, which sometimes make the reviewer confused. Furthermore, some discussions and conclusions are lack of evidence. As such, this manuscript can be considered for publication after the following specific comments are well addressed.

**Specific comments**

**Abstract:**

Firstly, English needs editing by a native English speaking professional or company. For example, line 16: … that shipping emissions is a major….".
Secondly, there are also some technical mistakes. One example, lines 14-15: Gaseous (NO, NO2, SO2, O3) and particulate concentrations (PM2.5)… It should be "The concentrations of gaseous pollutants (NO…..) and fine particulate matters (PM2.5)…". Also, both shipping emission and ship emission are used throughout the manuscript which should be consistent. Another problem at lines 18-20, the subject is "Single particle mass spectra of fresh shipping emission" but the last words became "…and nitrate peaks in aged particles". This is really confusing the reviewer.
Thirdly, the abstract should provide specific and detailed findings rather than common senses. The only specific finding described in the abstract is probably the last sentence. The others are all about common knowledge which is also applied to any other ports. What is the uniqueness of the study port?

**Introduction:**

As there are too many grammar errors, I have made some comments and revisions on the manuscript. I will submit my comments with the manuscript.

**Experimental:**

1) It is not clear whether the sampling site is downwind location of the port or not, or whether the ship plumes could really arrive at the sampling site or not. The authors should provide more

**Fig. 1.** Review comments

**Supplement:**

**Ambient measurement of shipping emissions in Shanghai port areas**

Xinning Wang1, Yin Shen1, Yanfen Lin1, Jun Pan1, Yan Zhang2, Peter K.K. Louie3, Mei Li4, Qingyan Fu1

5 1 Shanghai Environmental Monitoring Center, Shanghai 200030, P.R. China.

2 Department of Environmental Science and Engineering, Fudan University, Shanghai 200433, P.R. China.

4 Institute of Mass Spectrometer and Atmospheric Environment, Jinan University, Guangzhou, 510632, China.

Correspondence to: Qingyan Fu (qingyanf@semc.gov.cn)

10

Abgreact. Growing shipping activities in port areas have generated negative impacts on climate, air quality and human health. To better evaluate the environmental impact of shipping emissions, in the summer of 2016 ambient air quality measurement was carried out at Shanghai port, one of the busiest port in the world. Gaseous (NO, NO2, SO2, O3) and

- (15) particulate concentrations (PM2.5), particle sizes and chemical composition of individual shipping emission particles were continuously monitored for 3 months. High temporal resolution data show that shipping emissions is a major culprit of local air pollution problem. Shipping emission plumes were clearly observed using online measurement in port area. The SO2 and Vanadium particles numbers were found to correlate best with shipping emissions in Shanghai port. Single particle mass spectra of fresh shipping emission were identified based on the dominant peaks of Sulfate, EC and indicative metals of V, Ni,
- Fe and Ca, and nitrate peaks in aged particles. Fresh particles from ship emission mainly concentrated in ultra-fine size range where their number contributions are more apparent than their mass. Advanced measurement conducted in the present study show that in port region shipping emissions contributed 36.4 % SO2, 0.7 % NO, 5.1 % NO2, -0.9 % O3, 5.9 % PM2.5, 49.5 % Vanadium particles if land-based emissions were included, and 57.2 % SO2, 71.9 % NO, 30.4 % NO2, -16.6 % O3, 27.6 % PM2.5, 77.0 % Vanadium particles if land-based emissions were excluded.
- 25 Keywords

Shipping emission; Shanghai port; emission source contribution; SPAMS

**1** Introduction**

S Ship emission constitutes an important source of gaseous and particulate pollution world wide. Growing shipping activities in recent years are attracting much attention to assessing its impact on environment and health (Fuglestvedt et al., 2009). For

30 emissions from sea-going vessels in pristine marine environment, it is found that ship emission affect clouds properties along

<sup>3 Hong Kong Environmental Protection Department, Hong Kong, P.R. China.

cruising route (Petzold et al., 2008;Coggon et al., 2012), which is directly relevant to earth radiation budget and climate issues. In portside or coastal regions, ship emissions could have a negative impact on the air quality at variant degrees in these areas or eities (Donateo et al., 2014;Liu et al., 2017). growing contributions of air pollution from ship emissions, 
[revised manuscript text omitted]
 whole sampling period the Shanghai city SO2 concentration matched very well with ambient SO2 concentration in port area during non-plumes periods. This is also consistent with the observation data that pollution plumes arising from shipping emission superimpose on background and regional SO2.

As typical combustion products, the NO and NO2 concentrations also show corresponding elevated concentrations during

- 20 plumes under favorable wind fields (Fig. 2). However, during the whole study NO and NO2 are more importantly influenced by land traffics emissions (mostly from transportation diesel trucks) when the inland wind prevails. The shipping emission NOx plumes reached sampling site have been slightly aged. To understand the aging effect, when wind direction is in port sector ( $300^{\circ}-0^{\circ}-120^{\circ}$ ), the averaged NO/NO2 ratio is 0.6 (mostly fall in 0.1~4.5 range), lower than typical ratio of 4 at ship exhaust (Alföldy et al., 2013), suggesting the oxidation of primary NO into NO2 have occurred for some time (1~50 min,
- based on wind speed measurement and transportation distance). This result is evidenced by the apparent consumption of  $O_3$ in plumes as shown in Figure 2, commonly termed as titration effect between NOx and O3. The PM2.5 mass concentration did not show as apparent response as that of SO2 during ship emission plumes, despite that the typical Vanadium particles had reached the site, as shown. The obscure response of PM2.5 to ship plumes is explained by the fine particle sizes in relatively fresh shipping emissions, as discussed in following section.
- 30 Taking into consideration of all the relevant factors, the present study defines ship plume periods by ways of SO2 concentrations and Vanadium particle number concentrations. For SO2, a minimum threshold of  $\Delta_{SO2}$  = SO2(Port)-SO2(Shanghai) > 5 ug/m3 is applied to indicate ship plumes. For shipping emission particles, the number concentration of

[revised manuscript text omitted]

The size distribution agrees with the sizes of ship emission particles measured in other studies (Gonzalez et al., 2011;Merico et al., 2016). It is reported that for ship exhaust their particle number contributions are more considerable than their mass contributions (Donateo et al., 2014;Jonsson et al., 2011;Merico et al., 2016). Size distribution of fresh particles from ship

exhaust shows that the number concentration mainly concentrated in UF mode (<100 nm) (Gonzalez et al., 2011;Moldanova et al., 2013). The less significant increase in PM2.5 mass concentration than NOx and SO2 in plumes are due to, in one respect, that the emission factors of PM are typically much lower than NOx and SO2 (Agrawal et al., 2010;Moldanova et al., 2013), and second, that the smaller size of fresh vanadium particles before secondary accumulation happens in the atmosphere

7

(Moldanova et al., 2013). Considering that the secondary processing occurred on every particle, the particle number concentrations of fresh shipping emissions is more significance than their initial mass concentrations. The study suggests that particle number quantification is more appropriate in an accurate evaluation of primary ship emissions.

**3.2.2 Particle types in fresh ship emission plumes**

- 5 After the separation of background ship emission particles, fresh Vanadium particles are isolated to represent fresh ship emission particles. Further analysis is performed to study their composition and emission characters, which will be helpful in particle source identification of SPAMS in future studies. In general, the fresh Vanadium particles could be grouped into 4 types based on their chemical composition: V-OC, V-EC, V-ECFe and V-Ash, and the average mass spectra are shown in Figure 5. The negative mass spectra of the four types are common in the dominating SO4- peak in addition to other negative
- 10 EC peaks in spectra, consistent with the high SO2 concentration in ship emission-plumes. The major chemical differences of the four particle types are in the positive mass spectra as depicted in Figure 5. The V-OC type are characterized by the dominate OC peaks  $C_2H_3^+, C_2H_5^+, C_2H_3O^+$ , etc., with non or insignificant EC peaks. Considering the low ionization probability of OC in laser ionization, this particle type is deemed to be mainly composed of condensed organics in engine exhaust plume (Lack et al., 2009;Moldanova et al., 2013). The V-EC particles produce dominant EC peaks from  $C_1^+$  to  $C_{13}^+$
- 15 and metal peaks of V and Na, but without iron peaks Fe+. This type is also the most abundant type of all Vanadium particles. The V-ECFe type is similar as V-EC except for the addition of Fe+ and Ca+, Ni+ peaks of lower frequencies. The V-Ash particles produce minor or no EC peaks except some metal peaks of V, Fe and Ni in positive spectra. These metals are used as lubricant additives or inherently present in RFO, therefore their presence in ship emission particles are expected and commonly found (Becagli et al., 2012;Moldanova et al., 2013).
- 20 The chemical composition and size distribution of these four types suggest they have distinct physical properties (Fig. 6). Temporal trends of these 4 types show that their concentrations display daily fluctuations, with higher concentrations in daytime than nighttime. However, the inter-correlations among their concentrations are low ( $r^2

[revised manuscript text omitted]

---

## Referee Comment (RC3) · Anonymous Referee #1 · 11 Sep 2018

Review on the manuscript "Ambient measurement of shipping emissions in Shanghai port areas" (ACP-2018-737). Authors: Wang et al.

**General comments**

This study conducted field measurements from June to September in 2016 at Shanghai port in order to understand the impact of ship emissions on the air quality in portside. Trace gases, PM2.5 and vanadium particle number concentrations were continuously monitored at the site. Ship plumes were clearly captured by the instruments. SO2 and vanadium particle number concentrations correlated well with ship plumes. Four types of ship plumes were identified based on the mass spectra of Single Particle AMS. The contributions of ship emissions to different air pollutants in the atmosphere and in the air masses from port directions were quantified. Given that Shanghai port is the largest port in the world, this study will add values to existing literature of ship emission studies. However, the manuscript is not well organized/written and has room to be improved. In addition, there are quite a lot of grammar errors and technical mistakes, which sometimes make the reviewer confused. Furthermore, some discussions and conclusions are lack of evidence. As such, this manuscript can be considered for publication after the following specific comments are well addressed.

**Specific comments**

**Abstract:**

Firstly, English needs editing by a native English speaking professional or company. For example, line 16: … that shipping emissions is a major….".

Secondly, there are also some technical mistakes. One example, lines 14-15: Gaseous (NO, NO2, SO2, O3) and particulate concentrations (PM2.5)… It should be "The concentrations of gaseous pollutants (NO…..) and fine particulate matters (PM2.5)…". Also, both shipping emission and ship emission are used throughout the manuscript which should be consistent. Another problem at lines 18-20, the subject is "Single particle mass spectra of fresh shipping emission" but the last words became "…and nitrate peaks in aged particles". This is really confusing the reviewer.

Thirdly, the abstract should provide specific and detailed findings rather than common senses. The only specific finding described in the abstract is probably the last sentence. The others are all about common knowledge which is also applied to any other ports. What is the uniqueness of the study port?

**Introduction:**

As there are too many grammar errors, I have made some comments and revisions on the manuscript. I will submit my comments with the manuscript.

**Experimental:**

1) It is not clear whether the sampling site is downwind location of the port or not, or whether the ship plumes could really arrive at the sampling site or not. The authors should provide more

detailed description of the site. What were the prevailing winds during the sampling period and how to ensure the capture of ship plumes? There is also no information about the station. Is it a container or mobile vehicle? What is the height of the station if the outlet of the sampling tube was 3.5 m above the ground?

2) Was the CO measured? Though it was claimed that calibration and maintenance of the instruments were regularly performed, brief QA/QC procedures and detection limits are still requested. Nothing was mentioned about the QA/QC of PM2.5 monitoring.

3) It is not clear how the components in particles such as vanadium were identified and quantified by the SPAMS. Detailed information is needed.

4) Data analysis: in the results, pollution and wind roses were presented while nothing is described about the method to draw pollution and wind roses, and how to explain the pollution and wind roses. In addition, the method of calculation of shipping contributions is improperly placed in the "Results" section, which should be described in the "Experimental".

Page 5, line 3: what is ART-2a algorithm? This method was mentioned to be used to the searched clusters to generate sub-clusters of particles. However, no information at all about this method was provided.

**Results and discussions**

Page 5, line 10: which typically persist for a few hours: can you tell us the specific hours in your study rather than vague value like this?

Page 5, lines 21-25: The discussion here is questionable. By looking at Figure 2, whenever ship plumes were captured, both NO and SO2/vanadium levels were high and correlated well. On what basis, the authors claimed the NOx in plumes reaching the site was aged? Using the NO/NO2 ratio in the plumes? Compared to the ratio measured in other countries and probably different type of ships? This is not convincing. Besides, NO2 is also emitted from shipping as a primary pollutant.

Page 5, line 26-29: given that shipping emission is a major source of PM2.5, it is odd that no PM2.5 peaks were found during the ship plumes in Figure 2. The reason provided by the authors is quite confused. Is it because ship emits sub-micron particles or because the malfunction of the PM2.5 monitor?

Page 5, lines 31-32: what is the basis for the definition of ship plumes using the minimum threshold of delta SO2? In particular, the authors later claimed that in some cases the SO2 peaks were absent?

Page 6, lines 2-3: the reason for absent SO2 is contradictory to Figure 2. If the ships complied with the new regulations, why would you still see SO2 peaks in ship plumes? This kind of discussion is misleading.

Page 6, line 16: "This result suggests that shipping activities are the main source of SO2 plumes in port". Please comment on the NOx emission from ships - is it not important, because of impacts of land-based traffics? But later in Page 10, lines 15-18, you claimed NO was higher than SO2 in fresh ship plumes.

Page 6, line 18: it does not make sense to compare a site near sources with sites in a city without any detailed characteristics of the locations. It would be more meaningful to compare the ship emissions in this study with other similar studies conducted in Shanghai. In fact, there are a number of ship emission studies in this city.

Page 6, lines 21-24: the explanation of low PM2.5 levels at the port site is not convincing at all. Why would other pollutants from shipping emissions be higher if this was caused by clean air?

Page 6, line 27: the title reads awkward. Should it be "Particles in the background air and the ship plumes"?

Page 6, lines 28-31: These should be in "Experimental" section.

Pages 6-7: the first paragraph of section 3.2.1 is messy and lack of logic. It should be re-organized.

Page 7, line 13: The wind rose distribution….. in figure 3. This should be merged with the description of mass spectra for Figure 3 earlier. When it is first mentioned, you should describe all once.

Page 7, lines 16-17: "Background vanadium particles, … in all directions". This is not true in Fig 3(d).
Page 7, lines 20-21: There is no actual comparison at all. No idea about the particle size in background air, larger or smaller?
Page 7, line 22-23: "significant fine ship emission particles were still detected in fine size range". The terms have been randomly used everywhere. What do you mean fine ship emission particles? very non-professional description. It should be fine particles from ship emissions.
Is this contradictory to your previous claim that PM2.5 in ship plumes is lower than that in urban air? In other word, the PM2.5 in background air could be more significant and be detected more significantly?
Page 7, lines 24-27: These two sentences are repeated. the 2nd sentence contains grammar errors.
Page7, lines 28-30: these two sentences are not related. Lack of logic.
Page 7, line 30: "size distribution of fresh particles…". how do you define "fresh particles from ship exhaust" and "particles from ship emissions"?

Page 7, line 31:" …that the number concentration mainly concentrated in UF mode (<100 nm)". very poor English writing. It is non-professional at all.

Page 7, lines 32-33: No idea at all why PM2.5, NOx and SO2 were suddenly discussed here. This section is about SPAMS measurement data. Also, where showed that PM2.5 had less

increase than NOx and SO2? Moreover, are you sure EF of PM are typically much lower than NOx and SO2 in all types of ships with different fuel?

Line 34: Do not understand if there is any connection with less significant increase in PM2.5 mass concentration.

Page 8, lines 1-3: Terrible explanation. The authors lack basic knowledge.

Page 8, line 9: "The negative mass spectra" how? To me it should be negative m/z value for HSO4-. Mass spectra should not be negative. Please make it clearer. In addition, the whole sentence is confusing. "…other negative EC peaks…". Specifically what are they in the spectra?

Page 8, line 20: Firstly, this description is unclear. How could "chemical composition" suggest distinct physical properties"? Secondly, is Fig 6 about this? But it is clear that Fig 6 is about temporal pattern of particle number concentration.

Page 8, lines 26-27: Did you scan it using TEM? If not, how do you know yours is the same as other studies?

Page 8, lines 27-29: Again, you do not have evidence to say this - is the reason that the size distribution of V-OC is different from other types related to incomplete combustion?

Page 8, line 31: "…. suggesting they are principally emitted in specific phases of engine operations." Any references support this?

Page 8, line 33: "UF size…" I guess it means ultra-fine. If correct, what is the size range of ultra-fine particles?

Page 9, line 1: "….product of combustion of RFO (Moldanova et al., 2013)." But your measurement was conducted after the implementation of sulfur reduction regulations. This means the fuel used at berth is not RFO but clean fuel.

Line 2: are mainly detected in larger size range ( > 0.5 um) (Fig. 6). The peak particle size for V-EC, V-ECFe and V-Ash looks similar in Fig. 6. How would you say this?

Lines 3-5: This fits other types of particles as well. I don't understand why you said this here. This is basic knowledge.

Lines 5-6: "The origin of V-ECFe types are probably the result of internal mixing between V-EC and V-Ash particles". Any possible reasons for this speculation?

Line 7: how do you know or do you believe it?

Lines 9-18: The whole paragraph is nothing to do with your results but information about other studies. This should be in "Introduction" section.

Lines 19-20: Very confused statement and unclear purpose.

Lines 22-23: "… clearly showed that they are under the overwhelming influences of land emissions on the sampling site." Why? How do we read the pollution roses in Fig 7? How do we know they were affected by other sources?

Lines 26-27: "Because the air pollution in this two conditions are so different,……" Which two conditions?

Page 9, lines 30-32 and page 10, lines 1-2: all these should be in "Experimental" section.

Page 10, lines 8-12: Quite confused discussion. Re-written.

Page 10, lines 15-17: Here you finally evidenced the major contribution of ship emissions to NOx. But if you went to section 3.1, you claimed that NOx were mainly from land traffics while ship contribution is not important. The two explanations are contradictory.

Page 10, lines 19-20: Do you compare the absolute concentrations of these pollutants attributed to ship emissions in these two studies? This kind of comparison using percentage is very dangerous! I bet the total concentrations of these pollutants in these two studies are totally different.

Lines 24-25: Did you do t-test to say this? ~4% vs. 5.9% is similar to me. Importantly, compare absolute concentrations. The comparison based on percentage does not make sense and can mislead readers.

**Conclusions**

The conclusion section is poorly written. It must be re-organized and re-written.

---

## Author Comment (AC2) · 15 Oct 2018

The referee's comments in RC2 are incomplete and the existing part is identical with the referee's comment RC3. The response to the proposed comments in RC2 will be responded in RC3. The grammatic or other language problems, as raised in the attached file of RC2 by the referee, will be considered for correction in the next version of manuscript.

---

## Referee Comment (RC4) · Anonymous Referee #1 · 16 Oct 2018

After going through the responses, I realized that most of my comments were not addressed and/or understood by the authors. In addition, the authors presented an unfriendly attitude to the reviewer with arrogance and ignorance. Given the fact above, I have to reject this manuscript for potential publication in ACP. Specifically, I have some comments related to the responses of the authors as follows. 1) Experimental section, response to my first question. The authors were ignorant to ask the reviewer to read other published references about sampling design. Firstly, the authors were unable to understand the reviewer's questions of sampling design. Secondly, the fact that others successfully did the ship emission measurements at the ports does not mean your study is convincing if the sampling design is unclear. You should address the comment rather than rudely ask the reviewer to read other references. In fact,

those published papers provided clear description of the sampling. 2) Experimental section, response to my third question about the operation principle of SPAMS. Though the authors may be familiar with this equipment, it does not mean that others know it very well. How would this be "unnecessary to be included in the text"? 3) Results and discussion section. My question to Page 5, lines 21-25. The reviewer is asking the authors to clarify why NOx in the plumes reaching the site was aged given that the fresh signals of high NO, SO2 and vanadium were captured. Unfortunately, the authors were unable to understand my questions. They are more arrogant and ignorant to tell the reviewer about the reaction of NO+O3. The authors made a black humor to me given my research interest in O3 pollution for many years. 4) Their response to my question on Page 5, lines 26-29. Again, the authors did not understand my comment and rudely tell me that my comment is ridiculous. However, the Figure R2 provided by the authors clearly indicated that PM2.5, SO2 and SPAMS could concurrently have peaks on some other days. The authors gave themselves a slap loudly. Why didn't you see an obvious PM2.5 peak sometimes (in your Figure 2)? 5) Their response to my question on Page 5, lines 31-32. Again, the authors do not have the ability to understand the question. Their answer is not related to my question. my question was that why would you use $\Delta SO2 > 0.5$ ug/m3 not other values as a minimum threshold? 6) Their response to my question on Page 6, lines 2-3. That is exactly my question. Do you understand my question? 7) Their response to my question on Page 6, line 16. They do not understand the comment. 8) Their response to my question on Page 6, line 18. Firstly, the authors' English is too poor to understand the question. Secondly, I can only say that they are ignorant and arrogant. 9) Their response to my question on Page 6, lines 21-24. The authors are joking. Though the reviewer pointed out their problem and the authors admitted it, they still insisted this was their postulated explanation. 10) Their response to my question on Pages 6-7. Before you said this, can you ask other professionals to check with you if you are so overconfident? 11) Their response to my question of "is this contradictory to your previous claim that PM2.5 in ship plumes is lower than that in urban air?" Can you tell me how "fine particles are

only a fraction of PM2.5"? Isn't PM2.5 called "fine particle"? 12) Their response to my question on Page 8, lines 1-3. It is ridiculous that on one hand, the authors attacked the reviewer; on the other hand, they revised their explanation. This group is not doing real research. 13) Their response to my question on Page 8, line 9. How were the authors badly educated by saying so unreasonable words? At least I am confused. Can you guarantee others won't be confused? 14) Their response to my question on Page 8, lines 27-29. Their response is just robber's logic. If you do not have evidence, why do you still say this? 15) Their response to my question on Page 9, lines 3-5. Thank you for noticing that. Do you have basic ability to understand English and write a good scientific paper? 16) Their response to my question on Page 10, lines 15-17. So, can you roughly tell us the fraction of land traffic-emitted NOx and ship-emitted NOx, before you say NOx were mainly from land traffics? In fact, the reviewer hoped that the authors could clarify these. Overall, the reviewer spent much time and great efforts to help the authors improve their manuscript. Unfortunately they do not acknowledge the effort. Instead, they are arrogant and ignorant to reject or ignore the comments/suggestions of the reviewer. They are not willing to friendly discuss the scientific issues. They are wasting my time. More importantly, most of my comments were not well understood and addressed by the authors. The authors show strong resistance to significantly improve the manuscript. As such, I suggest to reject this manuscript.

---

## Author Comment (AC3) · 16 Oct 2018

**Response to Referee's Comments on acpd-2018-737:**
**"Ambient measurement of shipping emissions in Shanghai port areas"**

The many items of comments raised by the referee #1 are responded individually as following. Text is present in the format of:

Referee's comments – Black;

The Authors' responses – Blue.

**Response to Referee #1**

This study conducted field measurements from June to September in 2016 at Shanghai port in order to understand the impact of ship emissions on the air quality in portside. Trace gases, PM2.5 and vanadium particle number concentrations were continuously monitored at the site. Ship plumes were clearly captured by the instruments. SO2 and vanadium particle number concentrations correlated well with ship plumes. Four types of ship plumes were identified based on the mass spectra of Single Particle AMS. The contributions of ship emissions to different air pollutants in the atmosphere and in the air masses from port directions were quantified. Given that Shanghai port is the largest port in the world, this study will add values to existing literature of ship emission studies. However, the manuscript is not well organized/written and has room to be improved. In addition, there are quite a lot of grammar errors and technical mistakes, which sometimes make the reviewer confused. Furthermore, some discussions and conclusions are lack of evidence. As such, this manuscript can be considered for publication after the following specific comments are well addressed.

**Specific Comments:**

Abstract:

Firstly, English needs editing by a native English speaking professional or company. For example, line 16: ... that shipping emissions is a major....".

Response:

The language problems, as the author responded in reply to RC2, will be corrected in the next edition of manuscript. The author acknowledge the referee's effort.

Secondly, there are also some technical mistakes. One example, lines 14-15: Gaseous (NO, NO2, SO2, O3) and particulate concentrations (PM2.5)... It should be "The concentrations of gaseous pollutants (NO.....) and fine particulate matters (PM2.5)...". Also, both shipping emission and ship emission are used throughout the manuscript which should be consistent. Another problem at lines 18-20, the subject is "Single particle mass spectra of fresh shipping emission" but the last words became "...and nitrate peaks in aged particles". This is really confusing the reviewer.

Response:

1. Original sentence revised as: "The concentrations of gaseous pollutants (NO, $NO_2$, $SO_2$, $O_3$) and fine particulate matters ($PM_{2.5}$), size distribution and chemical composition of ship emission particles were continuously monitored for 3 months".

2. Both of the "shipping emission" and "ship emission" are seen in literatures. This manuscript unify them to "ship emission".

3. The sentence will be "Single particle mass spectra of fresh shipping emission were identified based on the dominant peaks of Sulfate, EC and indicative metals of V, Ni, Fe and Ca".

Thirdly, the abstract should provide specific and detailed findings rather than common senses. The only specific finding described in the abstract is probably the last sentence. The others are all about common knowledge which is also applied to any other ports. What is the uniqueness of the study port?

Response:

Abstract section will be revised concentrating on the key findings and the uniqueness of present study. It will include aspects on fresh ship emission particle signatures, gaseous pollutants characters from ship emission and their contributions, size resolved ship emission particle contributions to portside and the importance of separation of land emissions.

Introduction:

As there are too many grammar errors, I have made some comments and revisions on the manuscript. I will submit my comments with the manuscript.

Response:

All the grammatical errors will be corrected in next edition.

Experimental:

It is not clear whether the sampling site is downwind location of the port or not, or whether the ship plumes could really arrive at the sampling site or not. The authors should provide more detailed description of the site. What were the prevailing winds during the sampling period and how to ensure the capture of ship plumes? There is also no information about the station. Is it a container or mobile vehicle? What is the height of the station if the outlet of the sampling tube was 3.5 m above the ground?

Response:

The referee seems to be suspicious about the possibility to detect ship emissions in port site. This question was answered by the many published studies in portside across the world (Healy et al., 2009;Ault et al., 2010;Contini et al., 2015;Merico et al., 2016).

For an online, continuous monitoring the prevailing wind notion is not useful because a prevailing wind direction does not imply wind will stay in that direction. They could shift to any directions during a long period of time. The monitoring site is certainly in the downwind direction of port when the wind is in 300°-0°-120° sector, which was clearly indicated in Fig. 1 in manuscript.

During the sampling period the prevailing wind is from southeast which is typical for summer season, as presented in Figure R1. Wind in other directions including port directions also occurred although with less frequency.

The station is on the south riverbank, as already stated in manuscript and illustrated clearly in Fig. 1 in original manuscript. The author could not understand how it is thought the station is on a container or mobile vehicle.

[Figure]

Figure R1. Portside wind rose during the study period.

Was the CO measured? Though it was claimed that calibration and maintenance of the instruments were regularly performed, brief QA/QC procedures and detection limits are still requested. Nothing was mentioned about the QA/QC of PM2.5 monitoring.

Response:

During the period the CO analyzer is not functional, so that the discussion on CO is not included in this study. The QA/QC procedures of NO-NO$_2$-NO$_x$, O$_3$, SO$_2$ and particulate PM$_{2.5}$ is under guidance of << Technical specifications for operation and quality control of ambient air quality continuous monitoring system for SO$_2$, NO$_2$, O$_3$ and CO >> and <<Technical specifications for operation and quality control of ambient air quality continuous monitoring system for Particulate matter (PM$_{10}$/PM$_{2.5}$)>> in China. QA/QC procedures in the national air quality monitoring stations are the same. Procedures and detection limits will be added. The section 2.2 will be revised as:

"The concentrations of gaseous NO-NO$_2$-NO$_x$, SO$_2$, and O$_3$ were measured continuously from Jun-21 to Sep-21, 2016. The gaseous pollutants were monitored by a suit of Thermal Scientific analyzers (NO-NO$_2$-NO$_x$, model 42i; SO$_2$, model 43i; O$_3$, 49i) installed in the monitoring station. Calibration and maintenance of instruments were regularly performed according to the requirement of relevant national standards in China. The regular practices were zero checks (through a zero air generator) and span checks (through standard NO$_2$ and SO$_2$ gas of known concentrations; the O$_3$ standard was generated through a calibration photometer system); The PM$_{2.5}$ concentrations were monitored by oscillating microbalance method (Thermo TEOM 1405-F). Calibration of TEOM is not relied on standard, for the aerosol mass on a filter was monitored by the oscillation frequency change of the tapered element during specified time. The regular maintenance of TEOM includes the changing of filters before the filter loading approach 100%. The flow rate of TEOM was regularly checked using a flowmeter. The lower detection limits of these

instruments are: 0.4 μg/m$^3$ (NO, NO$_2$); 0.5 μg/m$^3$ (SO$_2$); 0.5 μg/m$^3$ (O$_3$); 1 μg/m$^3$ (PM$_{2.5}$). Weather conditions (temperature, humidity, pressure, wind speed and direction) were monitored by a mini-weather station installed on the rooftop of the station. The monitor was about 1 m above the roof of the station and 3.5 m above the ground. Data from all the instruments and the monitor was managed in a customized database and set to 5 min resolution. Atmospheric pollutants concentrations in Shanghai city area, including gaseous pollutants and PM$_{2.5}$ concentrations, were monitored concurrently at 9 national air quality monitoring stations in 1h resolution. The averaged pollutants concentrations at these stations during the sampling period were included for comparison. ".

It is not clear how the components in particles such as vanadium were identified and quantified by the SPAMS. Detailed information is needed.

Response:

SPAMS identify particle composition, such as vanadium, in mass spectrometry method. In the ionization laser beam in SPAMS, the components in particles are ionized into ions carrying charge, then they are separated in the Time-of-Flight tube by atomic mass of the ion. Lighter ions, such as H$^+$, transit fastest in TOF tube and produce peaks in shortest time. The 208Pb$^+$ is heavier so that it reach the MS detector with longer time. This will result a spectrum sorted by ion atomic mass. Components of different atomic mass produce peaks in different position in mass spectrum. Vanadium in particles normally produce peaks at mass = 51V$^+$ and 67 (VO$^+$) and thus can be identified. This is the basics of MS and unnecessary to be included.

Data analysis: in the results, pollution and wind roses were presented while nothing is described about the method to draw pollution and wind roses, and how to explain the pollution and wind roses. In addition, the method of calculation of shipping contributions is improperly placed in the "Results" section, which should be described in the "Experimental".

Response:

Pollution roses are normal and frequently presented in literatures. The author has used the most normal method to draw a wind rose: firstly perform statistics (can either be minimum, maximum, counts, mean value, or any statistics) on relevant pollutant concentrations (e.g., number concentrations of vanadium particles) in every wind directions and then do the plot. If the wind speed data is also included, then another frame of information appears and a map is obtained , which will produce Fig. 7 in original manuscript.

The explanation of wind roses is equally straightforward, in that it could demonstrate, in a intuitive manner, the direction of emission sources relevant to the observation site. The author will explaine wind roses within text in appropriate places in next edition.

The method to quantify the contribution from ship emission will be moved to 'Experimental' section.

Page 5, line 3: what is ART-2a algorithm? This method was mentioned to be used to the searched clusters to generate sub-clusters of particles. However, no information at all about this method was provided.

Response:

The ART-2a is a classification algorithm conventionally adopted by SPAMS community to group similar particle based on the particle similarities. A reference on ART-2a algorithm will be added here in case that the reader want further information on the algorithm.

Results and discussions

Page 5, line 10: which typically persist for a few hours: can you tell us the specific hours in your study rather than vague value like this?

Response:

The specific value of hours is variant, most of them fall in range of 3-6 hours.

Page 5, lines 21-25: The discussion here is questionable. By looking at Figure 2, whenever ship plumes were captured, both NO and SO2/vanadium levels were high and correlated well. On what basis, the authors claimed the NOx in plumes reaching the site was aged? Using the NO/NO2 ratio in the plumes? Compared to the ratio measured in other countries and probably different type of ships? This is not convincing. Besides, NO2 is also emitted from shipping as a primary pollutant.

Response:

We have not tested NO/NO2 ratio in the exhaust because the observation was carried out in a station on land. The author also know that NO2, together with NO, is released as primary emissions. However their ratio, upon their emission into atmosphere, will subject to change quickly through the oxidation of NO into NO2 in the existence of ozone (O3), which is abundant in summer time. This was evidenced by the quickly reduced O3 level during plumes in Fig. 2 in manuscript. The referee seems to be doubtful about this reaction, which is the very basis that the NO-NO2-NOx analyzer is working on.

Page 5, line 26-29: given that shipping emission is a major source of PM2.5, it is odd that no PM2.5 peaks were found during the ship plumes in Figure 2. The reason provided by the authors is quite confused. Is it because ship emits sub-micron particles or because the malfunction of the PM2.5 monitor?

Response:

The authors have stated that $PM_{2.5}$ peaks was not as apparent as that of $SO_2$ and $NO_x$ in plumes, and have not states that no $PM_{2.5}$ peaks were found. It should be made clear that only a short period of data was shown in Fig.2 only with the purpose of illustrating temporal variation in plumes. In the last question the referee is suspecting that $PM_{2.5}$ analyzer was in malfunction. The author would like to illustrate another longer period of data in Figure R2 (shown below) and let the referee make his judgement. The sharp peaks of $SO_2$ in Figure R2 could help to locate the plumes. If the $PM_{2.5}$ instrument is in malfunction, how did the $PM_{2.5}$ instrument happened to malfunction only in plumes? In another aspect, the SPAMS particle number concentration have shown good correlation with the $PM_{2.5}$ measurement. How did the two instruments both malfunction? It sounds ridiculous.

[Figure]

Figure R2. Temporal variations of $PM_{2.5}$, $SO_2$ concentrations and SPAMS analyzed particle number concentrations during Jul-20 to Jul-30 in this study.

Page 5, lines 31-32: what is the basis for the definition of ship plumes using the minimum threshold of delta SO2? In particular, the authors later claimed that in some cases the SO2 peaks were absent?

Response:

Very clear, frequent sharp peaks of $SO_2$ over the background $SO_2$ concentration is suggesting that they are emitted from adjacent combustion sources. Concurrent increases of vanadium particles suggest they are combusting Residual oil. Wind directions during plumes mainly in port directions suggest they are from ships. The cases the $SO_2$ peaks were absent were rare( 3% cases), and will be explained in next edition of manuscript.

Page 6, lines 2-3: the reason for absent SO2 is contradictory to Figure 2. If the ships complied with the new regulations, why would you still see SO2 peaks in ship plumes? This kind of discussion is misleading.

Response:

As stated above, the cases the $SO_2$ peaks were absent were rare and will not affect significantly the result of the study. The new regulation only confine the Sulfur content in fuel, not eliminate the Sulfur from the fuel.

Page 6, line 16: "This result suggests that shipping activities are the main source of SO2 plumes in port". Please comment on the NOx emission from ships - is it not important, because of impacts of land-based traffics? But later in Page 10, lines 15-18, you claimed NO was higher than SO2 in fresh ship plumes.

Response:

The author think that NOx emission from the land-based traffic to the site is important considering all surrounding sources. The claim in Page 10, lines 15-18 was under the condition that land-based emission were excluded by considering the period when site was only under influence of port direction.

Page 6, line 18: it does not make sense to compare a site near sources with sites in a city without any detailed characteristics of the locations. It would be more meaningful to compare the ship

emissions in this study with other similar studies conducted in Shanghai. In fact, there are a number of ship emission studies in this city.

Response:

The comment is not acceptable. We made comparison between air pollution level in a port site with that at the urban area of the same city, How did it make non sense? The referee had better suggest a study which he think make more sense.

Page 6, lines 21-24: the explanation of low PM2.5 levels at the port site is not convincing at all. Why would other pollutants from shipping emissions be higher if this was caused by clean air?

Response:

The slight lower $PM_{2.5}$ concentration at the port site is a fact. The author only postulated possible explanations.

Page 6, line 27: the title reads awkward. Should it be "Particles in the background air and the ship plumes"?

Response:

The title is revised as: " Discrimination of fresh and background ship emission particles in port site".

Page 6, lines 28-31: These should be in "Experimental" section.

Response:

Will be revised.

Pages 6-7: the first paragraph of section 3.2.1 is messy and lack of logic. It should be re-organized.

Response:

Not acceptable. The referee think it is messy and lack of logic if he is not familiar.

Page 7, line 13: The wind rose distribution….. in figure 3. This should be merged with the description of mass spectra for Figure 3 earlier. When it is first mentioned, you should describe all once.

Response: Is it a really publicly accepted rule in scientific paper?

The text will be considered for some reorganization.

Page 7, lines 16-17: "Background vanadium particles, … in all directions". This is not true in Fig 3(d).

Response:

Original text has stated "prominent" and "nearly".

Page 7, lines 20-21: There is no actual comparison at all. No idea about the particle size in background air, larger or smaller?

Response:

Revised as: " The size distributions of fresh vanadium particles were dominated by fine particles ( < 0.5 um in diameter), as shown in Fig. 4. ".

Page 7, line 22-23: "significant fine ship emission particles were still detected in fine size range". The terms have been randomly used everywhere. What do you mean fine ship emission particles? very non-professional description. It should be fine particles from ship emissions.

Response:

Will be uniformed to 'fine particles'.

Is this contradictory to your previous claim that PM2.5 in ship plumes is lower than that in urban air? In other word, the PM2.5 in background air could be more significant and be detected more significantly?

Response:

It should be noted that particles are measured by SPAMS in number concentrations, while $PM_{2.5}$ is

measuring particle mass. The fine particles are only a fraction of $PM_{2.5}$ and not always correlated with $PM_{2.5}$. No contradictory identified.

Page 7, lines 24-27: These two sentences are repeated. the 2nd sentence contains grammar errors.

Response:

Revised as: "The non-spherical, fractal shape of fresh vanadium particles is consistent with the typical shapes from fresh combustion sources (Ault et al., 2010). ".

Page7, lines 28-30: these two sentences are not related. Lack of logic.

Response:

Revised as: "It was normally found that ship emission particles were enriched in ultrafine size range( <100 nm) as measured by other techniques, resulted an elevated particle number concentrations (PNC) more considerable than mass concentrations (Donateo et al., 2014;Jonsson et al., 2011;Merico et al., 2016). ".

Page 7, line 30: "size distribution of fresh particles…". how do you define "fresh particles from ship exhaust" and "particles from ship emissions"?

Response:

Uniformed to "ship emission articles".

Page 7, line 31:" …that the number concentration mainly concentrated in UF mode (<100 nm)". very poor English writing. It is non-professional at all.

Response:

Revised as: " The fine particle modes could be as small as 30 nm in fresh ship emission plume (Gonzalez et al., 2011;Moldanova et al., 2013).".

Page 7, lines 32-33: No idea at all why PM2.5, NOx and SO2 were suddenly discussed here. This section is about SPAMS measurement data. Also, where showed that PM2.5 had less increase than NOx and SO2? Moreover, are you sure EF of PM are typically much lower than NOx and SO2 in all types of ships with different fuel?

Response:

The less apparent peaks of $PM_{2.5}$ during plume period has been noted in section 3.1.

The sentence revised as: " With the size distribution of ship emission particles, the less apparent peaks of $PM_{2.5}$ mass than $NO_x$ and $SO_2$ during plume periods, as shown in section 3.1, is related with the small size of ship emission particles. In another aspect, the reported PM emission factors of ships are typically smaller than $NO_x$ and $SO_2$ (Agrawal et al., 2010;Moldanova et al., 2013), which would also contribute.";

Line 34: Do not understand if there is any connection with less significant increase in PM2.5 mass concentration.

Response:

Seen revisions proposed in last comment.

Page 8, lines 1-3: Terrible explanation. The authors lack basic knowledge.

Response:

The referee did not point out the what is the basic knowledge and how they are terrible. In discussion of aerosol mass and number concentrations, it is easy to understand that particle number multiplied the average mass of individual particles produce the total mass concentration. Consider the supposed two cases: An aerosol sample of low numbers of particles but with larger size, and an aerosol sample of larger particle numbers but small average size. The two case may generate the same mass concentration. However, the difference between the two cases is that

the latter produce larger surface area which favors the secondary accumulation (secondary formation of nitrate, sulphate, organics) to particle surface. This is why the number concentration matters. Hope the explanation is less terrible to the referee.

The text will be revised as "Since the size and mass of fresh exhaust particles are small, the mass concentration PM from exhaust pipes would be inappropriate to represent their real mass contribution after atmospheric aging. This study suggests, as other authors did, that particle number concentration (PNC) be adopted to fully characterize primary ship emitted particles.".

Page 8, line 9: "The negative mass spectra" how? To me it should be negative m/z value for HSO4-. Mass spectra should not be negative. Please make it clearer. In addition, the whole sentence is confusing. "…other negative EC peaks…". Specifically what are they in the spectra?

Response:

The positive and negative mass spectra a common nomenclature in SPAMS literatures, and will not generate confusions. The EC peaks is already noted in 3.2.1 section when it appear the first time.

Page 8, line 20: Firstly, this description is unclear. How could "chemical composition" suggest distinct physical properties"? Secondly, is Fig 6 about this? But it is clear that Fig 6 is about temporal pattern of particle number concentration.

Response:

The sentence is revised as "The four ship emission particle type detected in this study displayed distinct distributions (Fig. 6).".

Page 8, lines 26-27: Did you scan it using TEM? If not, how do you know yours is the same as other studies?

Response:

The several sentences will be revised as: "In previous study an organic particle type was identified by TEM images of ship emission particles (Moldanova et al., 2013). This organic particle contain vanadium impurities, which agree with organic and vanadium signatures in mass spectra in Figure 5. During the incomplete combustion (e.g., starting up phase) the organic vapours in fuels will condense onto particles (ash, EC particles) in the cooler ambient environment, resulting uniform size distribution compared with other types (lower right panel in Figure 6). The V-OC type are more transient in that the peak width of its concentration peaks are normally narrower than other types (~1 hours contrasting to 3~5 hours of other types).";

Page 8, lines 27-29: Again, you do not have evidence to say this - is the reason that the size distribution of V-OC is different from other types related to incomplete combustion?

Response:

This is an inference to explain the observed result. No existing study is found by author.

Page 8, line 31: "…. suggesting they are principally emitted in specific phases of engine operations." Any references support this?

Response:

This item was omitted, as shown in previous comment.

Page 8, line 33: "UF size…" I guess it means ultra-fine. If correct, what is the size range of ultra-fine particles?

Response:

It's size include < 400 nm (aerodynamic diameter) in this study. Next edition they will be uniformed

to "fine size range".

Page 9, line 1: "….product of combustion of RFO (Moldanova et al., 2013)." But your measurement was conducted after the implementation of sulfur reduction regulations. This means the fuel used at berth is not RFO but clean fuel.

Response:

Clean fuel with S <0.5% will also produce Soot particles.

Line 2: are mainly detected in larger size range ( > 0.5 um) (Fig. 6). The peak particle size for V-EC, V-ECFe and V-Ash looks similar in Fig. 6. How would you say this?

Response:

The size distribution shape of V-EC, V-ECFe and V-Ash types in upper-right panel is presented in absolute numbers. In smaller size range (< 0.5 μm) their number concentration all declined due to the decreased detection efficiencies of SPAMS in this size range. However, their relative contributions change as a function of size as shown in bottom-right panel.

Lines 3-5: This fits other types of particles as well. I don't understand why you said this here. This is basic knowledge.

Response:

The referee likes to speak "basic knowledge". Is it really your basic knowledge if I have not present it?

Lines 5-6: "The origin of V-ECFe types are probably the result of internal mixing between V-EC and V-Ash particles". Any possible reasons for this speculation?

Response:

This statement and the latter one has been removed.

Line 7: how do you know or do you believe it?

Response:

See previous statement.

Lines 9-18: The whole paragraph is nothing to do with your results but information about other studies. This should be in "Introduction" section.

Response:

Will be moved to "introduction".

Lines 19-20: Very confused statement and unclear purpose.

Response:

Revised as: " Due to the close proximity to urban region of Shanghai, the present coastal site is under stronger influences from land emissions, contrasted with the marine port far from coastal area (Zhao et al., 2013).".

Lines 22-23: "… clearly showed that they are under the overwhelming influences of land emissions on the sampling site." Why? How do we read the pollution roses in Fig 7? How do we know they were affected by other sources?

Response:

The information in wind roses in figure 7 is rather clear that the port site PM2.5, NOx are strongly influenced by land-based emissions, while the SO2, and ship emitted vanadium particles are under the major impact of ship emissions in port. The ozone wind rose indicates apparent depletions where NOx and SO2 are concentrated.

Lines 26-27: "Because the air pollution in this two conditions are so different,……" Which two conditions?

Response:

It is referring to periods when the site is influenced by land-based and port emissions.

Page 9, lines 30-32 and page 10, lines 1-2: all these should be in "Experimental" section.

Response:

Will be moved to "Experimental" section.

Page 10, lines 8-12: Quite confused discussion. Re-written.

Response:

Revised as "From the calculation method, it could be inferred that the wind frequencies at each directions will influence the calculated results. Therefore, The estimation of impacts from ship emission will produce biased estimation on the prevailing wind directions (that is, the sea direction). During the summer time when southeast winds are prevalent. Adjacent coastal regions other than port area may experience more ship emissions so that their contribution will be larger."

Page 10, lines 15-17: Here you finally evidenced the major contribution of ship emissions to NOx. But if you went to section 3.1, you claimed that NOx were mainly from land traffics while ship contribution is not important. The two explanations are contradictory.

Response:

The referee have not understood the point. The mentioned situation is after the land-based emission have been ruled out. It is nothing surprising to found that, if the emission in Shanghai is removed, the activities in port region will certainly be the dominant source of both $NO_x$ and $SO_2$.

Page 10, lines 19-20: Do you compare the absolute concentrations of these pollutants attributed to ship emissions in these two studies? This kind of comparison using percentage is very dangerous! I bet the total concentrations of these pollutants in these two studies are totally different.

Response:

The absolute contributions will be presented and discussed in next edition, together with suggestions raised by referee #2.

Lines 24-25: Did you do t-test to say this? ~4% vs. 5.9% is similar to me. Importantly, compare absolute concentrations. The comparison based on percentage does not make sense and can mislead readers.

Response:

No t-test. The absolute concentrations will be present.

Conclusions

The conclusion section is poorly written. It must be re-organized and re-written.

Response:

Conclusion will be revised.

Ault, A. P., Gaston, C. J., Wang, Y., Dominguez, G., Thiemens, M. H., and Prather, K. A.: Characterization of the Single Particle Mixing State of Individual Ship Plume Events Measured at the Port of Los Angeles, Environmental Science & Technology, 44, 1954-1961, 10.1021/es902985h, 2010.

Contini, D., Gambaro, A., Donateo, A., Cescon, P., Cesari, D., Merico, E., Belosi, F., and Citron, M.: Inter-annual trend of the primary contribution of ship emissions to PM2.5 concentrations in Venice (Italy): Efficiency of emissions mitigation strategies, Atmospheric Environment, 102, 183-190, 10.1016/j.atmosenv.2014.11.065, 2015.

Healy, R. M., O'Connor, I. P., Hellebust, S., Allanic, A., Sodeau, J. R., and Wenger, J. C.:

Characterisation of single particles from in-port ship emissions, Atmospheric Environment, 43, 6408-6414, 10.1016/j.atmosenv.2009.07.039, 2009.

Merico, E., Donateo, A., Gambaro, A., Cesari, D., Gregoris, E., Barbaro, E., Dinoi, A., Giovanelli, G., Masieri, S., and Contini, D.: Influence of in-port ships emissions to gaseous atmospheric pollutants and to particulate matter of different sizes in a Mediterranean harbour in Italy, Atmospheric Environment, 139, 1-10, 10.1016/j.atmosenv.2016.05.024, 2016.

Moldanova, J., Fridell, E., Winnes, H., Holmin-Fridell, S., Boman, J., Jedynska, A., Tishkova, V., Demirdjian, B., Joulie, S., Bladt, H., Ivleva, N. P., and Niessner, R.: Physical and chemical characterisation of PM emissions from two ships operating in European Emission Control Areas, Atmospheric Measurement Techniques, 6, 3577-3596, 10.5194/amt-6-3577-2013, 2013.

Zhao, M., Zhang, Y., Ma, W., Fu, Q., Yang, X., Li, C., Zhou, B., Yu, Q., and Chen, L.: Characteristics and ship traffic source identification of air pollutants in China's largest port, Atmospheric Environment, 64, 277-286, 10.1016/j.atmosenv.2012.10.007, 2013.

---

## Author Comment (AC4) · 13 Nov 2018

Now that the reviewer #1 has demonstrated hostile attitude toward this paper, the author may tell his real feeling about this reviewer's "comments". The reviewer's language is rude and offensive ("Come on!", "terrible explanations", "the author lack the basic knowledge!"); his questions are nonsense and ridiculous ( see the "comments" in RC3), which the author had answered in patience. The author thinks that this reviewer is not commenting, he is bothering. This ill intention will be rejected altogether. The author refuse to respond to any of the questions from reviewer #1, except the ones responded in AC3. This is the final and last response .

[Figure]

2018.
Interactive
comment

---

## Author Response (AR1)

**Response to Referee's Comments on acpd-2018-737:**
**"Ambient measurement of shipping emissions in Shanghai port areas"**

The authors are pleased to submit our responses to the comments raised by the anonymous referee #2. The authors appreciate and are thankful to the referee's valuable comments and suggestions which help greatly to improve the quality of the manuscript. Each item of the raised comments is responded individually in following pages in the format of:

Referee's comments – Black;

The Authors' responses – Blue,

Author's changes in manuscript – *Red, italic*

**Response to Referee #2**

General comment: The paper regards an analysis of the impact of shipping to atmospheric pollutants measured in the area of Shanghai harbour (China). The approach used is based on the identification and characterization of ship plumes using high temporal resolution measurements of gaseous pollutants and of particles using a SPAMS. The work is interesting and allowed to investigate the typical spectra of particles released by ships as well as to evaluate statistically the contribution of shipping to local air quality. The work is suitable for the Journal and generally well written (even if minor spell check is required), however, some aspects are not completely clear (see my specific comments) and an additional effort in the discussion of size distributions of the impacts should be included. In conclusion, I believe that the paper should be considered for publication after a major revision.

**Specific Comments:**

1. Title. I think that it is not correct to speak of "measurements of shipping emissions" because emission factors or measurements of specific emission rates are not given. I would suggest to change the title to put in evidence the core of the work: contribution of shipping to atmospheric pollution.

Authors' Response:

Thanks for comment. The authors agree with the referee's opinion about the title of the paper. To reflect the core of this work more accurately, the title of this manuscript would be changed to "Atmospheric pollution from shipping and their contributions to air quality degradation in a port site in Shanghai".

*changes in manuscript:*

*Page 1, Line 1-2:*

*" Atmospheric pollution from shipping and their contributions to air quality degradation in a port site in Shanghai "*

2. Introduction. The adoption of a DECA (Domestic ECA) is quite interesting and it would be even more interesting if a more detailed discussion is included. For example, it would possible to comment on the efficacy of this measure in reducing the impact of shipping on local pollution levels. It is also worth to mention that a recent work (Contini et al., 2015 – Atmospheric Environment 102, 183-190) showed that application of "domestic" restrictions on the fuel quality could be effective in reducing not only local $SO_2$ concentrations but also primary emissions of

particles from ships. I believe that a discussion on this aspect would be appreciated by the readers.

Authors' Response:

We acknowledge the referee's suggestion to include a further discussion on DECA. In fact, the DECA strategy has not only been implemented in YRD region, but also in Pearl River Delta (PRD) and Bohai-Rim area, which constitute the three major shipping regions in China. According to monitored data at several sites adjacent to port area in YRD, there has been >20% reduction in ambient $SO_2$ concentration in the same period before and after the DECA measure, although part of the $SO_2$ reduction are attributable to emission control measures in coal burning in power plant, boilers, furnaces and domestic use in China. There is a published study which dealt with the effectiveness of DECA in PRD region, estimating that the DECA measure could result an average reduction of 9.54% in $SO_2$ and 2.7% in $PM_{2.5}$ in land areas (Liu et al., 2018).

*changes in manuscript:*

*Page 2,Line 30 – Page 3, Line 4:*

*" This limitation level of sulphur is still higher than the implemented legislation in many harbors/ports in Europe and US (0.1%) (IMO, 2017). The DECA measure was currently implemented mainly in three major shipping areas including PRD, Pearl River Delta - PRD, and Bohai Rim region in China. Efficiency of the ECA measures has been tested in other places (Contini et al., 2015;Merico et al., 2017). It was shown that the control strategies in sulphur in fuel could generate synergetic reduction in both SO2 and primary PM release from ships. The benefits of DECA measure in YRD were also suggested by the reduction of SO2 concentration at several monitoring sites in port areas. There is a published study which dealt with the effectiveness of DECA in PRD region, estimating that the DECA measure could result average reduction of 9.54% SO2 and 2.7% PM2.5 in land areas (Liu et al., 2018)."*

3. Sections 2.3 and 2.4. It is often mentioned the high temporal resolution of SPAMS measurements, I would suggest to explicitly report the numerical value.

Authors' Response:

SPAMS is a real time measurement and particles drawn into instrument and analyzed consecutively. Depending on the analysis objective, the temporal resolution of SPAMS can be set to several minutes and this value is adjustable.

*changes in manuscript:*

*Page 5,Line 21-22:*

*" The temporal resolution of SPAMS (seconds or minutes) makes it suitable to couple with online gaseous data to identify ship emissions."*

4. Page 6 (lines 1-5). V-particles measured without the presence of $SO_2$ peaks are interpreted as due to the use of low-sulphur content fuel, however, it would not be possible that they are coming from other industrial (or anthropic in general) sources? Some words on this should be included.

Authors' Response:

Thanks for the valuable suggestion. The authors have not considered the fact that vanadium could also be released from other sources as petroleum refinery and other industries. There is indeed petroleum company in adjacent regions and other industries, whose influences should be considered. According to the measurement data, however, the occurrence probability of

vanadium particles plume without SO$_2$ peak is certainly small (3% in cases), so that the interferences of industrial sources will not greatly affect the results in this paper. In another aspect, the determination of contribution of shipping emissions will exclude the industrial influences by confining the wind directions only from the port sector, so that the industrial interferences (from land directions) were set to a minimum.

*changes in manuscript:*

*Page 7,Line 29-34:*

*" The occurrence probability of this kind of event is low (3% in cases). The causes of this kind of events are two-fold: firstly, it is maybe due to the anchored ships burning low sulfur content oil (<0.5 % m/m) to comply with regulations in the port region, which came into force on April 1, 2016; secondly, it is also possible that the vanadium particles be emitted from industry sources such as petroleum refinery companies in this region. The wind directions when these events happened support both of the proposed causes."*

5.   Page 7, line 23. To speak at this level of BC is not really useful, likely authors mean EC.

Authors' Response:

Thanks for the penetrating comments. 'BC' is replaced by 'EC' in manuscript.

*changes in manuscript:*

*Page 9,Line 25:*

*"such as EC particles, having significantly larger cross sections to reflect laser light and be detected in SPAMS"*

6.   Page 8, lines 2-3. This sentence is not clear and should be re-written. I believe that authors means that ultrafine particle concentrations could be a better metric compared to mass concentrations to investigate the impact of shipping to atmospheric aerosol.

Authors' Response:

This paragraph has been removed.

*changes in manuscript:*

*this sentences has been removed.*

7.   Page 9, line 30. The approach based on this formula was originally developed in Contini et al (Journal of Environmental Management 92 (2011) 2119-2129) and successively used by other authors. I believe that it would be fair to mention this aspect.

Authors' Response:

This item of reference was replaced by the suggested one.

*changes in manuscript:*

*Page 6,Line15-16:*

*"The calculations method of ship emission contributions used in this study, which was originally developed by (Contini et al., 2011), is based on the extraction of ship emission plumes from background concentrations of pollutants :"*

8.   Looking at the size distributions reported in figures 4 and 6, it appears that V particles are especially relevant for ultrafine particles, however this aspect is not deeply investigated on the evaluation of the impacts. It would be possible to use the approach discussed on page 9 to

investigate the size dependency of the impacts of shipping, eventually estimating the impacts for different size ranges. I believe that, if a sufficient statistics could be obtained, this will give very useful additional information compared to the impact on total particle number reported in Table 2.

Authors' Response:

Thanks for that suggestion. The authors agree with the advice to separate V particles into different size ranges and evaluate their impact individually. After inspection of V particle size distribution, the particle diameters will be grouped into three size ranges: <0.4µm;0.4-0.8µm;>0.8µm; Their impacts will be calculated and discussed as a function of size. The next version of manuscript, which will be submitted soon, will cover this topic together with discussions as suggested in 9# comment raised by the referee.

*changes in manuscript:*

*Page 13,Line4-6:*

*" Contributions of PNCv in different particle size ranges were also calculated in table 2. In either of reference periods (excluding or including land-based emissions), ship emission contributions to PNCv in smaller size range (0-0.4µm) are larger compared with PNCv in larger size ranges (0.4-0.8 µm, 0.8-2.5µm)."*

9. Page 10, lines 13-23. The comparison with shipping impact measured in other ports is certainly interesting, however, it is done on relative impacts and not on absolute contributions due to shipping activities this means that it depends not only on ship traffic but also on the contributions of the other sources acting on the specific measurement site. This should be mentioned because it could explain some of the apparent discrepancy mentioned by the authors. In addition, I would suggest to expand the comparison to other ports analysed with the high temporal resolution approach (Merico et al Transportation Research Part D 50 (2017) 431–445) but also with other complementary approaches (see for example Viana et al 2014 Atmos. Environ. 90, 96–105).

Authors' Response:

The authors agree with that advices. We acknowledge that the pollution absolute contributions from shipping are also important. The discussions with other port will also include more relevant studies in the literatures. Absolute contributions of shipping emissions and relevant discussions will be embodied in the next version of manuscript which will soon be submitted.

*changes in manuscript:*

*Page 13,Line7-17:*

*"The relative contributions of PNCv from ship emission is apparently higher than PM2.5 on mass concentration. Previous study showed that the direct PM2.5 contribution from ship traffics lies within 1-8% range (Contini et al., 2011;Contini et al., 2015). Recent studies carried in Mediterranean region found that ship emission contributed 0.3-7.4% PM2.5 concentrations in port areas (Merico et al., 2016). Ship emission studies in Europe and other regions was reviewed, and its concluded that shipping traffics contributions to PM2.5 were in 1-14% range, with higher contributions with decreasing particle size (Viana et al., 2014). The calculated value of PM2.5 in the present site is within the reported ranges. Recently (Merico et al., 2017) compared ship traffic atmospheric impacts using inventories, experimental data and modelling approaches in Adriatic-Ionian port areas, and found that ships contributed 0.5-7.4% PM2.5 in these areas. The same study*

*further found that ship traffics contribution to particle number concentrations (PNC) is 2-4 time larger than mass concentrations of PM2.5. The PNC is not currently measured, instead the size distributions, PNC contributions of vanadium particles in different sizes, as measured by SPAMS, apparently agrees with these previous work. "*

*Page 13,Line35-Page 14, Line 6:*
*"However, in an absolute sense, this study estimate that ship emissions contribute to 5.68 μg/m3 SO2, 3.00 μg/m3 NOx and 1.57 μg/m3 PM2.5 during the sampling period. These values are comparable or higher than the reported results in ports in other regions (Viana et al., 2014). For example, a previous study found that the ship emitted particles contributed 0.8 μg/m3 (primary particles) and 1.7 μg/m3 (secondary particles) in Bay of Algeciras (Viana et al., 2009). Due to the adjacency of the site to port, the calculated PM2.5 contribution could be largely deemed as primary for present site. The relative contributions of pollutants are partly compensated by the higher background pollution levels in this region."*

10. Regarding the impacts reported in Table 2, it would be possible to estimate the uncertainties?

Authors' Response:

In the preparation of Table 2 we have considered the estimation of uncertainties, which is a conventional practice in scientific report. The uncertainties of calculation may stem from sources such as the identification of plumes, the definition of port sector directions and the gaseous and particulate measurement itself. Some of these sources are found difficult to define. To be consistent with the original study (Contini et al., 2011), the uncertainties in this work will be estimated by inspection variations in slight changes of wind direction and the elimination of data of low wind velocity (< 0.5 m/s). The uncertainties will be included in the next version of manuscript.

*changes in manuscript:*
*Page 6,Line19-25:*
*"The uncertainties of $\varepsilon_A$ determined in this method could arise from several factors, such as the definition of port direction sector, the definition of plumes (the threshold level that discriminate plumes and the background), and pollutants and wind field measurements. This study estimate the uncertainties by subjecting $\varepsilon_A$ to the slight adjustment of the port directions by ±10° and pollutants threshold levels by 20% to inspect its variations. To conform to the original work (Contini et al., 2011), calm wind periods (wind speed < 0.5 m/s) were considered in the evaluation of uncertainties (either excluding or including calm wind periods)."*
*Page 27,Line5-10:*
*"*

| (%) | In port sector (excluding land-based emissions) | | Entire period (including land-based emissions) | |
|---|---|---|---|---|
| | *Average* | *range* | *Average* | *range* |
| $SO_2$ | 57.2 | (49.2, 64.8) | 36.4 | (29.2, 40.2) |
| $NO$ | 71.9 | (57.0, 84.6) | 0.7 | (0.2, 1.7) |
| $NO_2$ | 30.4 | (24.7, 34.6) | 5.1 | (3.7, 7.9) |
| $O_3$ | -16.6 | (-18.8, -13.4) | -0.9 | (-2.8, -0.4) |

| | | | | | |
|---|---|---|---|---|---|
| *PM$_{2.5}$* | | *27.6* | *(22.5, 33.2)* | *5.9* | *(3.4, 9.6)* |
| *Vanadium particles\** | *(0-0.4µm)* | *79.2* | *(73.9, 85.0)* | *57.1* | *(50.6, 64.0)* |
| | *(0.4-0.8µm)* | *75.3* | *(68.1, 82.0)* | *44.7* | *(38.1, 52.3)* |
| | *(0.8-2.5µm)* | *76.6* | *(70.4, 82.9)* | *47.0* | *(41.3, 52.9)* |
| | *(0-2.5µm)* | *77.0* | *(70.6, 83.1)* | *49.5* | *(43.0, 56.7)* |

*Length of sampling (in hours): Entire period: 2256; Port sector: 1136; In plume: 694;Non-plume: 1563; Non-plume (port sector): 625.*

*\* Particle number contribution*

*"*

11. Page 11 line 5. This sentence is not clear. Authors likely mean that the impact of shipping is more relevant and clearly discernible on SO$_2$ and V particles compared to the other pollutant analysed. Could authors clarify?

Authors' Response:

Thanks for the advice. The original sentence would be revised.

*changes in manuscript:*

*Page 15,Line8-9:*

*"During plumes the SO2 and vanadium particles concentrations has demonstrated well synchronized peaks, which could be reliably used to indicate the arrival of ship emission plumes."*

12. In the supplementary material it is reported "…in present study the online single particle measurement was utilized to indicate the occurrence of shipping emission Plumes…" however in the main text was mentioned that both particles and SO2 concentrations were used. Please clarify this apparent contradiction.

Authors' Response:

Thanks for pointing out this unclarity. The intention behind the supplementary text is to give the reader a extended discussion on the identification method of ship emitted particles. As explained in supplementary text, the adoption of vanadium tracer could not guarantee that every single ship emitted particles in a plume be identified. From figure S1 it could be inferred that only a fraction of ship emitted particles in plumes are identified by vanadium peak criteria, because not every individual particles in a emission plume contain a detectable vanadium content.

The mentioned sentence "…in present study the online single particle measurement was utilized to indicate the occurrence of shipping emission plumes…" is emphasizing this fact and not really mean that only the vanadium particles were used to indicate the influence of plumes. Actually both of vanadium particles and SO$_2$ concentration were applied to identify plumes in data analysis process. In fact, the SO$_2$ concentration is critical to identify plumes in which few vanadium particles were present.

To prevent possible confusion, the original sentence is revised.

*changes in manuscript:*

*Supplementary file: Page 2,Line26-28:*

*"in present study the online single particle measurement, together with synchronous SO2 concentration, was utilized to indicate the occurrence of shipping emission plumes"*

**Minor corrections**

13. Page 1, line 15. Better "particle size distributions".

Authors' Response:

Suggestion accepted.

*changes in manuscript:*

*Page 1,Line15:*

*"concentrations (PM2.5), particle size distributions and chemical composition of individual ship emission particles"*

14. Page 1, line 28. Please eliminate the initial S.

Authors' Response:

Suggestion accepted.

*changes in manuscript:*

*Page 2,Line2:*

*"Ship emission constitutes an important source of gaseous and particulate pollution world wide"*

15. Page 2, line 19. Subscript for $SO_2$. The same in page 4 (line 25).

Authors' Response:

It has been corrected.

*changes in manuscript:*

*Page 5,Line32:*

*"concurrent $SO_2$ concentrations were utilized to locate ship emission plumes when sharp SO2 peaks occurred"*

16. Page 7, line 14. Better "different size distributions…".

Authors' Response:

Accepted.

*changes in manuscript:*

*Page 9,Line14-15:*

*"Wind roses and size distributions of fresh and aged ship emission particles were also distinguishable."*

17. Page 8, line 12. Better "by the dominant". In addition, I would remove etc, if necessary please mention explicitly.

Authors' Response:

Suggestion accepted.

*changes in manuscript:*

*Page 10,Line16-17:*

*"In the positive mass spectra the V-OC type are characterized by the dominant organic peaks like $C_2H_3^+$, $C_2H_5^+$, $C_2H_3O^+$,".*

18. Page 8, line please remove etc. as above.

Authors' Response:

The 'etc.' is removed.

*changes in manuscript:*

*Page 10,Line31-33:*

*"The temporal concentrations of these particle types were poorly correlated ($R^2<0.4$), suggesting they were emitted differently. Since these particles were detected in a portside environment, they were assumed to be emitted by ships of different engine types or modes of operation."*

19. Page 8, line 25. Better "is therefore not attempted…"

Authors' Response:

Accepted.

*changes in manuscript:*

*Page 11,Line3:*

*"is not yet available, it is therefore not attempted to link V-OC particle plumes to specific ship types directly in the present"*

20. Page 9, line 2. > 0.5 μm

Authors' Response:

Original letter replaced with the Latin letter 'μ'.

*changes in manuscript:*

*Page 11,Line18-19:*

*"ash spheres from combustion process of inorganic constituents in RFO and lubricants, are mainly detected in larger size range ( > 0.5 μm)"*

21. Page 11, line 10 ozone without capital letter.

Authors' Response:

Accepted.

*changes in manuscript:*

*Page 15,Line15:*

*"$NO_x$ and $SO_2$ emissions in port regions, resulting 11-33 % ozone consumption compared with urban region of Shanghai."*

**Refereces**

Contini, D., Gambaro, A., Belosi, F., De Pieri, S., Cairns, W. R. L., Donateo, A., Zanotto, E., and Citron, M.: The direct influence of ship traffic on atmospheric PM2.5, PM10 and PAH in Venice, Journal of Environmental Management, 92, 2119-2129, 10.1016/j.jenvman.2011.01.016, 2011.

Contini, D., Gambaro, A., Donateo, A., Cescon, P., Cesari, D., Merico, E., Belosi, F., and Citron, M.: Inter-annual trend of the primary contribution of ship emissions to PM2.5 concentrations in Venice (Italy): Efficiency of emissions mitigation strategies, Atmospheric Environment, 102, 183-190, 10.1016/j.atmosenv.2014.11.065, 2015.

IMO: Emission Control Areas (ECAs) designated under MARPOL Annex VI, 2017.

Liu, H., Jin, X., Wu, L., Wang, X., Fu, M., Lv, Z., Morawska, L., Huang, F., and He, K.: The impact of marine shipping and its DECA control on air quality in the Pearl River Delta, China, Science of The Total Environment, 625, 1476-1485, https://doi.org/10.1016/j.scitotenv.2018.01.033, 2018.

Merico, E., Gambaro, A., Argiriou, A., Alebic-Juretic, A., Barbaro, E., Cesari, D., Chasapidis, L.,

Dimopoulos, S., Dinoi, A., Donateo, A., Giannaros, C., Gregoris, E., Karagiannidis, A., Konstandopoulos, A. G., Ivosevic, T., Liora, N., Melas, D., Mifka, B., Orlic, I., Poupkou, A., Sarovic, K., Tsakis, A., Giua, R., Pastore, T., Nocioni, A., and Contini, D.: Atmospheric impact of ship traffic in four Adriatic-Ionian port-cities: Comparison and harmonization of different approaches, Transportation Research Part D-Transport and Environment, 50, 431-445, 10.1016/j.trd.2016.11.016, 2017.

Ambient measurement of shipping emissions in Shanghai port areas
journal article response
en

**Response to Referee's Comments on acpd-2018-737:**
**"Ambient measurement of shipping emissions in Shanghai port areas"**

The many items of comments raised by the referee #1 are responded individually as following. Text is present in the format of:

Referee's comments – Black;

The Authors' responses – Blue.

Changes in manuscript - *Red, Italic*

**Response to Referee #1**

This study conducted field measurements from June to September in 2016 at Shanghai port in order to understand the impact of ship emissions on the air quality in portside. Trace gases, PM2.5 and vanadium particle number concentrations were continuously monitored at the site. Ship plumes were clearly captured by the instruments. SO2 and vanadium particle number concentrations correlated well with ship plumes. Four types of ship plumes were identified based on the mass spectra of Single Particle AMS. The contributions of ship emissions to different air pollutants in the atmosphere and in the air masses from port directions were quantified. Given that Shanghai port is the largest port in the world, this study will add values to existing literature of ship emission studies. However, the manuscript is not well organized/written and has room to be improved. In addition, there are quite a lot of grammar errors and technical mistakes, which sometimes make the reviewer confused. Furthermore, some discussions and conclusions are lack of evidence. As such, this manuscript can be considered for publication after the following specific comments are well addressed.

**Specific Comments:**

Abstract:

Firstly, English needs editing by a native English speaking professional or company. For example, line 16: … that shipping emissions is a major….".

Response:

The language problems, as the author responded in reply to RC2, will be corrected in the next edition of manuscript. The author acknowledge the referee's effort.

*Changes in manuscript:*

*Pages1 , Lines16 :*

*Original sentence removed*

Secondly, there are also some technical mistakes. One example, lines 14-15: Gaseous (NO, NO2, SO2, O3) and particulate concentrations (PM2.5)… It should be "The concentrations of gaseous pollutants (NO…..) and fine particulate matters (PM2.5)…". Also, both shipping emission and ship emission are used throughout the manuscript which should be consistent. Another problem at lines 18-20, the subject is "Single particle mass spectra of fresh shipping emission" but the last words became "…and nitrate peaks in aged particles". This is really confusing the reviewer.

Response:

1. Original sentence revised as: "The concentrations of gaseous pollutants (NO, $NO_2$, $SO_2$, $O_3$) and fine particulate matters ($PM_{2.5}$), size distribution and chemical composition of ship emission

particles were continuously monitored for 3 months".

*Changes in manuscript*:

*Pages 1 , Lines 14 -16 :*

*"The concentrations of gaseous (NO, NO2, SO2, O3) and particulate concentrations (PM2.5), particle size distributions and chemical composition of individual ship emission particles were continuously monitored for 3 months. "*

2.  Both of the "shipping emission" and "ship emission" are seen in literatures. This manuscript unify them to "ship emission".

*Changes in manuscript*:

*All the "shipping emission" are changed to "ship emission"*

3.  The sentence will be "Single particle mass spectra of fresh shipping emission were identified based on the dominant peaks of Sulfate, EC and indicative metals of V, Ni, Fe and Ca".

*Changes in manuscript*:

*Pages1 , Lines20-21 :*

*"Single particle mass spectra of fresh ship emission were identified based on the dominant peaks of sulfate, elemental carbon (EC) and indicative metals of V, Ni, Fe and Ca "*

Thirdly, the abstract should provide specific and detailed findings rather than common senses. The only specific finding described in the abstract is probably the last sentence. The others are all about common knowledge which is also applied to any other ports. What is the uniqueness of the study port?

Response:

Abstract section will be revised concentrating on the key findings and the uniqueness of present study. It will include aspects on fresh ship emission particle signatures, gaseous pollutants characters from ship emission and their contributions, size resolved ship emission particle contributions to portside and the importance of separation of land emissions.

*Changes in manuscript*:

*Pages1 , Lines12-28 :*

*"Abstract. Growing shipping activities in port areas have generated negative impacts on climate, air quality and human health. To better evaluate the environmental impact of ship emissions, in the summer of 2016 ambient air quality measurement was carried out at Shanghai port, one of the busiest ports in the world. The concentrations of gaseous (NO, NO2, SO2, O3) and particulate concentrations (PM2.5), particle size distributions and chemical composition of individual ship emission particles were continuously monitored for 3 months. In online measurement the ship emission plumes were clearly distinguishable of both gaseous and particulate matter, which have shown synchronized peaks during plumes. The SO2 and vanadium particles numbers were found to correlate best with ship emissions in Shanghai port. Single particle mass spectra of fresh ship emission were identified based on the dominant peaks of sulfate, elemental carbon (EC) and indicative metals of V, Ni, Fe and Ca. Temporal trends and size distributions of major ship emission particle types were discussed. The sampled ship emission particles in the port site mainly concentrated in smaller size range where their number contributions are more apparent than their mass. For a costal port close to urban region, the land-based emissions have generated important*

*impacts to the portside air quality, especially for NOx and PM2.5. Quantitative estimation conducted in the present study show that in port region ship emissions contributed 36.4 % SO2, 0.7 % NO, 5.1 % NO2, -0.9 % O3, 5.9 % PM2.5, 49.5 % vanadium particles if land-based emissions were included, and 57.2 % SO2, 71.9 % NO, 30.4 % NO2, -16.6 % O3, 27.6 % PM2.5, 77.0 % vanadium particles if land-based emissions were excluded.*

*"*

Introduction:

As there are too many grammar errors, I have made some comments and revisions on the manuscript. I will submit my comments with the manuscript.

Response:

All the grammatical errors will be corrected in next edition.

*Changes in manuscript*:

*The grammar errors has been corrected throughout the manuscript.*

Experimental:

It is not clear whether the sampling site is downwind location of the port or not, or whether the ship plumes could really arrive at the sampling site or not. The authors should provide more detailed description of the site. What were the prevailing winds during the sampling period and how to ensure the capture of ship plumes? There is also no information about the station. Is it a container or mobile vehicle? What is the height of the station if the outlet of the sampling tube was 3.5 m above the ground?

Response:

The referee seems to be suspicious about the possibility to detect ship emissions in port site. This question was answered by the many published studies in portside across the world (Healy et al., 2009;Ault et al., 2010;Contini et al., 2015;Merico et al., 2016).

For an online, continuous monitoring the prevailing wind notion is not useful because a prevailing wind direction does not imply wind will stay in that direction. They could shift to any directions during a long period of time. The monitoring site is certainly in the downwind direction of port when the wind is in 300°-0°-120° sector, which was clearly indicated in Fig. 1 in manuscript.

During the sampling period the prevailing wind is from southeast which is typical for summer season, as presented in Figure R1. Wind in other directions including port directions also occurred although with less frequency.

The station is on the south riverbank, as already stated in manuscript and illustrated clearly in Fig. 1 in original manuscript. The author could not understand how it is thought the station is on a container or mobile vehicle.

*Changes in manuscript*:

*Pages3 , Lines 30- Page4 ,12:*

*"The Waigaoqiao Port (31.337° N, 121.665° E) locates in the northeast of Shanghai city (Fig. 1) and is the largest port in China. The port has about 7 km of docks (3 km north section and 4 km south section). In 2016 the port has yearly traffic of 367 M-tons of goods and container volume of 37.13 million TEU (Twenty-foot Equivalent Unit). Ship categories in port consist of container vessel (62.4 %), tug (18.6 %), oil tanker (9.0 %), bulk (1.8 %), Ro-Ro (1.7 %) and other ships (6.5 %) (private data from authority). A power plant and a shipbuilding factory reside between the north and south section of port, which have their own docks. The portside air monitoring station locates on the*

*south bank of Yangtze River, 400 m away from the nearest dock. Gaseous and particulate matter instruments were installed within the station with the main sampling tube extending through the roof. The outlets of the main sampling tube was 1m above the station roof and 3.5m above the ground. Ship emission plumes could influence the site in wind direction of about 300°-0°-120° sector (Fig. 1). In the summer season the prevailing wind direction of the site is southeast direction. In the supplementary file the wind rose during the sampling period is provided (Fig. S1). In ~55% of time the site was under the impact from port emissions. To the south and west of site there were intense road traffics of container trucks and the Shanghai outer ring. Traffic emissions in south and west directions have important influences on air pollutions at the monitoring site when inland wind prevails.*

*"*

[Figure]

Figure R1. Portside wind rose during the study period.

Was the CO measured? Though it was claimed that calibration and maintenance of the instruments were regularly performed, brief QA/QC procedures and detection limits are still requested. Nothing was mentioned about the QA/QC of PM2.5 monitoring.

Response:

During the period the CO analyzer is not functional, so that the discussion on CO is not included in this study. The QA/QC procedures of NO-$NO_2$-$NO_x$, $O_3$, $SO_2$ and particulate $PM_{2.5}$ is under guidance of << Technical specifications for operation and quality control of ambient air quality continuous monitoring system for $SO_2$, $NO_2$, $O_3$ and CO >> and <<Technical specifications for operation and quality control of ambient air quality continuous monitoring system for Particulate matter ($PM_{10}$/$PM_{2.5}$)>> in China. QA/QC procedures in the national air quality monitoring stations are the same. Procedures and detection limits will be added.

*Changes in manuscript*:

*Pages 4 , Lines 14-28 :*

*"The concentrations of gaseous NO-NO2-NOx, SO2, and O3 were measured continuously from Jun-21 to Sep-21, 2016. The gaseous pollutants were monitored by a suit of Thermal Scientific analyzers (NO-NO2-NOx, model 42i; SO2, model 43i; O3, 49i) installed in the monitor station. Calibration and checking of instruments were regularly performed by zero checks (through a zero air generator) and span checks (through standard NO2 and SO2 gas of known concentrations; the O3 standard was generated through a calibration photometer system); The PM2.5 concentrations were monitored by oscillating microbalance method (Thermo TEOM 1405-F). Calibration of TEOM is not relied on standard, for the aerosol mass on a filter was monitored by the oscillation frequency change of the tapered element over specified time. The regular maintenance of TEOM includes the changing of filters before the filter loading approach 100%. The flow rate of TEOM was checked using a flowmeter. The lower detection limits of these instruments are: 0.4 µg/m3 (NO, NO2); 0.5 µg/m3 (SO2); 0.5 µg/m3 (O3); 1 µg/m3 (PM2.5). Weather conditions (temperature, humidity, pressure, wind speed and direction) were monitored by a mini-weather station installed on the rooftop of the station. The weather station sensor was about 1 m above the station roof    and 3.5 m above the ground. Data from all the instruments and the monitor was managed in a customized database and set to 5 min resolution. Atmospheric pollutants concentrations in Shanghai city area, including gaseous pollutants and PM2.5 concentrations, were monitored concurrently at 9 national air quality monitoring stations in 1h resolution. The averaged pollutants concentrations at these stations during the sampling period were included for comparison."*

It is not clear how the components in particles such as vanadium were identified and quantified by the SPAMS. Detailed information is needed.

Response:

SPAMS identify particle composition, such as vanadium, in mass spectrometry method. In the ionization laser beam in SPAMS, the components in particles are ionized into ions carrying charge, then they are separated in the Time-of-Flight tube by atomic mass of the ion. Lighter ions, such as $H^+$, transit fastest in TOF tube and produce peaks in shortest time. The $208Pb^+$ is heavier so that it reach the MS detector with longer time. This will result a spectrum sorted by ion atomic mass. Components of different atomic mass produce peaks in different position in mass spectrum. Vanadium in particles normally produce peaks at mass = $51V^+$ and 67 ($VO^+$) and thus can be identified. This is the basics of MS and unnecessary to be included.

*Changes in manuscript*:

*None*

Data analysis: in the results, pollution and wind roses were presented while nothing is described about the method to draw pollution and wind roses, and how to explain the pollution and wind roses. In addition, the method of calculation of shipping contributions is improperly placed in the "Results" section, which should be described in the "Experimental".

Response:

Pollution roses are normal and frequently presented in literatures. The author has used the most normal method to draw a wind rose: firstly perform statistics (can either be minimum, maximum, counts, mean value, or any statistics) on relevant pollutant concentrations (e.g., number concentrations of vanadium particles) in every wind directions and then do the plot. If the wind

speed data is also included, then another frame of information appears and a map is obtained , which will produce Fig. 7 in original manuscript.

The explanation of wind roses is equally straightforward, in that it could demonstrate, in a intuitive manner, the direction of emission sources relevant to the observation site. The author will explaine wind roses within text in appropriate places in next edition.

The method to quantify the contribution from ship emission will be moved to 'Experimental' section.

*Changes in manuscript*:

*Line 6, Line 14-25:*

*"**Evaluation of ship emission contribution***

*The calculations method of ship emission contributions used in this study, which was originally developed by (Contini et al., 2011), is based on the extraction of ship emission plumes from background concentrations of pollutants :*

$$\varepsilon_A = \frac{\Delta C_A F_{plm}}{C_A}$$

*Where: $\varepsilon_A$, ship emission contributions of pollutants A; $\Delta C_A$, the difference between average concentrations during plumes and non-plumes; $F_{plm}$, fraction of cases of plumes; $C_A$, the average concentration of pollutant A during reference period. The uncertainties of $\varepsilon_A$ determined in this method could arise from several factors, such as the definition of port direction sector, the definition of plumes (the threshold level that discriminate plumes and the background), and pollutants and wind field measurements. This study estimate the uncertainties by subjecting $\varepsilon_A$ to the slight adjustment of the port directions by ±10° and pollutants threshold levels by 20% to inspect its variations. To conform to the original work (Contini et al., 2011), calm wind periods (wind speed < 0.5 m/s) were considered in the evaluation of uncertainties (either excluding or including calm wind periods).*
*"*

Page 5, line 3: what is ART-2a algorithm? This method was mentioned to be used to the searched clusters to generate sub-clusters of particles. However, no information at all about this method was provided.

Response:

The ART-2a is a classification algorithm conventionally adopted by SPAMS community to group similar particle based on the particle similarities. A reference on ART-2a algorithm will be added here in case that the reader want further information on the algorithm.

*Changes in manuscript*:

*Line 6, Line 9-11:*

*"Then the ART-2a algorithm (Song et al., 1999) was applied to the searched clusters to generate sub-clusters of particles (Vigilance=0.85; Learning=0.05; Iteration=20)."*

Results and discussions

Page 5, line 10: which typically persist for a few hours: can you tell us the specific hours in your study rather than vague value like this?

Response:

The specific value of hours is variant, most of them fall in range of 3-6 hours.

*Changes in manuscript*:

*Page 6, Line 30:*

*"as arrival, hoteling and departure, which typically persist for a few (mostly 3-6) hours."*

Page 5, lines 21-25: The discussion here is questionable. By looking at Figure 2, whenever ship plumes were captured, both NO and SO2/vanadium levels were high and correlated well. On what basis, the authors claimed the NOx in plumes reaching the site was aged? Using the NO/NO2 ratio in the plumes? Compared to the ratio measured in other countries and probably different type of ships? This is not convincing. Besides, NO2 is also emitted from shipping as a primary pollutant.

Response:

We have not tested NO/NO2 ratio in the exhaust because the observation was carried out in a station on land. The author also know that NO2, together with NO, is released as primary emissions. However their ratio, upon their emission into atmosphere, will subject to change quickly through the oxidation of NO into NO2 in the existence of ozone (O3), which is abundant in summer time. This was evidenced by the quickly reduced O3 level during plumes in Fig. 2 in manuscript. The referee seems to be doubtful about this reaction, which is the very basis that the NO-NO2-NOx analyzer is working on.

*Changes in manuscript*:

*None*

Page 5, line 26-29: given that shipping emission is a major source of PM2.5, it is odd that no PM2.5 peaks were found during the ship plumes in Figure 2. The reason provided by the authors is quite confused. Is it because ship emits sub-micron particles or because the malfunction of the PM2.5 monitor?

Response:

The authors have stated that $PM_{2.5}$ peaks was not as apparent as that of $SO_2$ and $NO_x$ in plumes, and have not states that no $PM_{2.5}$ peaks were found. It should be made clear that only a short period of data was shown in Fig.2 only with the purpose of illustrating temporal variation in plumes. In the last question the referee is suspecting that $PM_{2.5}$ analyzer was in malfunction. The author would like to illustrate another longer period of data in Figure R2 (shown below) and let the referee make his judgement. The sharp peaks of $SO_2$ in Figure R2 could help to locate the plumes. If the $PM_{2.5}$ instrument is in malfunction, how did the $PM_{2.5}$ instrument happened to malfunction only in plumes? In another aspect, the SPAMS particle number concentration have shown good correlation with the $PM_{2.5}$ measurement. How did the two instruments both malfunction? It sounds ridiculous.

*Changes in manuscript*:

*None*

[Figure]

Figure R2. Temporal variations of $PM_{2.5}$, $SO_2$ concentrations and SPAMS analyzed particle number concentrations during Jul-20 to Jul-30 in this study.

Page 5, lines 31-32: what is the basis for the definition of ship plumes using the minimum threshold of delta SO2? In particular, the authors later claimed that in some cases the SO2 peaks were absent?

Response:

Very clear, frequent sharp peaks of $SO_2$ over the background $SO_2$ concentration is suggesting that they are emitted from adjacent combustion sources. Concurrent increases of vanadium particles suggest they are combusting Residual oil. Wind directions during plumes mainly in port directions suggest they are from ships. The cases the $SO_2$ peaks were absent were rare( 3% cases), and will be explained in next edition of manuscript.

*Changes in manuscript*:

*Page 7, Line 29-34:*

*"The occurrence probability of this kind of event is low (3% in cases). The causes of this kind of events are two-fold: firstly, it is maybe due to the anchored ships burning low sulfur content oil (<0.5 % m/m) to comply with regulations in the port region, which came into force on April 1, 2016; secondly, it is also possible that the vanadium particles be emitted from industry sources such as petroleum refinery companies in this region. The wind directions when these events happened support both of the proposed causes."*

Page 6, lines 2-3: the reason for absent SO2 is contradictory to Figure 2. If the ships complied with the new regulations, why would you still see SO2 peaks in ship plumes? This kind of discussion is misleading.

Response:

As stated above, the cases the $SO_2$ peaks were absent were rare and will not affect significantly the result of the study. The new regulation only confine the Sulfur content in fuel, not eliminate the Sulfur from the fuel.

*Changes in manuscript*:

*None*

Page 6, line 16: "This result suggests that shipping activities are the main source of SO2 plumes in port". Please comment on the NOx emission from ships - is it not important, because of impacts of land-based traffics? But later in Page 10, lines 15-18, you claimed NO was higher than SO2 in fresh ship plumes.

Response:

The author think that NOx emission from the land-based traffic to the site is important considering all surrounding sources. The claim in Page 10, lines 15-18 was under the condition that land-based emission were excluded by considering the period when site was only under influence of port direction.

*Changes in manuscript*:

*None*

Page 6, line 18: it does not make sense to compare a site near sources with sites in a city without any detailed characteristics of the locations. It would be more meaningful to compare the ship emissions in this study with other similar studies conducted in Shanghai. In fact, there are a number of ship emission studies in this city.

Response:

The comment is not acceptable. We made comparison between air pollution level in a port site with that at the urban area of the same city, How did it make non sense? The referee had better suggest a study which he think make more sense.

*Changes in manuscript*:

*None*

Page 6, lines 21-24: the explanation of low PM2.5 levels at the port site is not convincing at all. Why would other pollutants from shipping emissions be higher if this was caused by clean air?

Response:

The slight lower $PM_{2.5}$ concentration at the port site is a fact. The author only postulated possible explanations.

*Changes in manuscript*:

*None*

Page 6, line 27: the title reads awkward. Should it be "Particles in the background air and the ship plumes"?

Response:

The title is revised as: " Discrimination of fresh and background ship emission particles in port site".

*Changes in manuscript*:

*Page 8, line 25:*

*"3.2.1 Discrimination of fresh and background ship emission particles in port site"*

Page 6, lines 28-31: These should be in "Experimental" section.

Response:

Will be revised.

*Changes in manuscript*:

*This section has been removed*

Pages 6-7: the first paragraph of section 3.2.1 is messy and lack of logic. It should be re-organized.

Response:

Not acceptable. The referee think it is messy and lack of logic if he is not familiar.

*Changes in manuscript*:

*None*

Page 7, line 13: The wind rose distribution….. in figure 3. This should be merged with the description of mass spectra for Figure 3 earlier. When it is first mentioned, you should describe all once.

Response: Is it a really publicly accepted rule in scientific paper?

*Changes in manuscript*:

*This sentence has been removed.*

The text will be considered for some reorganization.

Page 7, lines 16-17: "Background vanadium particles, … in all directions". This is not true in Fig 3(d).

Response:

Original text has stated "prominent" and "nearly".

*Changes in manuscript*:

*None*

Page 7, lines 20-21: There is no actual comparison at all. No idea about the particle size in background air, larger or smaller?

Response:

Revised as: " The size distributions of fresh vanadium particles were dominated by fine particles ( < 0.5 um in diameter), as shown in Fig. 4. ".

*Changes in manuscript*:

*Page 9, line 21-22*

*"The size distributions of vanadium particles as shown in Figure 4 indicates fresh vanadium particles with dominate particle numbers in smaller size range (<0.5 μm), compared with background ones."*

Page 7, line 22-23: "significant fine ship emission particles were still detected in fine size range". The terms have been randomly used everywhere. What do you mean fine ship emission particles? very non-professional description. It should be fine particles from ship emissions.

Response:

Will be uniformed to 'fine particles'.

*Changes in manuscript*:

*The manuscript use " particles in smaller size range (<0.5 μm)" if appropriate.*

Is this contradictory to your previous claim that PM2.5 in ship plumes is lower than that in urban air? In other word, the PM2.5 in background air could be more significant and be detected more significantly?

Response:

It should be noted that particles are measured by SPAMS in number concentrations, while $PM_{2.5}$ is measuring particle mass. The fine particles are only a fraction of $PM_{2.5}$ and not always correlated with $PM_{2.5}$. No contradictory identified.

*Changes in manuscript*:

*None*

Page 7, lines 24-27: These two sentences are repeated. the 2nd sentence contains grammar errors.

Response:

Revised as: "The non-spherical, fractal shape of fresh vanadium particles is consistent with the typical shapes from fresh combustion sources (Ault et al., 2010). ".

*Changes in manuscript*:

*Page 9, Line 25-28.*

*"The non-spherical fractal shape of fresh vanadium particles is seen with soot particle from fresh combustion sources. Similar observations were reported in other studies using single particle mass spectrometer in ultra-fine size range (Ault et al., 2010)."*

Page7, lines 28-30: these two sentences are not related. Lack of logic.

Response:

This discussion will be removed.

*Changes in manuscript*:

*This section has been removed.*

Page 7, line 30: "size distribution of fresh particles…". how do you define "fresh particles from ship exhaust" and "particles from ship emissions"?

Response:

Uniformed to "ship emission particles".

*Changes in manuscript*:

*This term are unified to "ship emission particles"*

Page 7, line 31:" …that the number concentration mainly concentrated in UF mode (<100 nm)". very poor English writing. It is non-professional at all.

Response:

This discussion will be removed.

*Changes in manuscript*:

*This section has been removed.*

Page 7, lines 32-33: No idea at all why PM2.5, NOx and SO2 were suddenly discussed here. This section is about SPAMS measurement data. Also, where showed that PM2.5 had less increase than NOx and SO2? Moreover, are you sure EF of PM are typically much lower than NOx and SO2 in all types of ships with different fuel?

Response:

This discussion will be removed.

*Changes in manuscript*:

*This section has been removed.*

Line 34: Do not understand if there is any connection with less significant increase in PM2.5 mass concentration.

Response:

Seen revisions proposed in last comment.

*Changes in manuscript*:

*None*

Page 8, lines 1-3: Terrible explanation. The authors lack basic knowledge.

Response:

The referee did not point out the what is the basic knowledge and how they are terrible. In discussion of aerosol mass and number concentrations, it is easy to understand that particle number multiplied the average mass of individual particles produce the total mass concentration. Consider the supposed two cases: An aerosol sample of low numbers of particles but with larger size, and an aerosol sample of larger particle numbers but small average size. The two case may generate the same mass concentration. However, the difference between the two cases is that the latter produce larger surface area which favors the secondary accumulation (secondary formation of nitrate, sulphate, organics) to particle surface. This is why the number concentration matters. Hope the explanation is less terrible to the referee.

*Changes in manuscript*:

*Page 16, line 1-4*

*''Since the size and mass of fresh exhaust particles are small, the mass concentration PM from exhaust pipes would be inappropriate to represent their real mass contribution after atmospheric aging. This study supports that particle number concentration (PNC) be included to fully characterize primary ship emitted particles.''*

Page 8, line 9: "The negative mass spectra" how? To me it should be negative m/z value for HSO4-. Mass spectra should not be negative. Please make it clearer. In addition, the whole sentence is confusing. "…other negative EC peaks…". Specifically what are they in the spectra?

Response:

The positive and negative mass spectra a common nomenclature in SPAMS literatures, and will not generate confusions. The EC peaks is already noted in 3.2.1 section when it appear the first time.

*Changes in manuscript*:

*Page 5, line 14-15*

*"Chemical composition of ionized particle is measured by a dual polar time-of-flight mass spectrometer to record signal for both negative and positive ions."*

Page 8, line 20: Firstly, this description is unclear. How could "chemical composition" suggest distinct physical properties"? Secondly, is Fig 6 about this? But it is clear that Fig 6 is about temporal pattern of particle number concentration.

Response:

The sentence is revised as "Temporal concentrations and size distributions of these particle types

are shown in Figure 6.".

*Changes in manuscript*:

*Page 10, line 27*

*"Temporal concentrations and size distributions of these particle types are shown in Figure 6."*

Page 8, lines 26-27: Did you scan it using TEM? If not, how do you know yours is the same as other studies?

Response:

The several sentences will be revised as: "In previous study an organic particle type was identified by TEM images of ship emission particles (Moldanova et al., 2013). This organic particle contain vanadium impurities, which agree with organic and vanadium signatures in mass spectra in Figure 5. During the incomplete combustion (e.g., starting up phase) the organic vapours in fuels will condense onto particles (ash, EC particles) in the cooler ambient environment, resulting uniform size distribution compared with other types (lower right panel in Figure 6). The V-OC type are more transient in that the peak width of its concentration peaks are normally narrower than other types (~1 hours contrasting to 3~5 hours of other types).";

*Changes in manuscript*:

*Page 11, line 1-7:*

*"The V-OC particles, although having low ionization probabilities, were found to concentrate in specific cases of plumes. Since the information of individual ships is not yet available, it is therefore not attempted to link V-OC particle plumes to specific ship types directly in the present study. The V-OC particles concentrated in specific ship emission plumes and its' number concentration peaks were usually narrower (~ 1 hour) than the other particle types (3~5 hour). Sizes of V-OC particles were more uniformly distributed as compared with the other types (Fig. 6). Similar organic-rich particles were identified from ship exhaust by other technique (Moldanova et al., 2013)."*

Page 8, lines 27-29: Again, you do not have evidence to say this - is the reason that the size distribution of V-OC is different from other types related to incomplete combustion?

Response:

This is an inference to explain the observed result. No existing study is found by author.

*Changes in manuscript*:

*The discussion is removed.*

Page 8, line 31: "…. suggesting they are principally emitted in specific phases of engine operations." Any references support this?

Response:

This item was omitted, as shown in previous comment.

*Changes in manuscript*:

*The discussion is removed.*

Page 8, line 33: "UF size…" I guess it means ultra-fine. If correct, what is the size range of ultra-fine particles?

Response:

It's size include < 400 nm (aerodynamic diameter) in this study. Next edition they will be uniformed

to "smaller size range (<0.5 um)".

*Changes in manuscript*:

*Terms will be uniformed to "smaller size range (<0.5 um)".*

Page 9, line 1: "….product of combustion of RFO (Moldanova et al., 2013)." But your measurement was conducted after the implementation of sulfur reduction regulations. This means the fuel used at berth is not RFO but clean fuel.

Response:

Clean fuel with S <0.5% will also produce Soot particles.

*Changes in manuscript*:

*None*

Line 2: are mainly detected in larger size range ( > 0.5 um) (Fig. 6). The peak particle size for V-EC, V-ECFe and V-Ash looks similar in Fig. 6. How would you say this?

Response:

The size distribution shape of V-EC, V-ECFe and V-Ash types in upper-right panel is presented in absolute numbers. In smaller size range (< 0.5 μm) their number concentration all declined due to the decreased detection efficiencies of SPAMS in this size range. However, their relative contributions change as a function of size as shown in bottom-right panel.

*Changes in manuscript*:

*None*

Lines 3-5: This fits other types of particles as well. I don't understand why you said this here. This is basic knowledge.

Response:

The referee likes to speak "basic knowledge". Is it really your basic knowledge if I have not present it?

*Changes in manuscript*:

*None*

Lines 5-6: "The origin of V-ECFe types are probably the result of internal mixing between V-EC and V-Ash particles". Any possible reasons for this speculation?

Response:

This statement and the latter one has been removed.

*Changes in manuscript*:

*The discussion is removed.*

Line 7: how do you know or do you believe it?

Response:

See previous statement.

*Changes in manuscript*:

*The discussion is removed.*

Lines 9-18: The whole paragraph is nothing to do with your results but information about other

studies. This should be in "Introduction" section.

Response:

Will be moved to "introduction".

*Changes in manuscript*:

*It has been moved to "Introduction " section*

Lines 19-20: Very confused statement and unclear purpose.

Response:

Revised

*Changes in manuscript*:

*Page 12, line 5-7:*

*"For a coastal port, the evaluation of ship emission to air quality needs to identify impacts from land-based emissions. Obviously these land-based emissions are making greater influences to portside air quality than a marine port far from coast (Zhao et al., 2013)."*

Lines 22-23: "… clearly showed that they are under the overwhelming influences of land emissions on the sampling site." Why? How do we read the pollution roses in Fig 7? How do we know they were affected by other sources?

Response:

The information in wind roses in figure 7 is rather clear that the port site PM2.5, NOx are strongly influenced by land-based emissions, while the SO2, and ship emitted vanadium particles are under the major impact of ship emissions in port. The ozone wind rose indicates apparent depletions where NOx and SO2 are concentrated.

*Changes in manuscript*:

*None*

Lines 26-27: "Because the air pollution in this two conditions are so different,……" Which two conditions?

Response:

It is referring to periods when the site is influenced by land-based and port emissions.

*Changes in manuscript*:

*Page12, line18-21:*

*"Obviously the port site was receiving very different pollution impacts from land emission and the ship emissions in port. Present study tries to separate land-based emission influences by limiting wind directions only in port directions. In the calculation of ship emission contribution, two reference periods were considered in this study: the entire study period (irrespective of wind) and only when the site was in downwind directions of port."*

Page 9, lines 30-32 and page 10, lines 1-2: all these should be in "Experimental" section.

Response:

Will be moved to "Experimental" section.

*Changes in manuscript*:

*It has been moved to "Experimental" section.*

Page 10, lines 8-12: Quite confused discussion. Re-written.

Response:

This section will be removed

*Changes in manuscript*:

*Discussions has been removed in manuscript.*

Page 10, lines 15-17: Here you finally evidenced the major contribution of ship emissions to NOx. But if you went to section 3.1, you claimed that NOx were mainly from land traffics while ship contribution is not important. The two explanations are contradictory.

Response:

The referee have not understood the point. The mentioned situation is after the land-based emission have been ruled out. It is nothing surprising to found that, if the emission in Shanghai is removed, the activities in port region will certainly be the dominant source of both $NO_x$ and $SO_2$.

*Changes in manuscript*:

*None*

Page 10, lines 19-20: Do you compare the absolute concentrations of these pollutants attributed to ship emissions in these two studies? This kind of comparison using percentage is very dangerous! I bet the total concentrations of these pollutants in these two studies are totally different.

Response:

The absolute contributions will be presented and discussed in next edition, together with suggestions raised by referee #2.

*Changes in manuscript*:

*Page 13,Line7-17:*

*"The relative contributions of PNCv from ship emission is apparently higher than PM2.5 on mass concentration. Previous study showed that the direct PM2.5 contribution from ship traffics lies within 1-8% range (Contini et al., 2011;Contini et al., 2015). Recent studies carried in Mediterranean region found that ship emission contributed 0.3-7.4% PM2.5 concentrations in port areas (Merico et al., 2016). Ship emission studies in Europe and other regions was reviewed, and its concluded that shipping traffics contributions to PM2.5 were in 1-14% range, with higher contributions with decreasing particle size (Viana et al., 2014). The calculated value of PM2.5 in the present site is within the reported ranges. Recently (Merico et al., 2017) compared ship traffic atmospheric impacts using inventories, experimental data and modelling approaches in Adriatic-Ionian port areas, and found that ships contributed 0.5-7.4% PM2.5 in these areas. The same study further found that ship traffics contribution to particle number concentrations (PNC) is 2-4 time larger than mass concentrations of PM2.5. The PNC is not currently measured, instead the size distributions, PNC contributions of vanadium particles in different sizes, as measured by SPAMS, apparently agrees with these previous work. "*

*Page 13,Line35-Page 14, Line 6:*

*"However, in an absolute sense, this study estimate that ship emissions contribute to 5.68 µg/m3 SO2, 3.00 µg/m3 NOx and 1.57 µg/m3 PM2.5 during the sampling period. These values are comparable or higher than the reported results in ports in other regions (Viana et al., 2014). For example, a previous study found that the ship emitted particles contributed 0.8 µg/m3 (primary particles) and 1.7 µg/m3 (secondary particles) in Bay of Algeciras (Viana et al., 2009). Due to the adjacency of the site to port, the calculated PM2.5 contribution could be largely deemed as primary for present site. The relative contributions of pollutants are partly compensated by the higher background pollution levels in this region."*

Lines 24-25: Did you do t-test to say this? ~4% vs. 5.9% is similar to me. Importantly, compare absolute concentrations. The comparison based on percentage does not make sense and can mislead readers.

Response:

No t-test.

*Changes in manuscript*:

*The same as previous change.*

Conclusions

The conclusion section is poorly written. It must be re-organized and re-written.

Response:

Conclusion will be revised.

*Changes in manuscript*:

*Page 15,line 6 - page line 16,line4:*

[revised manuscript text omitted]

6  **1.** *Wind rose of the port site during the study*

[Figure]

8       *Figure S1. Portside wind rose during the study period.*

10  **2.** *Explanation of identification method of ship emission particles*

11    The identification method relying on Vanadium signatures left a problem that this method

12  lose some portion of shipping emission particles which produce no or insignificant Vanadium

13  peaks (Xiao et al., 2018) . However, within the analyzing capability of SPAMS, Vanadium

14  signatures are still the most reliable indicator of shipping emission particles in a real ambient

15  condition. The present site in port area is both influenced by emission sources from the shipping

16  activities and traffics on land. Single particle signature from diesel vehicles has displayed some

17  similarity with shipping emission (especially for low Sulfur fuel oil, like MGO, IFO) because of the

18  resemblance in chemical composition between them (Toner et al., 2008;Xiao et al., 2018). In this

19  situation, to identify 'true' shipping emission particles from total particles will became difficult or

20  even impossible if we discard the reliable clue of Vanadium. In this supplementary material we

21  illustrate the wind roses of several particle clusters of similar composition with the only major

22    difference of Vanadium (Figure S1). From the figure it is clear that single particles with Vanadium

23    is an ideal indication of shipping emission source from port directions, while the exclusion of

24    Vanadium will only result an unwanted  interferences of particles from land sources.

25    Therefore,

26     in present study the online single particle measurement,

27    together with synchronous SO2 concentration, was utilized to indicate the occurrence of shipping

28    emission plumes, not to dig out every shipping emission particles.

[Figure]

30    *Figure S2. Mass spectra and wind roses of representative particle clusters with and without*
31    *Vanadium peaks.*

---

## Editor Decision (ED1)

EDITOR'S REVIEW OF MANUSCRIPT ACP-2018-737 – REFERRING TO MANUSCRIPT VERSION 03:

RESPONSE TO COMMENTS FROM REFEREE #2

The authors have sufficiently responded to the comments of Referee #2. The referee has accepted their responses without requesting further modifications.

RESPONSE TO COMMENTS FROM REFEREE #1:

To start with, I repeat my email from 19 November 2018, in which I responded to the authors with respect to their inadequate response to the comments raised by Referee #1. In that email I wrote:

"As the handling co-editor of your manuscript acp-2018-737, I am investigating the current situation of this manuscript after the public discussion phase has been closed and you have responded to the referee #2, resulting in an immediate response by referee #2 that he/she will reject your manuscript in case you do not respond in an appropriate manner. Actually the manuscript is stuck.

I analysed the review provided by referee #1 and found it detailed and appropriate, and written in the usual way, good reviews are provided. In particular I cannot see any unfriendly, insulting remarks.

If you wish to move on towards the successful publication of your manuscript in ACP, I urgently recommend that you respond to the reviewer's remarks in a more open way. In particular, referee #1 has requested better explanation and discussions of results in some places and your response was, e.g., "will be presented and discussed in next edition". However, such a response is insufficient since it does not allow the reviewer to assess whether his or her request has been answered sufficiently."

In the light of this request for clear response, I analysed the authors' responses to the comments by Referee #1, as submitted on 4 January 2019. Most of the minor comments by Referee #1 have been answered appropriately. However, major and more fundamental concerns have not been answered at all or in an inadequate manner. Therefore I request the following changes before the manuscript can be accepted.

Before going into details I want to remind the authors that referees and co-editors work on a voluntary basis. They spend extra time to help reviewing publications and ensure journal quality standards. The authors' responses to referees' comments and suggestions should always reflect the additional work load the referees and co-editors have accepted!

SPECIFIC CONCERNS RAISED BY REFEREE #1

Detailed points are listed below. For clarity I repeat from the authors' responses to the referees as follows: Referee #1 comments (in black), authors' responses (in blue), and changes to the manuscript (in red). Editor's remarks are added in red italic letters.

REFEREE #1: It is not clear how the components in particles such as vanadium were identified and quantified by the SPAMS. Detailed information is needed.

AUTHORS' RESPONSE: SPAMS identify particle composition, such as vanadium, in mass spectrometry method. In the ionization laser beam in SPAMS, the components in particles are ionized into ions carrying charge, then they are separated in the Time-of-Flight tube by atomic mass of the ion. Lighter ions, such as H+, transit fastest in TOF tube and produce peaks in shortest time. The 208Pb+ is heavier so that it reach the MS detector with longer time. This will result a spectrum sorted by ion atomic mass. Components of different atomic mass produce peaks in different position in mass spectrum. Vanadium in particles normally produce peaks at mass = 51V+ and 67 (VO+) and thus can be identified. This is the basics of MS and unnecessary to be included.

CHANGES IN MANUSCRIPT: None

*EDITOR'S REMARK:* The authors explain basic explanation of how a MS works, which indeed can be expected as basic knowledge. However, Referee #1 requests detailed information about the quantification of Vanadium concentrations. This needs to be added.

REFEREE #1: Page 5, lines 21-25: The discussion here is questionable. By looking at Figure 2, whenever ship plumes were captured, both NO and SO2/vanadium levels were high and correlated well. On what basis, the authors claimed the NOx in plumes reaching the site was aged? Using the NO/NO2 ratio in the plumes? Compared to the ratio measured in other countries and probably different type of ships? This is not convincing. Besides, NO2 is also emitted from shipping as a primary pollutant.

AUTHORS' RESPONSE: We have not tested NO/NO2 ratio in the exhaust because the observation was carried out in a station on land. The author also know that NO2, together with NO, is released as primary emissions. However their ratio, upon their emission into atmosphere, will subject to change quickly through the oxidation of NO into NO2 in the existence of ozone (O3), which is abundant in summer time. This was evidenced by the quickly reduced O3 level during plumes in Fig. 2 in manuscript. The referee seems to be doubtful about this reaction, which is the very basis that the NO-NO2-NOx analyzer is working on.

*CHANGES IN MANUSCRIPT*: *None*

*EDITOR'S REMARK: I fully agree with Referee #1 who requests are more careful discussion. The measured levels of $NO_x$ and the separation into NO and $NO_2$ cannot be used to identify ageing of ship plumes. Since the authors do not know the level of chemical processing (depending on $O_3$, meteorological conditions and atmospheric radiation) of ship exhaust in the marine boundary layer during transport to the sampling site, it seems to be more appropriate looking only at the sum parameter $NO_x$. This suggestion is supported by the results presented in Table 1. Looking at NO there is no statistically significant difference between in-plume, non-plume and port average cases. An almost similar statement holds for $NO_2$.*

REFEREE #1: Given that shipping emission is a major source of PM2.5, it is odd that no PM2.5 peaks were found during the ship plumes in Figure 2. The reason provided by the authors is quite confused. Is it because ship emits sub-micron particles or because the malfunction of the PM2.5 monitor?

AUTHORS' RESPONSE: The authors have stated that PM2.5 peaks was not as apparent as that of SO2 and NOx in plumes, and have not states that no PM2.5 peaks were found. It should be made clear that only a short period of data was shown in Fig.2 only with the purpose of illustrating temporal variation in plumes. In the last question the referee is suspecting that PM2.5 analyzer was in malfunction. The author would like to illustrate another longer period of data in Figure R2 (shown below) and let the referee make his judgement. The sharp peaks of SO2 in Figure R2 could help to locate the plumes. If the PM2.5 instrument is in malfunction, how did the PM2.5 instrument happened to malfunction only in plumes? In another aspect, the SPAMS particle number

concentration have shown good correlation with the PM2.5 measurement. How did the two instruments both malfunction? It sounds ridiculous.
*CHANGES IN MANUSCRIPT*: *None*

*EDITOR'S REMARK: The authors' response to the referee is not acceptable. Moreover, the entire discussion of the $PM_{2.5}$ mass concentrations is confusing and inconsistent. Referee #1 simply requests are more consistent presentation of the $PM_{2.5}$ case, which I do as well. We know from earlier observations that most of the PM emitted by ship engines is far smaller than 1.0 μm in diameter and will thus not contribute significantly to the $PM_{2.5}$ mass concentration. This fact needs to be pointed out very clearly. In that respect the results presented in Table 1 are consistent but the explanation in the text is confusing and requires careful revision. In particular, the explanation announced on Page 7 line 21 of the annotated manuscript is either missing or well hidden.*

REFEREE #1: The reason for absent SO2 is contradictory to Figure 2. If the ships complied with the new regulations, why would you still see SO2 peaks in ship plumes? This kind of discussion is misleading.
AUTHORS' RESPONSE: As stated above, the cases the SO2 peaks were absent were rare and will not affect significantly the result of the study. The new regulation only confine the Sulfur content in fuel, not eliminate the Sulfur from the fuel.
*CHANGES IN MANUSCRIPT*: *None*

*EDITOR'S REMARK: I fully agree with Referee #1. If the authors do (for 3% of the cases) not find a correlation between $SO_2$ and V, how have these cases been treated? Where they excluded from the analysis? Clarification is requested.*

REFEREE #1: It does not make sense to compare a site near sources with sites in a city without any detailed characteristics of the locations. It would be more meaningful to compare the ship emissions in this study with other similar studies conducted in Shanghai. In fact, there are a number of ship emission studies in this city.
AUTHORS' RESPONSE: The comment is not acceptable. We made comparison between air pollution level in a port site with that at the urban area of the same city, How did it make non sense? The referee had better suggest a study which he think make more sense.
*CHANGES IN MANUSCRIPT*: *None*

*EDITOR'S REMARK: This response it not acceptable. It is not the duty of the referees to make the authors familiar with current literature! I fully support Referee #1 request for a comparison of the presented results with those from other studies in Shanghai port.*

REFEREE #1: the explanation of low PM2.5 levels at the port site is not convincing at all. Why would other pollutants from shipping emissions be higher if this was caused by clean air?
AUTHORS' RESPONSE: The slight lower PM2.5 concentration at the port site is a fact. The author only postulated possible explanations.
*CHANGES IN MANUSCRIPT*: *None*

*EDITOR'S REMARK: The authors have to provide consistent explanations for their observations. There is no reason why $PM_{2.5}$ should be influenced by clean air why gaseous pollutants will not. Another explanation is needed here.*
*Another remarks concerns Figure 6, which also shows the inconsistency of presentation of PM results. In the figure caption, the authors indicate that particle number concentrations are presented. In the figure itself, however, the y-axis title (left panel) indicates "Particles/hour". This is definitely not an*

*adequate unit for particle number concentrations! Furthermore, the y-axis of the upper right panel indicates "Particle number", whereas the authors mean particle number concentration in units of $cm^{-3}$. Both issues need to be clarified.*

*MINOR ISSUES:*

- *The company's name providing the instruments is THERMO SCIENTIFIC, not THERMAL SCIENTIFIC.*
- *Furthermore, another language editing is mandatory before publication.*

---

## Author Response (AR2)

**Authors' response to editor's remarks**
**ON MANUSCRIPT ACP-2018-737 – REFERRING TO MANUSCRIPT VERSION 03**

*(we present our responses in blue, original text in black. Between items of remarks & responses we inserted a separation line)*

REFEREE #1: It is not clear how the components in particles such as vanadium were identified and quantified by the SPAMS. Detailed information is needed.

AUTHORS' RESPONSE: SPAMS identify particle composition, such as vanadium, in mass spectrometry method. In the ionization laser beam in SPAMS, the components in particles are ionized into ions carrying charge, then they are separated in the Time-of-Flight tube by atomic mass of the ion. Lighter ions, such as H+, transit fastest in TOF tube and produce peaks in shortest time. The 208Pb+ is heavier so that it reach the MS detector with longer time. This will result a spectrum sorted by ion atomic mass. Components of different atomic mass produce peaks in different position in mass spectrum. Vanadium in particles normally produce peaks at mass = 51V+ and 67 (VO+) and thus can be identified. This is the basics of MS and unnecessary to be included.

CHANGES IN MANUSCRIPT: None

*EDITOR'S REMARK: The authors explain basic explanation of how a MS works, which indeed can be expected as basic knowledge. However, Referee #1 requests detailed information about the quantification of Vanadium concentrations. This needs to be added.*

**AUTHORS' RESPONSE TO REMARKS FROM EDITOR:**

We have added further descriptions on how vanadium particles were identified and quantified in SPAMS in section 2.3.

**CHANGES IN MANUSCRIPT:**

(Section 2.3, paragraph 2)

*"Specific composition in particles, such as vanadium, is identified by their characteristic mass peaks in particle spectra. Particles producing vanadium peaks were labelled as vanadium particles. SPAMS quantifies their concentrations in a semi-quantitative manner through the number of detected particles in a specific duration of time. Considering that the aerosol flow was introduced into SPAMS at fixed flow rate (0.1L/min), the detected particle numbers (or particle detecting velocity) could be utilized as indication of ambient particle concentrations. In ambient sampling it was shown that the particle numbers in SPAMS were positively correlated with ambient $PM_{2.5}$ concentrations ($R^2$=0.69 in this study). In present study, we use particle detecting speed of vanadium containing particles as a metric of their concentrations. To derive ambient particle number concentrations from SPAMS particle numbers, we need to consider the efficiency issues of SPAMS on AFL transmission, laser detection and laser ionization(Wenzel et al., 2003)."*
* * *
REFEREE #1: Page 5, lines 21-25: The discussion here is questionable. By looking at Figure 2, whenever ship plumes were captured, both NO and SO2/vanadium levels were high and correlated well. On what basis, the authors claimed the NOx in plumes reaching the site was aged? Using the NO/NO2 ratio in the plumes? Compared to the ratio measured in other countries and probably different type of ships? This is not convincing. Besides, NO2 is also emitted from shipping as a primary pollutant.

AUTHORS' RESPONSE: We have not tested NO/NO2 ratio in the exhaust because the observation was carried out in a station on land. The author also know that NO2, together with NO, is released as primary emissions. However their ratio, upon their emission into atmosphere, will subject to change quickly through the oxidation of NO into NO2 in the existence of ozone (O3), which is abundant in summer time. This was evidenced by the quickly reduced O3 level during plumes in Fig. 2 in manuscript. The referee seems to be doubtful about this reaction, which is the very basis that the NO-NO2-NOx analyzer is working on.

*CHANGES IN MANUSCRIPT: None*

EDITOR'S REMARK: I fully agree with Referee #1 who requests are more careful discussion. The measured levels of NOx and the separation into NO and NO2 cannot be used to identify ageing of ship plumes. Since the authors do not know the level of chemical processing (depending on O3, meteorological conditions and atmospheric radiation) of ship exhaust in the marine boundary layer during transport to the sampling site, it seems to be more appropriate looking only at the sum parameter NOx. This suggestion is supported by the results presented in Table 1. Looking at NO there is no statistically significant difference between in-plume, non-plume and port average cases. An almost similar statement holds for NO2.

**AUTHORS' RESPONSE TO REMARKS FROM EDITOR:**
Thanks the editor's remark. We admit the editor and reviewer's opinion that other factors like atmospheric radiation could also influence NO and NO2 concentrations. In ultraviolet radiation in sunlight the NO2 disintegrate into NO and O atom. Meanwhile, the O atom and O2 molecules also form O3, which in turn could oxidize NO. These are cyclic reactions which will reach their equilibrium if no extra NO and NO2 is introduced. In fresh emitted plumes the NO-NO2-O3 equilibrium is interrupted, usually resulting reduction of O3 concentration, as observed in sites elsewhere (Alföldy et al., 2013;Merico et al., 2016).

Since editor and the reviewer #1 suggested that using of NO/NO$_2$ ratio is not careful enough, we have searched literatures on related topics and found that the NO2 ratio was used by several studies(Alföldy et al., 2013;Kurtenbach et al., 2016). The NO2 ratio is defined as the ratio between NO2 and the summed NOx (NO+NO2) parameter, which was the suggested parameter by the editor. We calculated the NO2 ratio during plumes periods in this study. This was done by firstly converting the gas concentrations to molar unit (conforming to existing studies), and then the NO and NO2 concentrations were calculated by subtracting the background levels during plumes, similar to the case of SO2. The distribution of the calculated NO2 ratio in this study is shown in Figure 1, where the NO2 ratios of a comparable study is plotted for comparison.

The distribution of the NO2 ratio in present study showed several modes. The largest mode occurred at about 0.2 (20%). This mode was also observed in the comparison study, which was recognized as fresh engine emissions of ships. The major difference between two studies is that a significant fraction of NO2 ratios occurred in larger values (> 0.4) in this study, which was not observed in (Alföldy et al., 2013). Initially the plumes of larger NO2 ratios were thought to be emitted by specific type of ships. After we correlating the NO2 ratio with ambient O3 levels, however, there exist obvious positive correlation between NO2 ratio and ambient O3 concentrations, as shown in right panel in Figure 2. The relation between NO2 ratio and O3 became more obvious in higher ambient O3 levels. It is noted that the correlations between NO2 ratio and other parameters including wind directions, ambient temperature, humidity were very weak. This result is indicating that the higher NO2 ratio of some plumes were not due to the emission characteristics of ships, but due to the transformation of NO to NO2 in the ambient, for if the higher NO2 ratio were caused by higher NO2 emission at their discharges and no ambient transformation occurred, then there will be no reason to observe the dependence of NO2 ratio on an ambient condition of O3. This could be an evidence of oxidation of primary NO by O3, therefore ship emitted NO had contributed to the depletion of ozone.

[Figure]

Figure 1. The NO$_2$ ratio distribution during plumes in this study and a similar study (left) and the plot of NO$_2$ ratio against ambient ozone concentration during plumes periods (right).

**CHANGES IN MANUSCRIPT:**
(Section 3.1, paragraph 5-6)
*"In general the ozone concentrations in the port site were lower than Shanghai urban region by 13-33%. To inspect whether the O3 depletion was related to the oxidation of primary NO emissions in the port site, we calculated the NO2 ratios to analyse NOx composition in plumes. The NO2 ratio is defined as the ratio between NO2 and the NOx (NO+NO2), which was used by several relevant characterizations of ship emissions (Alföldy et al., 2013;Kurtenbach et al., 2016). Before the calculation of NO2 ratio we firstly converted NO and NO2 mass concentrations to molar unit, and then the background NO and NO2 levels were subtracted to make sure peaks were due to plumes. The distribution of the NO2 ratio in this study was shown in Figure 3, where the NO2 ratio distribution from ship plumes in another study was compared. The distribution of the NO2 ratios in present study showed several modes. The largest mode occurred at about 0.2 (20%). Obviously this mode was also present in the comparison study (Alföldy et al., 2013), which was recognized as fresh engine emissions from ships. A major difference between two studies is that significant fraction of NO2 ratios occurred in larger range (> 0.4) in present study, which was not observed in Alföldy et al., 2013. The larger NO2 ratios were once thought to be emitted from unidentified type of ships. When we correlated the NO2 ratio with ambient O3 concentrations, however, we found there was obvious positive correlation between them, as shown in right panel in Figure 3. This result suggests that the higher NO2 ratio of some plumes were not due to the emission characteristics of ships, but due to the transformation of NO to NO2 in the ambient, for if the NO2 ratio were higher at the discharges and no ambient transformation occurred, then there will be no reason to observe the dependence of NO2 ratio on an ambient condition of O3. This is an evidence that the primary NO emission (from ships or on-road traffics ) had contributed to the O3 depletion in this area. ".*

REFEREE #1: Given that shipping emission is a major source of PM2.5, it is odd that no PM2.5 peaks were found during the ship plumes in Figure 2. The reason provided by the authors is quite confused. Is it because ship emits sub-micron particles or because the malfunction of the PM2.5 monitor?

AUTHORS' RESPONSE: The authors have stated that PM2.5 peaks was not as apparent as that of SO2 and NOx in plumes, and have not states that no PM2.5 peaks were found. It should be made clear that only a short period of data was shown in Fig.2 only with the purpose of illustrating temporal variation in plumes. In the last question the referee is suspecting that PM2.5 analyzer was in malfunction. The author would like to illustrate another longer period of data in Figure R2 (shown below) and let the referee make his judgement. The sharp peaks of SO2 in Figure R2 could help to locate the plumes. If the PM2.5 instrument is in malfunction, how did the PM2.5 instrument happened to malfunction only in plumes? In another aspect, the SPAMS particle number concentration have shown good correlation with the PM2.5 measurement. How did the two instruments both malfunction? It sounds ridiculous.

CHANGES IN MANUSCRIPT: *None*

EDITOR'S REMARK: *The authors' response to the referee is not acceptable. Moreover, the entire discussion of the PM2.5 mass concentrations is confusing and inconsistent. Referee #1 simply requests are more consistent presentation of the PM2.5 case, which I do as well. We know from earlier observations that most of the PM emitted by ship engines is far smaller than 1.0 μm in diameter and will thus not contribute significantly to the PM2.5 mass concentration. This fact needs to be pointed out very clearly. In that respect the results presented in Table 1 are consistent but the explanation in the text is confusing and requires careful revision. In particular, the explanation announced on Page 7 line 21 of the annotated manuscript is either missing or well hidden.*

**AUTHORS' RESPONSE TO REMARKS FROM EDITOR:**
We have removed relevant discussions on PM2.5 to Section 3.1, paragraph 7

**CHANGES IN MANUSCRIPT:**

(Section 3.1, paragraph 7)
*"For particulate matter, the PM2.5 concentrations in port area were slightly lower than Shanghai city, although PNCv in plumes were times higher than non-plumes (Table 1). Longer period of PM2.5 data suggested the lower PM2.5 concentration is a general trend in this port site. This trend is not unique to the port regions because we observed it in other coastal area as well, which is readily observed in PM2.5 spatial distribution of Shanghai (Figure S4 in supplementary file). In the spatial distribution there was a general trend of decreasing PM2.5 concentrations from inner to coastal areas in Shanghai. This fact is assumed to be caused by the dispersion or advection of clean air from the sea. The primary PM in portside from ship emissions are mostly ultrafine particles, with mass emission factors much smaller than NOx and SO2 (Zhang et al., 2017). Therefore the primary PM from ships or other traffics could not contribute significantly to ambient PM mass concentrations. The vanadium particle number fractions in total particles in SPAMS were obviously larger (6.7 % on average) in portside than urban areas in Shanghai (1-2 %) (Liu et al., 2017)."*
* * *
REFEREE #1: The reason for absent SO2 is contradictory to Figure 2. If the ships complied with the new regulations, why would you still see SO2 peaks in ship plumes? This kind of discussion is misleading.

AUTHORS' RESPONSE: As stated above, the cases the SO2 peaks were absent were rare and will not affect significantly the result of the study. The new regulation only confine the Sulfur content in fuel, not eliminate the Sulfur from the fuel.

*CHANGES IN MANUSCRIPT*: *None*

EDITOR'S REMARK: I fully agree with Referee #1. If the authors do (for 3% of the cases) not find a correlation between SO2 and V, how have these cases been treated? Where they excluded from the analysis? Clarification is requested.

**AUTHORS' RESPONSE TO REMARKS FROM EDITOR:**
We are sorry to forget to make this clarifications when replying #2 Referee's related questions. In data analysis, the treatment of those cases were determined by checking the prevailing wind of that time. If the wind directions of that time was favorable, we will classify the plumes as from ship emissions, if not, they will be just excluded from analysis, because we are not sure about their real sources. The results in table I and table 2 were recalculated using this treatment when we prepared the manuscript version 3.

**CHANGES IN MANUSCRIPT:**
(Section 3.1, second paragraph, Line 8-9)
*"To identify plumes, we excluded the possible industries influences by limiting the prevailing winds only to port directions."*
* * *
REFEREE #1: It does not make sense to compare a site near sources with sites in a city without any detailed characteristics of the locations. It would be more meaningful to compare the ship emissions in this study with other similar studies conducted in Shanghai. In fact, there are a number of ship emission studies in this city.

AUTHORS' RESPONSE: The comment is not acceptable. We made comparison between air pollution level in a port site with that at the urban area of the same city, How did it make non sense? The referee had better suggest a study which he think make more sense.

*CHANGES IN MANUSCRIPT*: *None*

EDITOR'S REMARK: This response it not acceptable. It is not the duty of the referees to make the authors familiar with current literature! I fully support Referee #1 request for a comparison of the presented results with those from other studies in Shanghai port.

**AUTHORS' RESPONSE TO REMARKS FROM EDITOR:**
We have added relevant discussions with results from similar study in Shanghai in section 3.1, paragraph

4.

**CHANGES IN MANUSCRIPT:**
(section 3.1, paragraph 4)
*"Generally the concentrations of SO2 and NOx in the port site is 40~70% higher than Shanghai city (Table 1). The SO2 concentrations in non-plume periods were comparable with that in Shanghai city, irrespective of wind directions, therefore the non-plume SO2 can be recognized as background SO2 in this area. Contrastingly, the NOx concentrations showed obvious dependence on wind directions in non-plumes, whose concentrations were higher when inland wind prevails, suggesting the importance of land-based emissions to port in coastal areas. In a similar ambient observation at Yangshan port, (Zhao et al., 2013) obtained the average concentration of 29.4 and 63.7 μg/m3 for SO2 and NO2 respectively, higher than the present level of 15.6 μg/m3 (SO2), 53.2μg/m3 (NO2). Noting that the SO2 and NO2 were only intermittently measured for about 20 days in that study (May and August, 10 days each month). Therefore it is not feasible to make direct comparison. In plume period, the SO2 maximum hourly concentration in Yangshan (119.0 μg/m3) were close to present study (124 μg/m3); Due to land-based emissions, the NO2 maximum hourly concentration in Waigaoqiao port (260 μg/m3) is higher than Yangshan port (199.8 μg/m3)."*
* * *
REFEREE #1: the explanation of low PM2.5 levels at the port site is not convincing at all. Why would other pollutants from shipping emissions be higher if this was caused by clean air?
AUTHORS' RESPONSE: The slight lower PM2.5 concentration at the port site is a fact. The author only postulated possible explanations.
CHANGES IN MANUSCRIPT: *None*
EDITOR'S REMARK: *The authors have to provide consistent explanations for their observations. There is no reason why PM2.5 should be influenced by clean air why gaseous pollutants will not. Another explanation is needed here.*

**AUTHORS' RESPONSE TO REMARKS FROM EDITOR:**
The lower PM2.5 concentrations in the port site than Shanghai city were not limited only to the port site. In air monitoring networks in Shanghai, we found the general trend that the coast sites have lower PM2.5 concentrations than the inner sites in west. In Figure 2 we prepared two maps showing the PM2.5 spatial distributions in Shanghai areas during the study period, and the year 2016. In either cases, the average PM2.5 concentrations in coastal areas in east were lower than the inner areas, which could only be explained by the clean air advection or dispersion from sea areas. The lower PM2.5 concentration in the port site was within this trend.
The higher NOx and SO2 levels were not conflicting the lower PM2.5 concentrations in port site, for the mass emission factors (EF) of NOx and SO2 from combustion sources (either on-road traffics or ships) is generally several times larger than that of PM. Many published EFs of PM, $SO_2$,$NO_x$ support this. To quote as example, in china the measured EFs of NOx and SO2 from the main engines of ships, were 12.7-14.3 and 10.3-13.5 g/Kwh, respectively, while the EFs of PM were in 0.2-1.7 g/Kwh range (Zhang et al., 2017). The much less EFs of PM than gaseous pollutants were also applicable to auxiliary engines in ships and diesel trucks (Zavala et al., 2017). Obviously the insignificant primary PM emission in portside have not compensated the cleaning impacts from the sea, resulted lower $PM_{2.5}$ concentration than Shanghai city. We added the PM2.5 spatial distribution maps into supplementary file, and moved this section to section 3.1, paragraph 7.

**CHANGES IN MANUSCRIPT:**
(Section 3.1, in paragraph 7 )
*"For particulate matter, the PM2.5 concentrations in port area were slightly lower than Shanghai city, although PNCv in plumes were times higher than non-plumes (Table 1). Longer period of PM2.5 data suggested the lower PM2.5 concentration is a general trend in this port site. This trend is not unique to the port regions because we observed it in other coastal area as well, which is readily observed in PM2.5*

*spatial distribution of Shanghai (Figure S4 in supplementary file). In the spatial distribution there was a general trend of decreasing PM2.5 concentrations from inner to coastal areas in Shanghai. This fact is assumed to be caused by the dispersion or advection of clean air from the sea. The primary PM in portside from ship emissions are mostly ultrafine particles, with mass emission factors much smaller than NOx and SO2 (Zhang et al., 2017). Therefore the primary PM from ships or other traffics could not contribute significantly to ambient PM mass concentrations. The vanadium particle number fractions in total particles in SPAMS were obviously larger (6.7 % on average) in portside than urban areas in Shanghai (1-2 %) (Liu et al., 2017)."*

[Figure]

Figure 2. Spatial distributions of PM$_{2.5}$ in Shanghai area in the study period (left) and in 2016 (right).

Another remarks concerns Figure 6, which also shows the inconsistency of presentation of PM results. In the figure caption, the authors indicate that particle number concentrations are presented. In the figure itself, however, the y-axis title (left panel) indicates "Particles/hour". This is definitely not an adequate unit for particle number concentrations! Furthermore, the y-axis of the upper right panel indicates "Particle number", whereas the authors mean particle number concentration in units of cm-3. Both issues need to be clarified.

**AUTHORS' RESPONSE TO REMARKS FROM EDITOR:**
We have revised the caption of figure 6 (now figure 7) to make its meaning clear.

**CHANGES IN MANUSCRIPT:**
*(Figure 7 caption).*
*"Figure 7: Temporal trends of particle numbers detected per hour by SPAMS of four fresh vanadium particle types (Left panel); The upper right panel is the number-size distribution of the 4 types, with the y-axis representing particle numbers detected at each size bin in the entire study. The Lower right panel is obtained by normalizing the particles numbers of 4 types to give their relative contributions at each size."*

MINOR ISSUES:
The company's name providing the instruments is THERMO SCIENTIFIC, not THERMAL SCIENTIFIC.
**AUTHORS' RESPONSE TO REMARKS FROM EDITOR:**
We have made this correction. Thanks.

**CHANGES IN MANUSCRIPT:**
(Section 2.2, first paragraph, Line 1-2)
*"From Jun-21 to Sep-21, 2016, the portside gaseous pollutants NO-NO2-NOx, SO2, and O3 were monitored continuously with a suit of Thermo Scientific analyzers (NO-NO2-NOx, model 42i; SO2, model 43i; O3, 49i)."*

Furthermore, another language editing is mandatory before publication.
**AUTHORS' RESPONSE TO REMARKS FROM EDITOR:**
We have made another language editing throughout the manuscript.

**CHANGES IN MANUSCRIPT:**
*(see blue text in the revised manuscript).*

[Figure]

Figure S1. Portside wind rose during the study period.

*2. Explanation of identification method of ship emission particles*

The identification method relying on Vanadium signatures left a problem that this method lose some portion of shipping emission particles which produce no or insignificant Vanadium peaks (Xiao et al., 2018) . However, within the analyzing capability of SPAMS, Vanadium signatures are still the most reliable indicator of shipping emission particles in a real ambient condition. The present site in port area is both influenced by emission sources from the shipping activities and traffics on land. Single particle signature from diesel vehicles has displayed some similarity with shipping emission (especially for low Sulfur fuel oil, like MGO, IFO) because of the resemblance in chemical composition between them (Toner et al., 2008;Xiao et al., 2018). In this situation, to identify 'true' shipping emission particles from total particles will became difficult or even impossible if we discard the reliable clue of Vanadium. In this supplementary material we illustrate the wind roses of several particle clusters of similar composition with the only major difference of Vanadium (Figure S1). From the figure it is clear that single particles with Vanadium is an ideal indication of shipping emission source from port directions, while the exclusion of

Vanadium will only result an unwanted interferences of particles from land sources. Therefore, in present study the online single particle measurement, together with synchronous SO2

concentration, was utilized to indicate the occurrence of shipping emission plumes, not to dig out every shipping emission particles.

[Figure]

Figure S2. Mass spectra and wind roses of representative particle clusters with and without vanadium peaks.

**3. *Temporal variation of PM$_{2.5}$, vanadium particles, SO$_2$ concentrations in the portside during a***

***period from Jul-03 to Jul-06.***

[Figure]

Figure S3. Temporal variations of SO$_2$, PM$_{2.5}$, vanadium particles numbers in port site and PM$_{2.5}$

in Shanghai city from Jul-03 to Jul-06.

**4. *Spatial distribution of PM$_{2.5}$ concentrations in Shanghai area***

[Figure]

Figure S4. Spatial distributions of PM$_{2.5}$ in Shanghai area in the study period (left) and in 2016

(right).

---

## Author Response (AR3)

**Authors' response to the request of technical corrections from editor on manuscript ACP-2018-737**

*(the editor's requests are in black fond, authors' responses in blue)*

The corrections are requested for the following items (page and line numbers refer to the annotated manuscript attached to your latest Author's Response):

1. page 7, line 33: Do you mean here ""peak searching and clustering algorithm" ?

Author's response:

The editor may refer to page 5, line 19 in the manuscript. Yes, "peak searching and clustering algorithm" is more appropriate here, we have replaced the original text.

2. Figure 2: You may want to use the character "μ" for the axis titles.

Author's response:

We have replaced the letter "u" with "μ" in Figure 2, 3 and Figure S3. Thanks.

3. Figure 3, right panel: I suggest using single - line axis titles.

Author's response:

The y-axis title of figure 3 is corrected as suggested.

4. Figure 5, top panel: Please report the unit on the y - axis for the particle size distribution. If it is similar to Fig. 7 I suggest adding "with the y-axis representing particle numbers detected at each size bin at constant sample flow in the entire study" or adequate. I propose the same modification to Fig. 7 capture.

Author's response:

The captures of Figure 5 and Figure 7 are revised accordingly.

A marked version of manuscript highlighting relevant corrections is attached in following pages (corrections in cyan color).

[revised manuscript text omitted]

*1. Wind rose of the port site during the study*

[Figure]

           Figure S1. Portside wind rose during the study period.

**2.** *Explanation of identification method of ship emission particles*

The identification method relying on Vanadium signatures left a problem that this method lose some portion of shipping emission particles which produce no or insignificant Vanadium peaks (Xiao et al., 2018) . However, within the analyzing capability of SPAMS, Vanadium signatures are still the most reliable indicator of shipping emission particles in a real ambient condition. The present site in port area is both influenced by emission sources from the shipping activities and traffics on land. Single particle signature from diesel vehicles has displayed some similarity with shipping emission (especially for low Sulfur fuel oil, like MGO, IFO) because of the resemblance in chemical composition between them (Toner et al., 2008;Xiao et al., 2018). In this situation, to identify 'true' shipping emission particles from total particles will became difficult or even impossible if we discard the reliable clue of Vanadium. In this supplementary material we illustrate the wind roses of several particle clusters of similar composition with the only major difference of Vanadium (Figure S2). From the figure it is clear that single particles with Vanadium is an ideal indication of shipping emission source from port directions, while the exclusion of

Vanadium will only result an unwanted interferences of particles from land sources. Therefore, in present study the online single particle measurement, together with synchronous SO2

concentration, was utilized to indicate the occurrence of shipping emission plumes, not to dig out every shipping emission particles.

[Figure]

Figure S2. Mass spectra and wind roses of representative particle clusters with and without vanadium peaks.

**3. *Temporal variation of PM$_{2.5}$, vanadium particles, SO$_2$ concentrations in the portside during a period from Jul-03 to Jul-06.***

[Figure]

Figure S3. Temporal variations of SO$_2$, PM$_{2.5}$, vanadium particles numbers in port site and PM$_{2.5}$ in Shanghai city from Jul-03 to Jul-06.

**4. *Spatial distribution of PM$_{2.5}$ concentrations in Shanghai area***

[Figure]

Figure S4. Spatial distributions of PM$_{2.5}$ in Shanghai area in the study period (left) and in 2016 (right).